EMBO
Molecular Medicine

# Cysteamine–bicalutamide combination therapy corrects proximal tubule phenotype in cystinosis

Amer Jamalpoor[1] [ID], Charlotte AGH van Gelder[2,3,†], Fjodor A Yousef Yengej[4,5,†], Esther A Zaal[2,6], Sante P Berlingerio[7], Koenraad R Veys[7], Carla Pou Casellas[1], Koen Voskuil[1], Khaled Essa[1], Carola ME Ammerlaan[4,5], Laura Rita Rega[8], Reini EN van der Welle[9] [ID], Marc R Lilien[10], Maarten B Rookmaaker[5], Hans Clevers[4] [ID], Judith Klumperman[9] [ID], Elena Levtchenko[7], Celia R Berkers[2,6], Marianne C Verhaar[5], Maarten Altelaar[2,3] [ID], Rosalinde Masereeuw[1] [ID] & Manoe J Janssen[1,*] [ID]

## Abstract

Nephropathic cystinosis is a severe monogenic kidney disorder caused by mutations in *CTNS*, encoding the lysosomal transporter cystinosin, resulting in lysosomal cystine accumulation. The sole treatment, cysteamine, slows down the disease progression, but does not correct the established renal proximal tubulopathy. Here, we developed a new therapeutic strategy by applying omics to expand our knowledge on the complexity of the disease and prioritize drug targets in cystinosis. We identified alpha-ketoglutarate as a potential metabolite to bridge cystinosin loss to autophagy, apoptosis and kidney proximal tubule impairment in cystinosis. This insight combined with a drug screen revealed a bicalutamide–cysteamine combination treatment as a novel dual-target pharmacological approach for the phenotypical correction of cystinotic kidney proximal tubule cells, patient-derived kidney tubuloids and cystinotic zebrafish.

**Keywords** alpha-ketoglutarate; Bicalutamide combination therapy; cysteamine; cystinosis; renal Fanconi syndrome
**Subject Categories** Genetics, Gene Therapy & Genetic Disease; Organelles; Urogenital System

## Introduction

Nephropathic cystinosis (MIM219800) is a lysosomal storage disease (LSD) caused by mutations in *CTNS*, a gene that codes for the lysosomal cystine/proton symporter cystinosin (Town *et al*, 1998). The loss of cystinosin leads to the lysosomal accumulation of cystine throughout the body and causes irreversible damage to various organs, particularly the kidneys (Cherqui & Courtoy, 2017). The first clinical signs develop during infancy (age of ~6 months) in the form of renal Fanconi syndrome and over time patients develop chronic kidney disease and finally renal failure in the first or second decade of life (Gahl *et al*, 2002). For the past decades, great efforts have been directed towards reducing cellular cystine accumulation in cystinotic patients. The cystine-depleting drug cysteamine can efficiently lower the lysosomal cystine levels and postpone disease progression. However, it poses serious side effects and does not correct the established proximal tubulopathy associated with cystinosis (Gahl *et al*, 2002; Brodin-Sartorius *et al*, 2012).

Recent discoveries in lysosomal gene expressions and their regulatory networks have provided new insights in the studies of LSDs, including cystinosis (Gabande-Rodriguez *et al*, 2014; Lieberman *et al*, 2012; Canonico *et al*, 2016). Several studies based on *in vitro* and *in vivo* models have demonstrated that the loss of cystinosin is indeed associated with disrupted lysosomal autophagy dynamics, accumulation of distorted mitochondria, and increased reactive oxygen species (ROS) levels, leading to abnormal proliferation and dysfunction of proximal tubule cells (Levtchenko *et al*, 2005;

1 Division of Pharmacology, Department of Pharmaceutical Sciences, Faculty of Science, Utrecht University, Utrecht, The Netherlands
2 Biomolecular Mass Spectrometry and Proteomics, Bijvoet Center for Biomolecular Research and Utrecht Institute for Pharmaceutical Sciences, Utrecht University, Utrecht, The Netherlands
3 Netherlands Proteomics Center, Utrecht, The Netherlands
4 Hubrecht Institute-Royal Netherlands Academy of Arts and Sciences and University Medical Center Utrecht, Utrecht, The Netherlands
5 Department of Nephrology and Hypertension, University Medical Center Utrecht, Utrecht, The Netherlands
6 Division of Cell Biology, Cancer & Metabolism, Department of Biomolecular Health Sciences, Faculty of Veterinary Medicine, Utrecht University, Utrecht, The Netherlands
7 Department of Pediatric Nephrology & Growth and Regeneration, University Hospitals Leuven & KU Leuven, Leuven, Belgium
8 Renal Diseases Research Unit, Genetics and Rare Diseases Research Area, Bambino Gesù Children's Hospital, IRCCS, Rome, Italy
9 Section Cell Biology, Center for Molecular Medicine, University Medical Center Utrecht, Utrecht University, Utrecht, The Netherlands
10 Department of Pediatric Nephrology, Wilhelmina Children's Hospital, University Medical Centre Utrecht, Utrecht, The Netherlands
*Corresponding author. Tel: +31 30 2531599; E-mail: m.j.janssen1@uu.nl
†These authors contributed equally to this work.

Sansanwal *et al*, 2010; Sansanwal & Sarwal, 2012; Ivanova *et al*, 2015; Festa *et al*, 2018). Regardless of the observed cellular defects associated with cystinosis, the mechanisms linking cystinosin loss, lysosomal defects and epithelial dysfunction remain unknown, hampering the development of an enduring intervention to combat the disease.

In this work, we aimed to develop a novel pharmacological strategy to treat cystinosis by correcting its kidney tubule phenotype. By combining CRISPR/Cas9 technology and a dual proteomics and metabolomics approach, we identified alpha-ketoglutarate (αKG) as a novel metabolite affected in cystinosis which is also known to play a pivotal role in the regulation of autophagy and apoptosis. Using this knowledge, we discovered the bicalutamide–cysteamine combination treatment as a novel strategy for the phenotypical correction of cystinotic proximal tubule cells. This approach was validated further in cystinotic patient-derived kidney tubuloids and in cystinotic zebrafish, demonstrating the therapeutic potential for this combination therapy to treat patients with cystinosis.

## Results

### CRISPR-generated $CTNS^{-/-}$ ciPTEC display increased cystine accumulation and impaired lysosomal autophagy dynamics

To understand the effect of *CTNS* loss on proximal tubule cell function, we used two well-characterized conditionally immortalized proximal tubule epithelial cell (ciPTEC) lines previously generated from urine samples of a cystinosis patient (ciPTEC $CTNS^{Patient}$) and an age-matched healthy volunteer (ciPTEC $CTNS^{WT}$) (Wilmer *et al*, 2005; Wilmer *et al*, 2010; Wilmer *et al*, 2011). However, as these cell lines were obtained from different individuals, the genetic variation is likely to introduce phenotypical changes independent of *CTNS* loss. To overcome this limitation, we created an isogenic *CTNS*-deficient cell line (ciPTEC $CTNS^{-/-}$) by specifically knocking out *CTNS* in the control ciPTEC. A guide RNA (gRNA) targeting exon 4 of the *CTNS* gene was used to introduce mutations by CRISPR/Cas9 in the ciPTEC $CTNS^{WT}$. After cell sorting and subsequent clonal cell expansion, three clones with biallelic mutations (lines 3, 7 and 35) were initially selected and all displayed a similar phenotype (Fig EV1), therefore, line 3 (hereafter referred to as $CTNS^{-/-}$) was

used for subsequent experiments. This model enables direct evaluation of the effect of *CTNS* loss on proximal tubule epithelial cells independent of chronic exposure to other disease related changes in the body. As a reference, we also included the non-isogenic patient-derived cystinotic ciPTEC $CTNS^{Patient}$ line bearing the homozygous 57-kb *CTNS* deletion (Peeters *et al*, 2011; Wilmer *et al*, 2011). In accordance with the pathophysiology of the disease, $CTNS^{-/-}$ cells displayed significantly increased levels of cystine compared to control $CTNS^{WT}$ cells (5.19 ± 0.30 versus 0.05 ± 0.02 nmol/mg protein), comparable to $CTNS^{Patient}$ cells (Fig 1A). Next, we evaluated the effect of *CTNS* loss on mammalian target of rapamycin complex 1 (mTORC1) (Ivanova *et al*, 2015; Andrzejewska *et al*, 2016; Hollywood *et al*, 2020). Under normal conditions, mTORC1 is bound to the lysosomes and is responsible for regulating a wide range of cellular processes, including autophagy (Laplante & Sabatini, 2013; Martina & Puertollano, 2013). In the presence of nutrients (standard medium, fed condition), mTOR located to the lysosomal membranes of $CTNS^{WT}$ cells (Fig EV2). Upon starvation (−AA), mTOR was released from the lysosomes and relocalized upon reintroduction of nutrients. In contrast, in $CTNS^{-/-}$ and $CTNS^{Patient}$ cells the fed condition revealed a less pronounced colocalization, and no difference was seen between the fed and starved condition (Fig EV2). Accurate measurement of the lysosomal size and quantifying the colocalization with mTOR was not feasible due to high level of clusterization of endosomal vesicles. We further evaluated mTOR activity in the cells by tracking the subcellular localization of transcription factor EB (TFEB). If mTOR is deactivated, unphosphorylated TFEB can translocate to the nucleus, where it regulates gene transcription and activates autophagy. A ~2.5-fold increase in TFEB nuclear translocation was observed after transfection with TFEB-GFP in $CTNS^{-/-}$ and $CTNS^{Patient}$ cells compared to $CTNS^{WT}$ cells (Fig 1B). As TFEB will downregulate its own expression after activation (Rega *et al*, 2016), we found that the endogenous *TFEB* mRNA expression was also reduced in $CTNS^{-/-}$ and $CTNS^{Patient}$ cells (twofold) compared to control cells (Fig 1C). During autophagy, LC3-II is recruited to autophagosomes and p62/SQSTM1 is degraded after the fusion of autophagosomes with the lysosomes (Tanida *et al*, 2008). Although levels of LC3-II and p62/SQSTM1 are similar at baseline, blocking the lysosomal fusion with bafilomycin showed a significant increase in both markers in $CTNS^{-/-}$ and $CTNS^{Patient}$ cells compared to $CTNS^{WT}$ cells (Fig 1D–H), indicating increased

---

**Figure 1.  CRISPR-generated $CTNS^{-/-}$ ciPTEC demonstrate cystinosis phenotype.**

A       Quantification of cystine levels (nmol/mg protein) in control ($CTNS^{WT}$) (*n* = 4), CRISPR-generated cystinotic cells ($CTNS^{-/-}$) (*n* = 6), and patient-derived cystinotic cells ($CTNS^{Patient}$) (*n* = 4).

B       Quantification of transcription factor EB (TFEB)-GFP nuclear translocation in $CTNS^{WT}$, $CTNS^{-/-}$, and $CTNS^{Patient}$, respectively. Data are demonstrated as the ratio between number of the cells with nucleus-TFEB positive over the total number of TFEB-transfected cells. The ratios were then presented as a fold change compared to control cells (*n* = 3).

C       *TFEB* mRNA expression in $CTNS^{-/-}$ and $CTNS^{Patient}$ cells compared to control cells (*n* = 3).

D, E    Representative confocal micrographs and quantification of LC3-II accumulation in $CTNS^{WT}$ and $CTNS^{-/-}$ cells in presence and absence of 25 nM bafilomycin (BafA1) for 4 h, respectively (*n* = 3). Scale bars are 20 μm.

F–H     Western blotting and densitometric analyses for LC3-II/LC3-I ratio and SQSTM1 protein levels in $CTNS^{WT}$, $CTNS^{-/-}$ and $CTNS^{Patient}$ cells cultured in the presence or in the absence of 25 nM BafA1 for 4 h, respectively (*n* = 3).

I, J    Representative confocal micrographs and quantification of DQ-BSA and BSA in $CTNS^{WT}$, $CTNS^{-/-}$ and $CTNS^{Patient}$ cells, respectively (*n* = 7–14 quantified images). Scale bars are 20 μm.

Data information: Data are expressed as mean ± SEM. *P*-values < 0.05 were considered to be significant. One-way ANOVA with Dunnett's correction (A, B, C, E, G, H, I and J) or Unpaired *t*-test (E, G and H). Exact *P*-values and statistical tests are listed in Appendix Table S1.
Source data are available online for this figure.

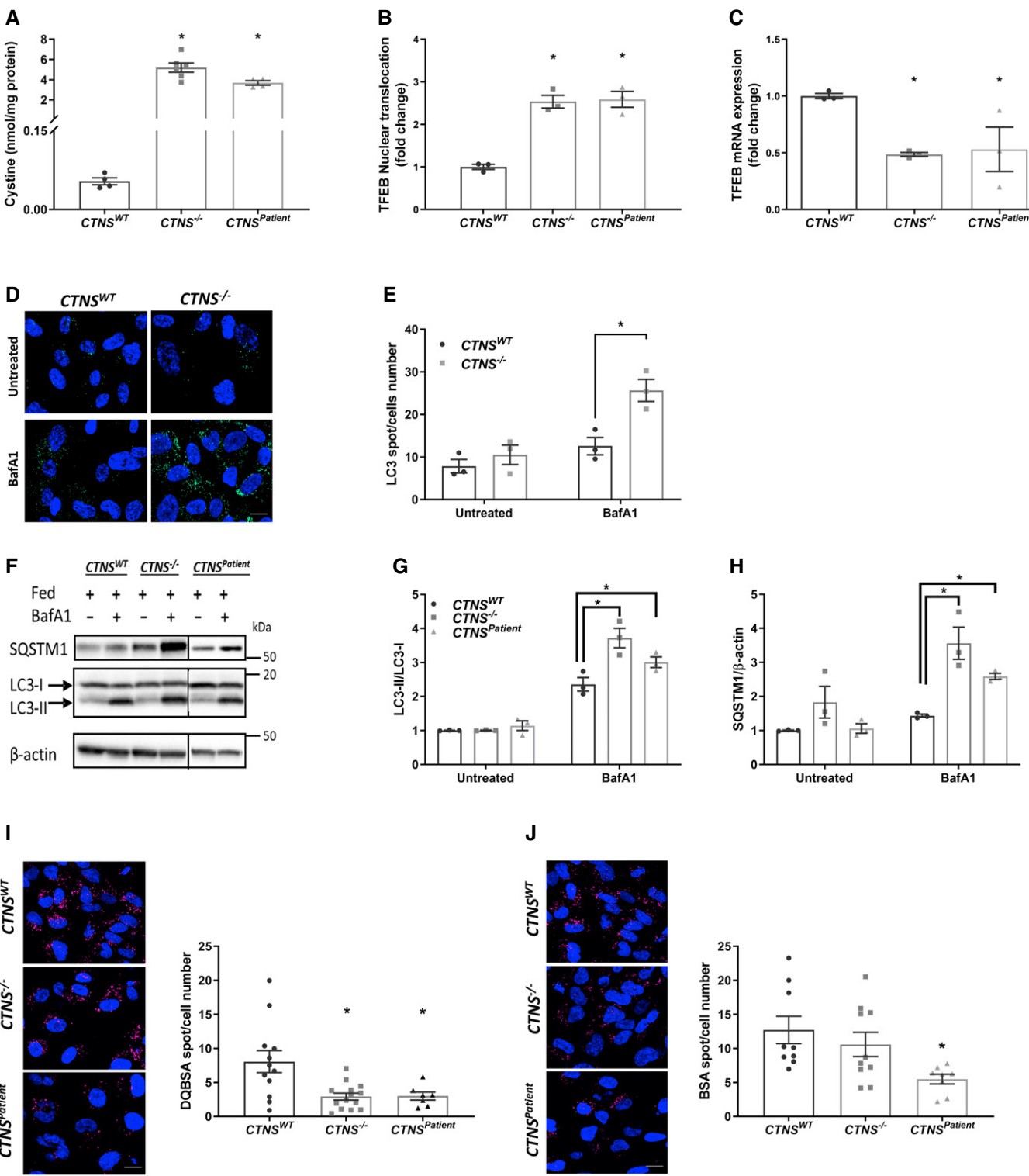

**Figure 1.**

autophagic flux (Yoshii & Mizushima, 2017). Next, we evaluated the ability of these cells to process BODIPY dye-conjugated bovine serum albumin (DQ BSA), a dye that is endocytosed and becomes fluorescent after degradation inside the lysosomes. A delayed lysosomal cargo degradation (~2.5-fold) of $CTNS^{-/-}$ and $CTNS^{Patient}$ cells

compared to control cells was observed (Fig 1I). $CTNS^{Patient}$ cells, but not $CTNS^{-/-}$ cells, demonstrated a decreased BSA uptake compared to control cells (Fig 1J), indicating the defect is related to degradation rather than protein uptake in $CTNS^{-/-}$ cells. Overall, our data confirm that the CRISPR-generated cystinotic ciPTEC are a

good isogenic model of healthy controls and they show a robust cystinotic phenotype of increased cystine accumulation, autophagy activation and reduced lysosomal cargo degradation.

## Metabolomic and proteomic profiling reveal alpha-ketoglutarate accumulation in cystinosis

To gain in-depth knowledge of the affected cellular pathways and proximal tubule impairments in cystinosis, we performed targeted metabolomic profiling of 100 key metabolites (Dataset EV1) and untargeted proteomics in the three cell lines (Fig 2). Principal component analysis (PCA) of the metabolites and over 4,500 identified proteins (Fig 2A and F) showed that $CTNS^{Patient}$ cells account for most of the variability in the data, indicating that the different genetic background of the $CTNS^{Patient}$ cells affect the data more than the $CTNS$ loss itself. This was further visualized by unsupervised hierarchical clustering in which the isogenic $CTNS^{-/-}$ cells clustered with $CTNS^{WT}$ rather than $CTNS^{Patient}$ cells (Fig 2B, and Appendix Fig S1). To explore which pathways are directly linked to $CTNS$ loss, we focused on the metabolites and proteins that were significantly altered in $CTNS^{-/-}$ cells compared to the $CTNS^{WT}$. Pathway enrichment analysis of the metabolites distinctively affected in $CTNS^{-/-}$ cells revealed several affected pathways, including cystine/cysteine metabolism, the TCA cycle, glycerolipid metabolism and amino acid metabolism (Fig 2C). Proteomic profiling, on the other hand, revealed a total of 337 proteins that were up- or downregulated in $CTNS^{-/-}$ cells compared to control cells (Fig 2G). The differentially abundant proteins were then subjected to gene ontology (GO) classification via the Panther Classification System database (Thomas et al, 2003) to highlight their molecular role in the cell (Fig 2H). The analysis showed an overall reduction in proteins involved in lipid metabolism and the breakdown of micromolecular cellular structures (peptidases, amino acid catabolism, reductases and hydrolases), and an increase in the glycolysis, TCA cycle, DNA replication and DNA repair, in line with the metabolomic data. Consistent with the reduced cargo degradation in cystinotic cells, we found an overall reduction in lysosomal catalytic proteins expression including, lysosomal acid lipase (LIPA), lysosomal acid phosphatase (ACP2), and the potentially representative

cysteine protease, cathepsin S (CTSS) (Fig 2I). Moreover, cation-independent mannose 6-phosphate receptor (IGF2R) was found downregulated in $CTNS^{-/-}$ cells (Fig 2I). IGF2R is responsible for the delivery of many newly synthesized lysosomal enzymes from Golgi to the lysosome, and its downregulation could result in decreased lysosomal activity and production of ROS (Probst et al, 2006; Takeda et al, 2019).

When cross-checking the affected metabolites of both $CTNS^{-/-}$ and $CTNS^{Patient}$ cells, we identified a set of key metabolites that were significantly changed as a result of $CTNS$ loss: cystine, cysteine, αKG, serine, hypoxanthine and oleoyl-L-carnitine levels were increased, whereas the levels of betaine, L-carnitine and glucosamine-6-phosphate were decreased in both $CTNS^{-/-}$ and $CTNS^{Patient}$ compared to control cells (Fig 2D and Appendix Fig S2). αKG was of particular interest as αKG is known to play a central role in the TCA cycle, maintaining the redox balance, regulating mTOR and autophagy. Moreover, αKG is a well-known antioxidant and its cellular levels can be increased in response to oxidative stress, contributing to the prevention and/or treatment of several disorders induced by such stress (Mailloux et al, 2009; Satpute et al, 2010; Starkov, 2013). This is done through downregulation of the mitochondrial enzyme alpha-ketoglutarate dehydrogenase (AKGDH), reported to be severely diminished in human pathologies where oxidative stress is thought to play a vital role (Starkov, 2013). Indeed, we found a decreased expression of AKGDH in $CTNS^{-/-}$ cells (Fig 2I). We also observed a significantly increased level of αKG in plasma of cystinotic patients compared to healthy controls (Fig 2E, and Table 1), signifying that loss of $CTNS$ also leads to an increase in aKG levels in patients. This prompted us to evaluate further the effect of increased levels of αKG in cystinosis pathology.

## Alpha-ketoglutarate regulates the phenotypic alterations in cystinosis

The link between cystinosin loss-of-function and increased oxidative stress has been long-established (Rizzo et al, 1999; Chol et al, 2004; Levtchenko et al, 2005; Wilmer et al, 2011). In line with this, both $CTNS^{-/-}$ and $CTNS^{Patient}$ cells presented increased ROS levels (~1.5-fold) compared to control cells (Fed condition; Fig 3A). Starvation

---

**Figure 2. Metabolomic and proteomic profiling reveal αKG accumulation in cystinosis.**

A  Principal component analysis (PCA) of control ($CTNS^{WT}$), CRISPR-generated cystinotic ($CTNS^{-/-}$) and patient-derived cystinotic ($CTNS^{Patient}$) cells based on the metabolites measured. Each dot represents one sample, and the dots of the same colour are biological replicates ($n = 3$).

B  Heat map analysis of top 25 metabolites distinctively expressed in control and $CTNS^{-/-}$ cells. Metabolites significantly decreased ($P < 0.05$) were displayed in green, while metabolites significantly increased ($P < 0.05$) were displayed in red ($n = 3$).

C  Global test pathway enrichment analysis of the intracellular metabolic interactions distinctively affected in $CTNS^{-/-}$ cells compared to healthy control cells ($n = 3$). Larger circles further from the y-axis and orange-red colour show higher impact of pathway.

D  List of metabolites that were shared and significantly altered in both the $CTNS^{-/-}$ and $CTNS^{Patient}$ cells compared to control cells ($n = 6$).

E  Plasma levels of αKG (μM) in healthy individuals ($n = 4$) and cystinotic patients ($n = 6$). All cystinotic patients were on cysteamine treatment at the time of blood sampling. For this experiment, non-parametric Mann–Whitney t-test was used to demonstrate the significance.

F  Principal component analysis (PCA) of the measured proteins in $CTNS^{WT}$, $CTNS^{-/-}$ and $CTNS^{Patient}$ ($n = 3$).

G  Volcano plot illustrates differentially abundant proteins ($n = 3$). The -$\log_{10}$ (Benjamini–Hochberg corrected P-value) is plotted against the $\log_2$ (fold change: $CTNS^{WT}$/$CTNS^{-/-}$).

H  Gene ontology (GO) analysis illustrates classes of proteins differing between $CTNS^{WT}$ and $CTNS^{-/-}$ cells ($n = 3$).

I  List of proteins that were significantly upregulated and downregulated in $CTNS^{-/-}$ cells compared to control cells ($n = 3$). AKGDH; Alpha-ketoglutarate dehydrogenase, LIPA; Lysosomal acid lipase, ACP2; Lysosomal acid phosphatase, CTSS; Cathepsin S, CTSC; Dipeptidyl peptidase, IGF2R; Cation-independent mannose-6-phosphate receptor, SORT1; Sortilin, CAT; Catalase, CTSA; Lysosomal protective protein and SOD; Superoxide dismutase.

Data information: Data are expressed as mean ± SEM. *P-values < 0.05 were considered to be significant. One-way ANOVA with Dunnett's correction (A, B, C, D, E, F, G, H and I). Exact P-values and statistical tests are listed in Appendix Table S1.

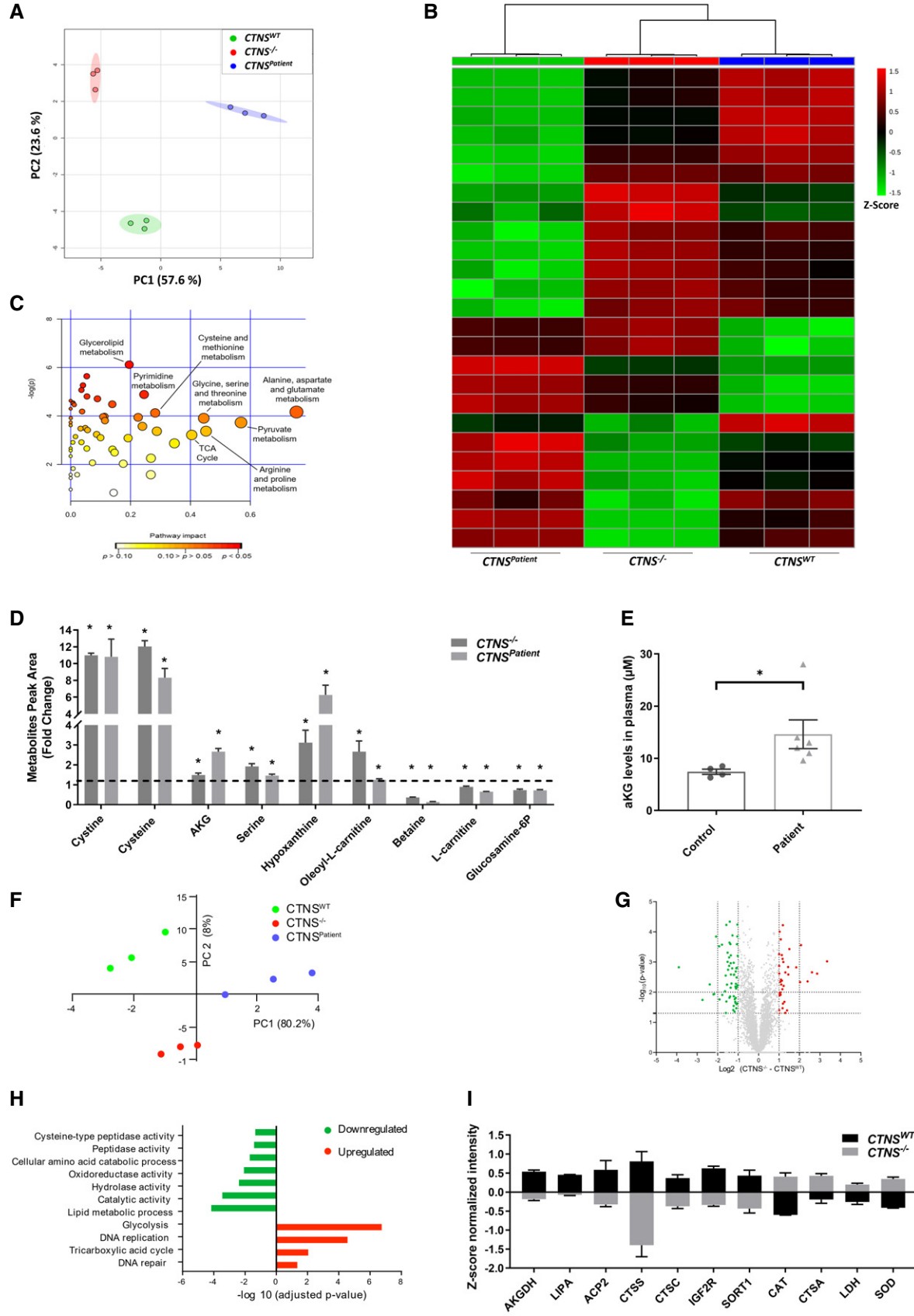

Figure 2.

Table 1. Plasma alpha-ketoglutarate levels in control and cystinotic patients.

| Individuals | Age (Month) | Sex (m/f) | Mutations | Phenotype | αKG level (μM) |
|---|---|---|---|---|---|
| Control | 37 | M | Wild type | — | 8.5 |
| Control | 29 | F | Wild type | — | 8.0 |
| Control | 38 | M | Wild type | — | 6.3 |
| Control | 32 | M | Wild type | — | 6.9 |
| Cystinotic | 34 | F | Hom 57kb del | INF | 28 |
| Cystinotic | 14 | F | 57kb del + c.del 198-218 (exon 5) | JUV | 14 |
| Cystinotic | 17 | M | Hom 57kb del | INF | 9.6 |
| Cystinotic | 16 | F | Hom 57kb del | INF | 13 |
| Cystinotic | 42 | F | 57kb del + 922insG | INF | 11 |
| Cystinotic | 6 | M | 57kb del + intronic IVS10-7G>A (intron 10) | INF | 12 |

m, male; f, female; JUV, Juvenile; INF, Infantile.

triggered ROS production further in all three cell lines (Fig 3B–D). Supplementing starved cells with dimethyl αKG (DMKG), a cell permeable form of αKG, for 4 h led to a reduction in ROS levels in $CTNS^{WT}$ cells (2.1-fold), a modest reduction in $CTNS^{-/-}$ cells (1.2-fold), but increased ROS levels in $CTNS^{Patient}$ cells (Fig 3B–D), showing limited anti-oxidative effect of αKG in CTNS-deficient cells. Of note, exposure to DMKG for 4 h was not toxic in any of the three cell lines (Fig EV3A). However, adding DMKG during the 24 h starvation had little to no effect in control cells, but led to massive cell death and apoptosis activation in cystinotic cells (Figs 3E–G, and EV3B and C). We, therefore, hypothesized that the increased level of αKG could be linked to autophagy and promoting cell death in cystinotic proximal tubule cells (Duran et al, 2012; Chin et al, 2014; Villar et al, 2017). DMKG exposure indeed resulted in an increased level of LC3-II/LC3-I ratio in $CTNS^{-/-}$ and $CTNS^{Patient}$ but not in control cells, confirming that DMKG induces autophagy in cystinotic cells (Fig 3H and I). In light of our data, we identified αKG as metabolite bridging cystinosin loss to increased oxidative stress, activation of autophagy and proximal tubule dysfunction in cystinosis.

## Bicalutamide and cysteamine combination treatment phenotypically corrects $CTNS^{-/-}$ proximal tubule cells

Cysteamine is used to efficiently reduce cystine accumulation in patients with cystinosis, but it cannot correct the established proximal tubulopathy associated with the disease. We hypothesized that this may be linked to the inability of this drug to lower αKG and target other metabolic pathways associated with CTNS loss. Cysteamine, as expected, boosted GSH and lowered cystine and cysteine levels. However, it had no significant impact on the other metabolites or on the proteome profiles (Figs 4A–E and EV4A). We, therefore, screened different candidate drugs based on their ability to reduce cystine and αKG levels and to restore the metabolic profile using metabolomics (Dataset EV1). The concentrations of the drugs tested were within a non-cytotoxic range (Fig EV3D–J). Among the candidate drugs tested, bicalutamide, an anti-androgenic agent, did not restore the high cystine and cysteine levels, however improved the overall metabolic phenotype, including αKG, serine, betaine and oleoyl-L-carnitine (Fig 4A). Treatment with bicalutamide alone also resulted in the upregulation of several metabolic enzymes involved in cysteine conversion, lysosomal degradation and the TCA cycle, including AKGDH (Figs 4C–E and EV4B). Furthermore, the cysteamine–bicalutamide combination treatment showed an additive effect and unsupervised clustering shows that the cells receiving the combination therapy are more similar to the $CTNS^{WT}$ cells than the other conditions (Fig 4B). The combination treatment also led to the upregulation of more than 50 proteins, revealing an enrichment of proteins involved in the TCA cycle and in the metabolism of macromolecules such as lipids and vitamins that were shown to be

**Figure 3. αKG is a key metabolite responsible for impaired autophagy and proximal tubule dysfunction in cystinotic proximal tubule cells.**

A  Relative reactive oxygen species (ROS/mg protein) production in control ($CTNS^{WT}$), CRISPR-generated cystinotic cells ($CTNS^{-/-}$) and patient-derived cystinotic ($CTNS^{Patient}$) cells under fed condition (n = 3).

B–D  ROS production (ROS/mg protein) in $CTNS^{WT}$, $CTNS^{-/-}$ and $CTNS^{Patient}$ cells upon starvation for 4 h in the presence and absence of DMKG (4 h), respectively (n = 3).

E  Viability test in starved (−AA) $CTNS^{WT}$, $CTNS^{-/-}$ and $CTNS^{Patient}$ cells in the presence or absence of DMKG (2 mM) for 24 h. Results are shown relative to the fed condition (n = 3).

F, G  Representative confocal micrographs (scale bars are 20 μm) and immunofluorescence analysis of caspase 3/7 activation in DMKG (2 mM)-treated $CTNS^{WT}$, $CTNS^{-/-}$ and $CTNS^{Patient}$ cells for 24 h (n = 3).

H, I  Western blotting and densitometric analyses for LC3-II/LC3-I ratio in $CTNS^{WT}$, $CTNS^{-/-}$ and $CTNS^{Patient}$ cells cultured in the presence or in the absence of BafA1 (25 nM) and DMKG (2 mM) for 4 h, respectively (n = 3).

Data information: Data are expressed as mean ± SEM. P-values < 0.05 were considered to be significant. One-way ANOVA with Dunnett's correction (A, B, C, D, E, G and I) or Unpaired t-test (B, C, D, E and G). Exact P-values and statistical tests are listed in Appendix Table S1.
Source data are available online for this figure.

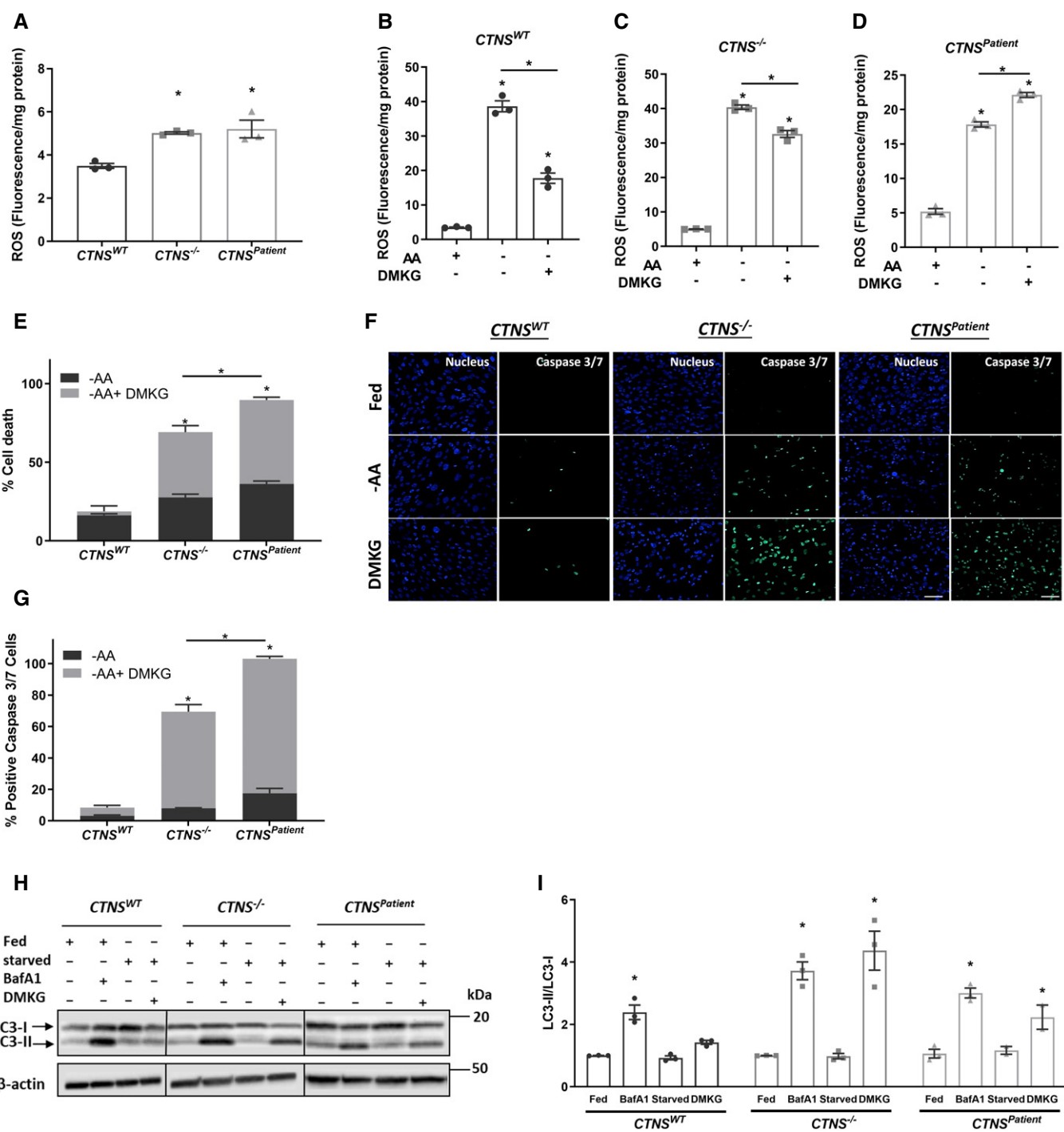

Figure 3.

downregulated in $CTNS^{-/-}$ cells (Figs 4C–E and EV4C). As bicalutamide reduced the elevated level of αKG in cystinotic cells, we hypothesized that it could also resolve αKG-mediated downstream effects. Indeed, bicalutamide (but not cysteamine) reduced the DMKG-mediated cell death and increase in LC3-II/LC3-I ratio in $CTNS^{-/-}$ cells (Fig 5A and B). Although cysteamine alone showed no effect on autophagy, combined with bicalutamide it additively reduced the DMKG-mediated increase in LC3-II/I in $CTNS^{-/-}$ cells

compared to bicalutamide alone (Fig 5B). Notably, cysteamine, bicalutamide and their combination had no effect on the basal autophagy activity in absence of bicalutamide (Fig EV4D).

We next tested whether bicalutamide alone or in combination with cysteamine could recover the antioxidant effect of αKG in cystinotic cells. Similar to cysteamine, bicalutamide was able to retrieve the effect of DMKG in $CTNS^{-/-}$ cells (2.1-fold; Fig 5C). Further reduction in ROS levels was observed when bicalutamide and

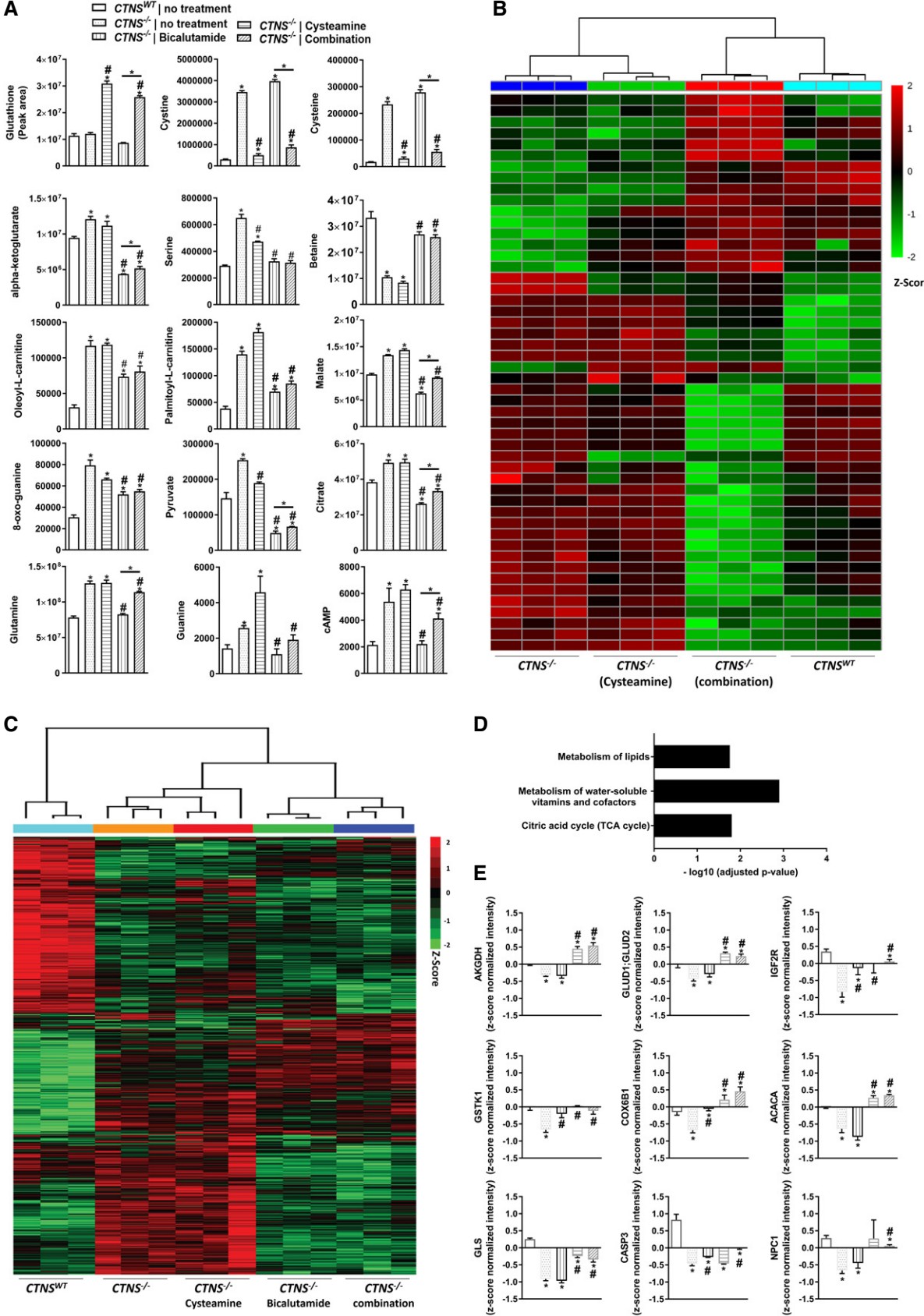

**Figure 4.**

◀

**Figure 4. Bicalutamide–cysteamine combination treatment corrects the metabolome and proteome profile of cystinotic proximal tubule cells.**

A   Metabolomic analysis of CRISPR-generated cystinotic cells ($CTNS^{-/-}$) treated with cysteamine (100 μM), bicalutamide (35 μM) and a combination of cysteamine and bicalutamide (100 and 35 μM, respectively) ($n = 3$).

B   Heat map analysis of the measured metabolites in $CTNS^{-/-}$ cells upon treatment with cysteamine or cysteamine–bicalutamide combination treatment ($n = 3$). Metabolites significantly decreased were displayed in green, while metabolites significantly increased were displayed in red.

C, D   Heat map and REACTOME analysis of the altered proteins in $CTNS^{-/-}$ cells upon treatment with cysteamine, bicalutamide and cysteamine–bicalutamide combination treatment ($n = 3$). Proteins significantly decreased were displayed in green, while metabolites significantly increased were displayed in red.

E   Proteomic analysis of $CTNS^{-/-}$ cells treated with cysteamine, bicalutamide, and a combination of cysteamine and bicalutamide, respectively ($n = 3$). AKGDH: Alpha-ketoglutarate dehydrogenase, GLUD1: GLUD2; Mitochondrial glutamate dehydrogenase 1/2, IGF2R; Cation-independent mannose-6-phosphate receptor, GSTK1: Glutathione S-transferase kappa 1, COX6B1: Cytochrome c oxidase subunit 6B1, ACACA: Acetyl-CoA carboxylase 1-Biotin carboxylase, GLS: Glutaminase kidney isoform, mitochondrial, CASP3: Caspase-3, NPC1: Niemann-Pick C1 protein.

Data information: Data are expressed as mean ± SEM. *P*-values < 0.05 were considered to be significant. One-way ANOVA with Dunnett's correction (A,B, C, D and E). * significantly different from $CTNS^{WT}$ cells ($P < 0.05$). # significantly different from $CTNS^{-/-}$ cells ($P < 0.05$). Exact *P*-values and statistical tests are listed in Appendix Table S1.

cysteamine were given together. Of note, bicalutamide alone had no effect on GSH levels (Fig 4A) but could significantly lower the ROS levels in presence of DMKG in cystinotic cells (Fig 5C), suggesting that bicalutamide, through a yet unknown mechanism, restores the antioxidant property of αKG in $CTNS^{-/-}$ cells.

Recently, bicalutamide has been patented for the treatment of several LSDs, promoting mTOR-associated autophagy and TFEB-mediated cellular exocytosis (Farrera-Sinfreu *et al*, 2014), and therefore allowing the relief and/or treatment of the symptoms of many LSDs. Indeed, bicalutamide was able to induce TFEB nucleus translocation in both control and $CTNS^{-/-}$ cells and downregulate *TFEB* mRNA expression in control cells (and Fig EV4E and F). In agreement with the increased TFEB activity, bicalutamide also restored endocytic cargo processing in cystinotic cells (Fig 5D and E), allowing us to test whether the increased degradation of lysosomal cargo results in decreased cystine accumulation. We used a more sensitive and quantitative LC-MS/MS method (Jamalpoor *et al*, 2018) to measure cystine levels in cystinotic cells upon treatment with cysteamine, bicalutamide, or their combination. In line with metabolomics data, $CTNS^{-/-}$ cells treated with bicalutamide alone did not exhibit any reduction in cystine levels. Cysteamine alone resulted in a reduction of cystine in $CTNS^{-/-}$ cells (0.37 ± 0.08 nmol/mg protein; Fig 5F), but this was still higher (7.5-fold) than that found in control cells (0.05 ± 0.02 nmol/mg protein). Interestingly, combination treatment of bicalutamide and

cysteamine resulted in a 2.8-fold decrease in cystine when compared to cysteamine alone (0.13 ± 0.003 versus 0.37 ± 0.08 nmol/mg protein; Fig 5F), bringing lysosomal cystine close to control. This effect was also found in $CTNS^{Patient}$ cells (Fig 5G).

## Bicalutamide and cysteamine combination treatment efficiently lowers cystine and αKG levels in patient-derived cystinotic kidney tubuloids

Next, we evaluated the safety and efficacy of cysteamine–bicalutamide combination treatment in patient-derived tubuloids. Tubuloids are an advanced *in vitro* model that was successfully used before to model genetic, infectious and malignant kidney disease and to screen for therapeutic efficacy (Schutgens *et al*, 2019). We established kidney tubuloids from urine samples of two paediatric cystinotic patients ($CTNS^{Patient-1}$ and $CTNS^{Patient-2}$) and compared their characteristics to tubuloids derived from healthy kidney tissue of two donors ($CTNS^{WT-1}$ and $CTNS^{WT-2}$). Urine-derived tubuloids were positive for paired-box gene 8 (PAX8) and negative for tumour protein p63 (Fig 6A), confirming that these structures indeed consisted of kidney epithelium (PAX8+/TP63−) and not urothelium (PAX8−/TP63+) (Saito *et al*, 2006; Albadine *et al*, 2010). Moreover, both patient and control tubuloids robustly expressed markers of the proximal tubule, loop of Henle, distal tubule and collecting duct (Fig EV5A) (Schutgens *et al*, 2019). Both $CTNS^{Patient}$ tubuloids



**Figure 5. Cysteamine–bicalutamide combination treatment efficiently lowers lysosomal cystine, abolishes αKG-mediated autophagy distortion and cell death in cystinotic proximal tubule cells.**

A   Cell viability test of DMKG (2 mM)-treated CRISPR-generated cystinotic cells ($CTNS^{-/-}$) upon pre-treatment with cysteamine (100 μM), bicalutamide (35 μM), and a combination of cysteamine and bicalutamide (100 and 35 μM, respectively). Results are shown relative to the starved condition ($n = 3$).

B   Western blotting and densitometric analyses for LC3-II/LC3-I ratio in DMKG (2 mM)-treated $CTNS^{WT}$ and $CTNS^{-/-}$ cells upon pre-treatment with cysteamine, bicalutamide, and a combination of cysteamine and bicalutamide ($n = 3$).

C   Relative reactive oxygen species (ROS/mg protein) production in DMKG-treated $CTNS^{WT}$ and $CTNS^{-/-}$ cells upon pre-treatment with cysteamine, bicalutamide, and a combination of cysteamine and bicalutamide ($n = 3$).

D, E   Representative confocal micrographs and quantification of DQ-BSA in $CTNS^{WT}$, and $CTNS^{-/-}$ cells upon treatment with bicalutamide. Scale bars are 20 μm ($n = 8$–14 quantified images).

F   Quantification of cystine levels (nmol/mg protein) by HPLC-MS/MS in $CTNS^{-/-}$ in the absence of the drug (NT) ($n = 4$) or upon treatment with cysteamine ($n = 6$), bicalutamide ($n = 6$), and a combination of cysteamine and bicalutamide ($n = 6$).

G   Quantification of cystine levels (nmol/mg protein) by HPLC-MS/MS in $CTNS^{Patient}$ in the absence of the drug (NT) ($n = 4$) or upon treatment with cysteamine ($n = 4$), bicalutamide ($n = 4$), and a combination of cysteamine and bicalutamide ($n = 6$).

Data information: Data are expressed as mean ± SEM. *P*-values < 0.05 were considered to be significant. One-way ANOVA with Dunnett's correction (A, B, C, D, E, F and G) or unpaired *t*-test (B, C, E, F and G). Exact *P*-values and statistical tests are listed in Appendix Table S1.
Source data are available online for this figure.

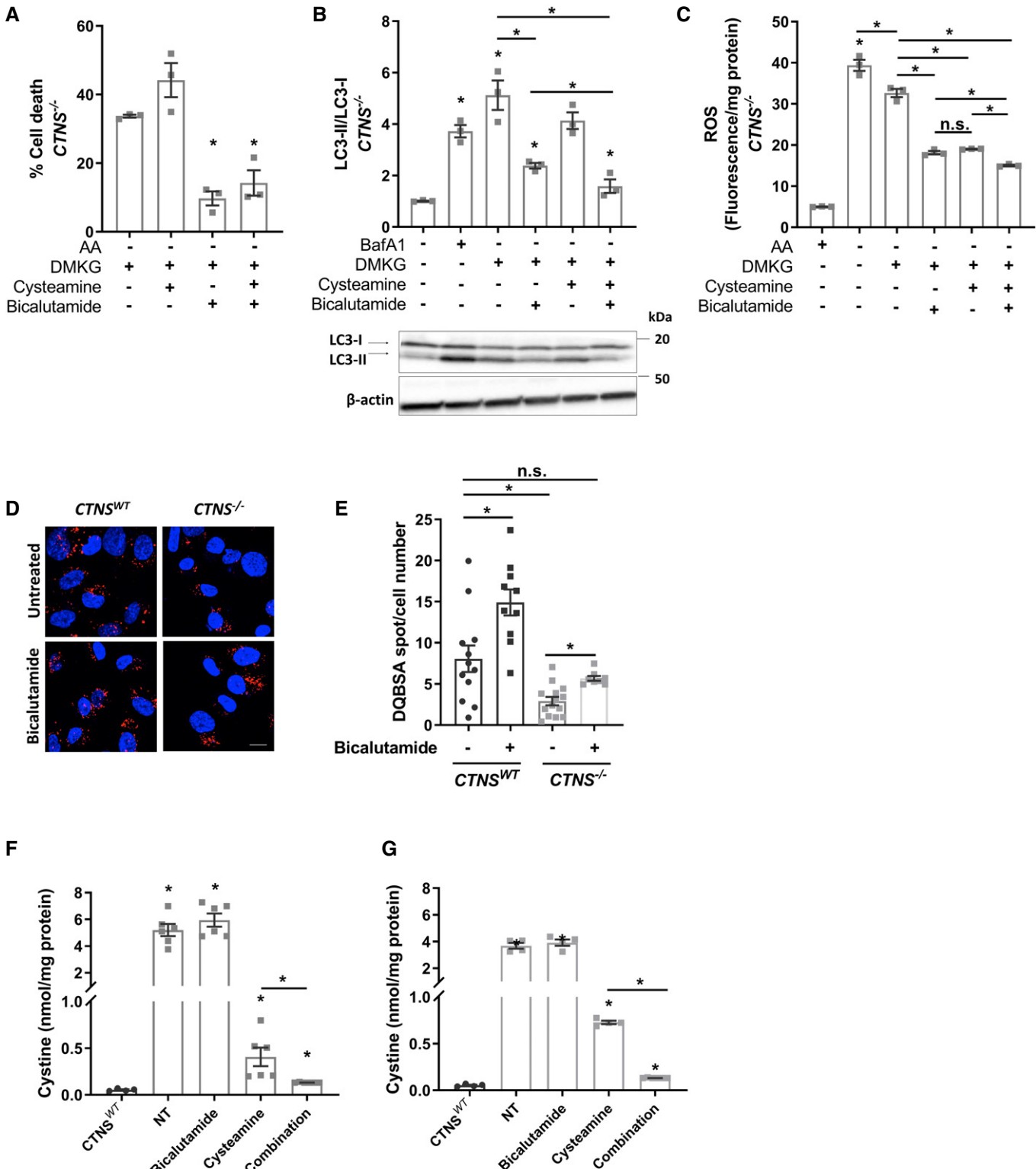

**Figure 5.**

displayed the cystinotic phenotype with increased cystine accumulation (~7-fold) compared to the controls (Fig 6B). In line with the previous findings in ciPTEC, treatment of $CTNS^{Patient}$ tubuloids with a non-nephrotoxic concentration of both cysteamine and bicalutamide (Fig EV5B–D) resulted in a more potent reduction in cystine levels (~2-fold) than with cysteamine treatment alone (Fig 6C and D). Furthermore, we performed targeted metabolomic profiling (Dataset EV2) and identified that bicalutamide (but not cysteamine), either alone or in combination with cysteamine, reduced the αKG levels in $CTNS^{Patient-2}$ tubuloids (Fig 6E).

### Bicalutamide and cysteamine combination treatment efficiently lowers cystine levels and improves survival of cystinotic zebrafish

Finally, we assessed the combination therapy in cystinotic zebrafish, a well-characterized *in vivo* model of cystinosis (Elmonem *et al*, 2017; Elmonem *et al*, 2018). Cystinotic zebrafish displayed significantly increased levels of cystine, dysmorphic features, delayed hatching and reduced survival compared to controls (Figs 6F–K and EV5E–G). As expected, cysteamine was able to reduce cystine levels (1.5-fold; Fig 6H) and a trend towards an additive effect of bicalutamide was observed (1.8-fold reduction for the combination), though this was not significant. However, bicalutamide either alone or in combination with cysteamine significantly improved the survival and reduced the percentage of dysmorphism in the cystinotic zebrafish without having an effect on their hatching rate, when compared to cysteamine monotherapy (Fig 6I–K). This indicates that the combination treatment does not induce toxicity and is beneficial in cystinotic zebrafish compared to cysteamine treatment alone.

## Discussion

In this work, we identified a new therapeutic target in nephropathic cystinosis by evaluating the persistent cellular link between cystinosin loss-of-function and proximal tubule cell dysfunction. Using an omics-based strategy, we identified several proteins and

metabolites to be consistently altered across the proximal tubule cell models used. Increased levels of αKG were particularly of interest as this metabolite is involved in autophagy regulation, oxidative stress and apoptosis, which are known to be dysregulated in cystinosis. Our results also revealed that a bicalutamide–cysteamine combination treatment could provide a novel pharmacological approach for the phenotypical correction of cystinosis.

Consistent with previous reports, we found that *CTNS*-deficient cells show high levels of cystine accumulation, activation of autophagy, delayed protein degradation and an increase in baseline ROS levels (Rizzo *et al*, 1999; Chol *et al*, 2004; Levtchenko *et al*, 2005; Settembre *et al*, 2008; Wilmer *et al*, 2011; Platt *et al*, 2012; Raggi *et al*, 2014; Ivanova *et al*, 2015; Festa *et al*, 2018). This was confirmed further by metabolic and proteomic analyses, where we found a reduction in lysosomal catabolic proteins and an upregulation of enzymatic antioxidizing agents such as catalase and superoxide dismutase in cystinotic cells, which was consistent with delayed lysosomal cargo degradation and increased oxidative stress, respectively. In addition, we found that aKG levels are increased in *CTNS*-deficient cells and in cystinotic patients, a metabolite known for its antioxidant properties and potential for the treatment of disorders induced by oxidative stress (Mailloux *et al*, 2009; Satpute *et al*, 2010; Starkov, 2013; Liu *et al*, 2018). During starvation, both control and *CTNS*-deficient cells showed a strong increase in ROS (after 4 h), and the *CTNS*-deficient cells seem to handle this well with only a slight reduction in cell viability compared to control cells after 24 h. Addition of DMKG (to increase intracellular aKG levels) during starvation led to a strong reduction in ROS in $CTNS^{WT}$ cells but had a minor effect in $CTNS^{-/-}$ cells and even an opposite effect in $CTNS^{Patient}$ cells. Even though DMKG had a minor effect on ROS levels in $CTNS^{-/-}$ cells during short-term exposure, it still led to massive cell death after 24 h in *CTNS*-deficient cell lines.

Many studies have investigated the role of αKG in response to ROS, mTOR and autophagy regulation, but the outcomes vary greatly depending on the context and the cell type studied. Overall, αKG is regarded as a health-promoting metabolite, reducing liver fibrosis (Zhao *et al*, 2016), delaying age-related disease and increasing life span in *in vivo* models such as *Caenorhabditis elegans* and mice (Chin *et al*, 2014; Su *et al*, 2019) and as a potential anti-cancer

▶

**Figure 6. Cysteamine–bicalutamide combination treatment is effective in patient-derived cystinotic tubuloids and cystinotic zebrafish.**

A  Immunocytochemistry of patient-derived cystinotic tubuloids ($CNTS^{Patient-1}$ and $CNTS^{Patient-2}$) and healthy kidney tissue-derived control tubuloids ($CNTS^{WT-1}$ and $CNTS^{WT-2}$) for PAX8, TP63 and F-actin. Scale bar = 100 μm.

B  Quantification of cystine levels (nmol/mg protein) by HPLC-MS/MS in control and cystinotic tubuloids ($n = 3$).

C, D  Quantification of cystine levels (nmol/mg protein) by HPLC-MS/MS in two different patient-derived cystinotic tubuloids in the absence of the drugs (NT) or upon treatment with cysteamine (100 μM), bicalutamide (35 μM) or cysteamine (100 μM)-bicalutamide (35 μM) combination treatment ($n = 3$).

E  αKG levels measured in patient-derived cystinotic tubuloids ($CNTS^{Patient-1}$ and $CNTS^{Patient-2}$) in the absence of the drugs (NT) or upon treatment with cysteamine, bicalutamide or cysteamine–bicalutamide combination treatment using metabolomics ($n = 3$).

F  Representative images of control and cystinotic zebrafish.

G  Quantification of cystine levels (nmol/mg protein) by HPLC-MS/MS in control and cystinotic zebrafish ($n = 3$).

H  Quantification of cystine levels (nmol/mg protein) by HPLC-MS/MS in cystinotic zebrafish after treatment with cysteamine (1,000 μM), bicalutamide (10 μM), and a combination of cysteamine and bicalutamide (1,000 and 10 μM, respectively) ($n = 40$ embryos per group).

I  Survival rates in $ctns^{-/-}$ zebrafish upon treatment with cysteamine, bicalutamide, and a combination of cysteamine and bicalutamide ($n = 40$ embryos per group).

J  Deformity rates in $ctns^{-/-}$ zebrafish after treatment with cysteamine, bicalutamide, and a combination of cysteamine and bicalutamide ($n = 40$ embryos per group).

K  Hatching rates in surviving $ctns^{-/-}$ zebrafish evaluated at 72- and 96-h post-fertilization (hpf) with cysteamine, bicalutamide, and a combination of cysteamine and bicalutamide ($n = 40$ embryos per group).

Data information: Data are expressed as mean ± SEM. *P-values < 0.05 were considered to be significant. One-way ANOVA with Dunnett's correction (B, C, D, E, G, H and J) or unpaired *t*-test (C, D and H). Exact P-values and statistical tests are listed in Appendix Table S1.

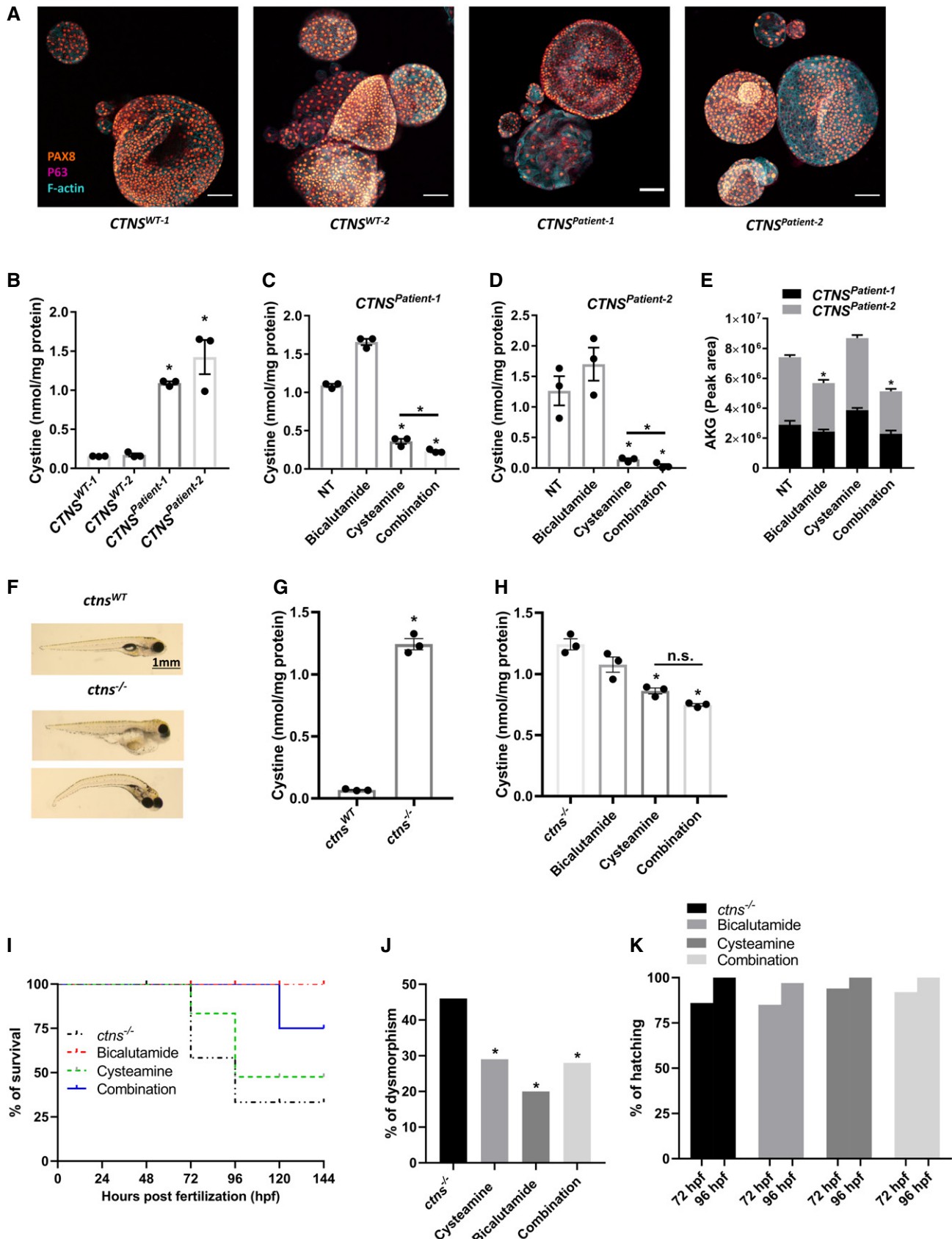

Figure 6.

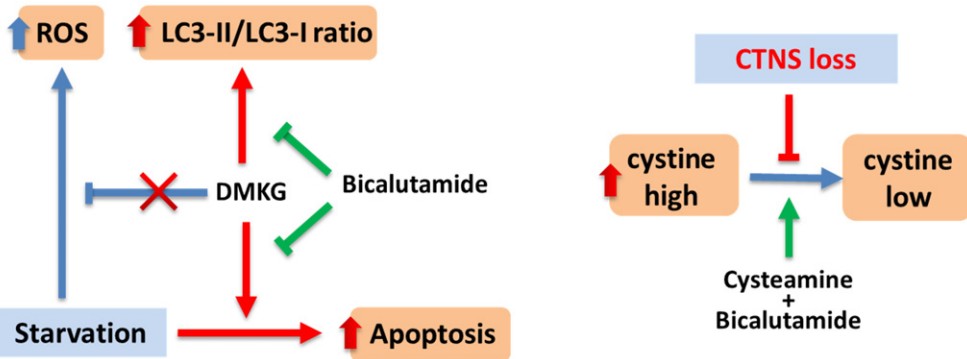

**Figure 7. Working model summarizing the results obtained in this work.**

The blue arrows indicate wild-type situation, red arrows/text indicate the changes in cystinotic cells, and green arrows indicate intervention with medication. DMKG: dimethyl α-ketoglutarate, ROS: Reactive oxygen species.

agent (Rzeski et al, 2012; Villar et al, 2017). On the other hand, Villar et al (2017) showed that addition of DMKG in cancer cells for 24 h inhibited autophagy and induced glutaminolysis-mediated apoptosis during starvation.

In our cells, addition of bicalutamide but not cysteamine could prevent the DMKG induced cell death and reduced LC3-II/LC3-I levels. As cysteamine alone is able to efficiently induce glutathione synthesis in cystinotic cells, it is likely that the DMKG decreased cell viability was not due to a reduced availability of antioxidants. The marked increase in LC3-II/LC3-I levels in *CTNS*-deficient cells indicates activation of autophagy. However, the reduced protein degradation (shown here by DQ-BSA analysis and in previously published reports) (Ivanova et al, 2015; Festa et al, 2018) question the efficiency of this autophagic process. The high intracellular levels of αKG together with a defective autophagy may therefore induce glutaminolysis-mediated apoptosis during starvation specifically in cystinotic cells. As bicalutamide is known to upregulate the autophagic flux (Hao et al, 2017), the addition of bicalutamide could restore the autophagic flux in cystinotic cells and prevent this glutaminolysis-mediated apoptosis.

Several reports also indicate a role for mitochondria in cystinosis disease pathology, including affected mitochondrial function and dynamics (De Rasmo et al, 2019), as well as reduced autophagy-mediated clearance of damaged mitochondria resulting in increased ROS in their cells (Sansanwal et al, 2010; Festa et al, 2018). This is also in line with the decreased expression of AKGDH in our $CTNS^{-/-}$ cells, which can then lead to the observed increase in αKG. However, other TCA metabolites that we evaluated (cis-aconitate, succinic acid, fumarate, malate and citrate) remained unchanged. As there are multiple pathways in the cell that can either generate or catabolize aKG, it is also possible that another pathway is responsible for the increase in aKG. Studying the role of mitochondria in the progression of cystinosis remains important to further expand our knowledge on the complexity of the disease and prioritize drug targets for cystinosis.

Recent studies show that overexpression of TFEB activity stimulates lysosomal excretion and rescues the delayed endocytic cargo processing, allowing the relief and/or treatment of the symptoms of many LSDs (Platt et al, 2012). This in line with the study of Rega

et al (2016), where stimulation of endogenous TFEB activity by genistein was shown to lower cystine levels and rescue the delayed endocytic cargo processing in cystinotic proximal tubule cells. Furthermore, inhibition of mTOR signalling by everolimus was shown to activate autophagy, rescue the number of large lysosomes, and in combination with cysteamine, reverse the cystine/cysteine loading defect in patient-specific and CRISPR-edited cystinotic induced pluripotent stem cells and kidney organoids (Hollywood et al, 2020).

Bicalutamide has recently been patented for the treatment of several LSDs, promoting mTOR-associated autophagy and cellular exocytosis (Farrera-Sinfreu et al, 2014). Here we find that in $CTNS^{-/-}$ cells, bicalutamide was able to increase endogenous TFEB activity, restore endocytic cargo processing and, in combination with cysteamine, normalize lysosomal cystine levels in both patient-derived and CRISPR-edited cystinotic proximal tubule cells. Furthermore, cysteamine–bicalutamide combination treatment, but not cysteamine alone, reversed the proteomic and metabolic phenotype and resulted in reduction of αKG levels, and upregulation of AKGDH and IGF2R proteins, resolving the αKG-mediated downstream effects in cystinotic proximal tubular cells (Fig 7). So far, several murine models for cystinosis have been developed that recapitulate some of the clinical features seen in patients with cystinosis, including high tissue cystine levels (Johnson et al, 2013; Gaide Chevronnay et al, 2014; Gaide Chevronnay et al, 2015; Napolitano et al, 2015). However, the renal phenotype is often mild, has a late onset or lacks signs of a proximal tubulopathy or renal failure (Cherqui et al, 2002). Furthermore, unlike the human situation where treatment with cysteamine does not improve the renal phenotype and only improves some parts of the cystinotic phenotype (Cherqui, 2012), it can completely resolve any renal symptoms in mice. This means that the unmet medical need that is currently present in humans is not observed in Ctns$^{-/-}$ mice, making this model unsuitable for studying any drug combination therapy aimed at improving the renal phenotype. Hence, to extend our knowledge on the beneficial effect of the combination treatment and bring the therapy one step closer to clinical application, we evaluated the safety and efficacy in patient-derived tubuloids, an advanced *in vitro* model, and *in vivo* in cystinotic zebrafish. Tubuloids offer several advantages over

conventional mammalian cell lines. (i) They are human-derived three dimensional (3D) renal tubule epithelial cultures, offering physiological heterogeneity and recapitulating the *in vivo* situation. (iiii) Tubuloids are grown in 3D, allowing neighbouring cells to interact in a more physiological way than in conventional 2D culture models. (iii) Tubuloids are not genetically modified or reprogrammed and can accurately mimic the donor genotype and phenotype, thereby allowing personalized medicine (Schutgens *et al*, 2019). Zebrafish models may generate symptoms that are not directly linked to clinical features in patients (Santoriello & Zon, 2012; Vliegenthart *et al*, 2014), but are valuable both for screening drug toxicity and as genetic disease models (MacRae & Peterson, 2015). The *ctns*$^{-/-}$ zebrafish presents a robust and versatile model of cystinosis with early phenotypic characteristics of the disease that can be used for the *in vivo* screening of novel therapeutic agents (Elmonem *et al*, 2017; Elmonem *et al*, 2018). The fact that the improved cystine lowering-efficacy of the combination treatment could be reproduced in tubuloids from two different patients and in cystinotic zebrafish, underlines the robustness of these findings and increases the likelihood that this treatment can be successfully extrapolated to cystinosis patients.

Bicalutamide, a non-steroidal anti-androgenic agent, is also one of the most widely prescribed drugs for treating prostate cancer (Osguthorpe & Hagler, 2011). The use of any androgen deprivation therapy is generally associated with the risk of acute kidney injury (AKI) (Lapi *et al*, 2013; Peng *et al*, 2019), and up to 36% of patients who have taken bicalutamide for 1 to 6 months may experience AKI (Peng *et al*, 2019). However, this association is mainly driven by a combination of gonadotropin- or luteinizing-releasing hormone agonists with bicalutamide (Lapi *et al*, 2013). Bicalutamide–cysteamine combination treatment at an effective concentration was not only found to be safe in our models after short-term exposure, but also improved the survival and reduced the percentage of dysmorphism in the cystinotic zebrafish, confirming the safety profile of bicalutamide both in *in vitro* and *in vivo*. Anti-androgenic therapy is also known to reduce testosterone levels (Braga-Basaria *et al*, 2006; Lapi *et al*, 2013), complicating the beneficial effect of bicalutamide in cystinotic male patients with delayed maturation (Fivush *et al*, 1988; Chik *et al*, 1993; Winkler *et al*, 1993). Interestingly, bicalutamide is a racemate and its anti-androgenic activity is attributed solely to the (R)-enantiomer (Tucker & Chesterson, 1988; Mukherjee *et al*, 1996), while the (S)-enantiomer with little, if any, anti-androgenic activity mediates autophagy effects (Farrera-Sinfreu *et al*, 2014). Therefore, it would be of great scientific benefit to investigate further the effect of the (S)-bicalutamide and/or develop any structural analogues to treat cystinosis.

Taken together, we identified a new therapeutic target by evaluating the persistent cellular abnormalities using an omics-based strategy. We identified αKG as an important metabolite bridging cystinosin loss to lysosomal autophagy defect, apoptosis activation and proximal tubule cell impairment in cystinosis. Bicalutamide, but not cysteamine, was able to reduce αKG levels and resolve αKG-mediated downstream effects in cystinotic cells. Bicalutamide in combination with cysteamine demonstrated additively reduced cystine levels both *in vitro* and *in vivo*, suggesting this combination therapy holds great potential to treat patients with cystinosis. Preclinical studies in a suitable animal model should determine whether cysteamine–bicalutamide combination therapy is able to improve or prevent the development of renal Fanconi syndrome and renal failure. Translation to the clinic should be further facilitated by the fact that bicalutamide is already an approved drug with a known safety profile, although a separate study should be performed to determine the appropriate dose for cystinotic patients.

# Materials and Methods

### Reagent and antibodies

All chemicals and reagents were obtained from Sigma-Aldrich (Zwijndrecht, The Netherlands) unless specified otherwise. Primary antibodies used were mouse anti-LAMP1 (Santa Cruz Biotechnology Cat#sc-17768, RRID:AB_626851, dilution 1:200), rabbit anti-mTOR (Cell Signaling Technology Cat#2983, RRID:AB_2105622, dilution 1:400), rabbit anti-LC3 (Novus Cat#NB600-1384, RRID:AB_669581, dilution 1:1,000), mouse anti-SQSTM1 (p62) (BD Transduction Laboratories Cat#610832, dilution 1:1,000), rabbit anti-β-actin (Cell Signaling Technology Cat#4970, RRID:AB_2223172, dilution 1:4000), Rabbit anti-PAX8 (Proteintech Cat#10336-1-AP, RRID:AB_ 2236705, dilution 1:200), mouse anti-p63 (Abcam Cat#ab124762, dilution 1:200) and Alexa Fluor® 488 Phalloidin (Cell Signaling Technology Cat#8878, dilution 1:100). Polyclonal goat anti-rabbit (#P0448, dilution 1:5,000) and polyclonal goat anti-mouse (#P0447, dilution 1:5,000) secondary antibodies were obtained from Dako products (CA, USA). Alexa-488 goat anti-mouse (dilution 1:500), Alexa-647 goat anti-rabbit (dilution 1:200), donkey anti-rabbit-AF647 (dilution 1:300), and donkey anti-mouse-AF568 (dilution 1:200) secondary antibodies were from Life Technologies Europe BV (The Netherlands).

### Generation of *CTNS*$^{-/-}$ isogenic cell line of ciPTEC using CRISPR

Guide RNAs (gRNAs) targeting exon 4 of the *CTNS* gene were designed using the online gRNA designing tool available at chop-chop.cbu.uib.no. In order to maximize specificity, guide sequences with high scores for on-target efficiency and no predicted off-targets having at least 3 base pair mismatches in the genome were selected. Optimal gRNA (5′-GTCGTAAAGCTGGAGAACGG-3′) was cloned into the pSPCas9(BB)-2A-GFP plasmid (Addgene #48138) as described previously by Ran et al (2013) and introduced into healthy ciPTEC using PolyPlus JetPrime. 72 h post-transfection, GFP-positive singlet cells were sorted using FACS Aria-II flow cytometer and expanded in 96-wells plate. The gRNA cut site was amplified with PCR using the primers flanking the cut region (F.CTNS_ex4 5′-GGCCTGTTTTCCTCCATCTCTG-3′; R.CTNS_ex4 5′-AAGTGCCAACCAGCAGCTC-3′). Knockouts were confirmed by Sanger sequencing followed by intracellular cystine accumulation.

### CiPTEC culture

The ciPTEC used in this study (ciPTEC *CTNS*$^{WT}$ and ciPTEC *CTNS*$^{patient}$) were obtained from Cell4Pharma (Nijmegen, The Netherlands, MTA #A16-0147). These cell lines have been generated previously from proximal tubule cells isolated from urine samples of a cystinotic patient and an age-matched healthy control, followed by immortalization (Wilmer *et al*, 2005; Wilmer *et al*, 2010; Wilmer

*et al*, 2011). In brief, primary cells were immortalized by transfection with SV40T ts A58 (SV40T) and hTERT (human telomerase reverse transcriptase) followed by subcloning to obtain homozygous cell populations. Of each donor, one subclone was selected based on its proximal tubular characteristics including morphology, expression pattern and transport activity. In this study, we used the existing ciPTEC *CTNS*$^{WT}$ cell line to generate an isogenic *CTNS* knockout line, named *CTNS*$^{-/-}$. Mycoplasma contamination was checked every 2 months and was found to be negative in all cell lines used.

Cells were cultured as previously described by Wilmer *et al*, 2010 (Wilmer *et al*, 2010). The culture medium was Dulbecco's modified Eagle medium DMEM/F-12 (GIBCO) supplemented with foetal calf serum 10% (v/v), insulin 5 μg/ml, transferrin 5 μg/ml, selenium 5 μg/ml, hydrocortisone 35 ng/ml, epidermal growth factor 10 ng/ml and tri-iodothyronine 40 pg/ml. In short, cells were seeded at a density of 55,000 cells/cm$^2$ and grown at 33°C for 24 h to enable them to proliferate and subsequently cultured at 37°C for 7 days to mature into fully differentiated epithelial cells.

### Tubuloid culture

Tubuloids were established from the urine of two paediatric cystinosis patients and healthy kidney tissue from two donors as previously described (Schutgens *et al*, 2019). Urine was collected, cooled to 4°C, rinsed with phosphate-buffered saline (PBS) and with advanced Dulbecco's modified Eagle's medium (ADMEM)-F12 (Gibco) supplemented with 1% HEPES, 1% glutamax, 1% penicillin/streptomycin, 0.1 mg/ml primocin (Invivogen) and 10 μM Y-27632 (Abmole). The pellets were resuspended in basement membrane extract (BME, SanBio/Trevigen) and plated. After BME droplets had solidified, ADMEM-F12 supplemented with 1% HEPES, 1% glutamax, 1% penicillin/streptomycin, 0.1 mg/ml primocin, 1,6% B27 (Gibco), 1% Rpsondin-3 conditioned medium (U-Protein Express), 50 ng/ml EGF (PeproTech), 100 ng/ml FGF10 (PeproTech), 1 mM N-acetylcysteine, 5 μM A83-01 (Tocris Bioscience) and 10 μM Y-27632 was added as expansion medium. In order to establish tubuloids from kidney tissue, the tissue was digested with 1 mg/ml collagenase for 45 min. Tissue fragments were resuspended in BME and plated. After BME droplets had solidified, expansion medium was added. For specific experiments, tubuloids were differentiated using ADMEM-F12 with 1% HEPES, 1% glutamax and 1% penicillin/streptomycin (differentiation medium).

### Cell treatment

Standard starvation medium was Hank's balanced salt solution (HBSS; GIBCO). When indicated, the cell permeable form of αKG, dimethyl α-ketoglutarate (DMKG) was added to a final concentration of 2 mM for 4 or 24 h. Cysteamine (100 μM), bicalutamide (35 μM), or the combination of cysteamine and bicalutamide (100 and 35 μM, respectively) were used as a treatment for 96 h (5 days).

### Zebrafish maintenance and breeding

The animal care and experimental procedures were carried out in accordance with the ethical committee guidelines for laboratory animal experimentation at KU Leuven. Zebrafish (Danio rerio) were AB strain wild-type and *ctns*$^{-/-}$ mutant. Cystinotic and wild-type larvae were raised at 28.5°C in egg water (Instant Ocean Sea Salts, 60 μg/ml). At 48-h post-fertilization, larvae were treated with cysteamine 1,000 μM, bicalutamide 10 μM or their combination. Drugs were administered at 48-h post-fertilization in all experiments dissolved in the swimming water. The medium was refreshed every day, and dead embryos were sorted out.

### Intracellular cystine quantification by HPLC-MS/MS

Cystine levels were quantified using HPLC-MS/MS; a rapid and sensitive assay that has been developed and validated in house (Jamalpoor *et al*, 2018). In brief, ciPTEC, tubuloids and zebrafish larvae pellets were suspended in N-Ethylmaleimide (NEM) solution containing 5 mM NEM in 0.1 mM sodium phosphate buffer (pH 7.4). The cell suspension was precipitated, and protein was extracted with sulfosalicylic acid 15% (w/v) and centrifuged at 20,000 *g* for 10 min at 4°C. Protein concentration was determined by the method of the Pierce™ BCA protein assay kit according to the manufacturer's protocol (Thermo Fisher, The Netherlands), and the cystine concentration was measured using HPLC-MS/MS. Data are expressed as the cystine values (nmol) corrected for total protein content (mg).

### Quantitative real-time PCR

The mRNAs were extracted from cells using the Qiagen RNeasy mini kit according to the manufacturer's instructions. Total mRNA (600 ng) was reverse transcribed using iScript Reverse Transcriptase Supermix (Bio-Rad). Quantitative real-time PCR was performed using iQ Universal SYBR Green Supermix (Bio-Rad) with the specific sense and anti-sense primers for *TFEB* (forward: 5′-GCAGTCCTAC CTGGAGAATC-3′; reverse: 5′-GTGGGCAGCAAACTTGTTCC-3′). The ribosomal protein S13 (*RPS-13*) (forward: 5′-GCTCTCCTTTCGTT GCCTGA-3′; reverse: 5′- ACTTCAACCAAGTGGGGACG-3′) was used as the reference gene for normalization and relative expression level were calculated as fold change using the $2^{-\Delta\Delta Ct}$ method.

Tubuloids were grown in expansion medium for a few days and then differentiated for 7 days. Tubuloids were lysed and RNA was isolated using the RNEasy Mini Kit (Qiagen) according to the manufacturer's protocol. Quantitative real-time qPCR was performed using the iQ SYBR Green Supermix (Bio-Rad). A 384-well plate was used with a reaction volume of 12.5 μl and duplicates for each reaction. For the read-out, the CFX384 Touch Real-Time PCR Detection System (Bio-Rad) was used. Expression levels of the following genes were measured: *RPS-13* (housekeeping gene), *ANPEP*, *ABCC3*, *HNF1A*, *HNF4A*, *SLC12A1*, *SLC12A3*, *CALB1*, *AQP2* and *AQP3*. Primer sequences are provided in Table 2. Expression was normalized to the expression of *RPS-13* within the same sample (ΔCt).

### Immunofluorescence and confocal microscopy

To investigate mTORC1/LAMP1 colocalization, control and cystinotic ciPTEC lines were cultured on coverslips with the respective treatments. Thereafter, cells were fixed with 4% paraformaldehyde in PBS for 10 min, permeabilized with 0.1% Triton-X solution for 10 min and blocked with 1% bovine serum albumin (BSA) diluted

**Table 2. Sequences of the primers used for tubuloids study.**

| Gene name | Forward primer 5′→3′ | Reverse primer 5′→3′ |
|---|---|---|
| ANPEP | TGAGCTGTTTGACGCCATCT | GCCCTGCTTGAATACGTCCT |
| ABCC3 | CACCAACTCAGTCAAACGTGC | GCAAGACCATGAAAGCGACTC |
| HNF1A | CCAGTAAGGTCCACGGTGTG | TTGGTGGAGGGGTGTAGACA |
| HNF4A | CACGGGCAAACACTACGGT | TTGACCTTCGAGTGCTGATCC |
| SLC12A1 | AACTTTGGGCCACGCTTCAC | CCACACAGGCCCCTACACAA |
| SLC12A3 | CTCCACCAATGGCAAGGTCAA | GGATGTCGTTAATGGGGTCCA |
| CALB1 | TGATCAGGACGGCAATGGAT | AGCTTCCCTCCATCCGACAA |
| AQP2 | CTCCATGAGATCACGCCAGC | TCATCGGTGGAGGCGAAGAT |
| AQP3 | CTGGATCAAGCTGCCCATCT | CATTGGGGCCCGAAACAAAA |

in PBS for 30 min. Subsequently, cells were stained with the primary antibodies diluted in blocking buffer overnight at 4°C. After three washes with PBS, the cells were incubated for 2 h at room temperature with the corresponding secondary antibodies. Nuclei were stained with Hoechst 33342 (1 μM), and cells were imaged with a DeltaVision confocal microscope (Cell Microscopy Core, Department of Cell Biology, University Medical Centre Utrecht).

To assess TFEB intracellular distribution, cells were seeded in special optic 96-well plate until to reach 50% confluence. Cells were then transfected with the TFEB-GFP plasmid (a kind gift from Dr. Annelies Michiels (Viral Vector Core, Leuven, Belgium)) using Poly-Plus JetPrime reagent according to the manufacturer's instructions. After 48 h from transient transfection, cells were stained with Hoechst 33342 (1 μM) for 10 min and imaged using a Cell Voyager 7000 (CV7000) confocal microscope (Yokogawa Electric corporation, Tokyo, Japan). TFEB nuclear translocation data are expressed as number of cells with nucleus-TFEB positive over the total number of TFEB-transfected cells.

For tubuloid experiments, tubuloids were differentiated for 7 days. Tubuloids were fixed for 45 min with 4% formaldehyde and then permeabilized and blocked with 0.5% Triton X-100 plus 0.5% BSA in PBS for 30 min. Subsequently, tubuloids were rinsed with 0.5% BSA plus 0.1% Tween-20 in PBS and then stained with the primary antibodies diluted in 0.5% BSA plus 0.1% Tween-20 in PBS overnight at 4°C. Tubuloids were rinsed twice with 0.5% BSA plus 0.1% Tween-20 in PBS and then stained for 2 h at room temperature with the corresponding secondary antibodies and DAPI (1:1,000) in 0.5% BSA plus 0.1% Tween-20 in PBS. Tubuloids were washed twice with 0.5% BSA + 0.1% Tween-20 in PBS, mounted and imaged using a Leica SP8 confocal microscope.

## Immunoblots

The ciPTEC were seeded in 6-well plates in triplicate with the respective treatments. Subsequently, cells were washed twice with PBS and lysed with RIPA buffer containing protease inhibitor cocktail (Roche). Protein quantification was performed using the method of the Pierce™ BCA protein assay kit according to the manufacturer's protocol. After the electrophoresis, the proteins were transferred to a nitrocellulose membrane (midi kit, Bio-Rad) with Trans-Blot Turbo Transfer System (Bio-Rad). Finally, membranes

were imaged using the ChemiDoc™ XRS+ (Bio-Rad) and analysed using ImageJ software.

## Endocytosis assay

The endocytic uptake was monitored in ciPTEC following incubation for 1.5 h at 37°C with 50 μg/ml of either BSA-AlexaFluor-647 (A34785, Thermo Fisher Scientific) or DQ Red BSA (D12051, Invitrogen). The cells were then fixed and stained with Hoechst 33342 (1 μM) for 10 min and imaged using a CV7000 confocal microscope (Yokogawa Electric corporation, Tokyo, Japan). Data were quantified with Columbus™ Image Data Storage and analysis software (PerkinElmer, Groningen, the Netherlands). Data are expressed as the number of BSA/DQ-BSA spots per cell.

## Metabolomics

CiPTEC cells were washed with ice cold PBS, and metabolites were extracted in 1 ml lysis buffer containing methanol/acetonitrile/dH$_2$O (2:2:1). Samples were centrifuged at 16,000 g for 15 min at 4°C, and supernatants were collected for LC-MS analysis.

LC-MS analysis was performed on an Exactive mass spectrometer (Thermo Scientific) coupled to a Dionex Ultimate 3000 autosampler and pump (Thermo Scientific). The MS operated in polarity-switching mode with spray voltages of 4.5 kV and −3.5 kV. Metabolites were separated using a Sequant ZIC-pHILIC column (2.1 × 150 mm, 5 μm, guard column 2.1 × 20 mm, 5 μm; Merck) with elution buffers acetonitrile (A) and eluent B (20 mM (NH$_4$)$_2$CO$_3$, 0.1% NH$_4$OH in ULC/MS grade water (Biosolve)). Gradient ran from 20% eluent B to 60% eluent B in 20 min, followed by a wash step at 80% and equilibration at 20%, with a flow rate of 150 μl/min. Analysis was performed using LCquan software (Thermo Scientific). Metabolites were identified and quantified on the basis of exact mass within 5 ppm and further validated by concordance with retention times of standards. Peak intensities were normalized based on total peak intensities, and data were analysed using MetaboAnalyst (Chong et al, 2019).

## Proteomics

CiPTEC pellets were lysed in boiling guanidinium lysis buffer containing 6 M guanidinium HCl (GuHCl), 5 mM tris(2-carboxyethyl)phosphine (TCEP), 10 mM chloroacetamide, 100 mM Tris–HCl pH 8.5, supplemented with protease inhibitor (cOmplete mini EDTA-free, Roche). Pellets were boiled for 10 min at 99°C, sonicated for 12 rounds of 5 s (Bioruptor Plus, Diagenode), and spun down at 20,000 × g for 15 min. Protein concentration was determined using Pierce™ BCA protein assay kit. Equal amounts of protein per condition were digested with Lys-C (1:100, Wako) for 4 h at 37°C, diluted to a final concentration of 2 M GuHCl, followed by trypsin digestion (1:100, Sigma-Aldrich) overnight at 37°C. Tryptic peptides were acidified to a final concentration of 1% formic acid (FA) (Merck), cleaned up using Sep-Pak cartridges (Waters) and dried in vacuo.

Peptide samples were analysed with an UHPLC 1290 system (Agilent technologies) coupled to an Orbitrap Q Exactive HF × mass spectrometer (Thermo Scientific). Peptides were trapped (Dr Maisch Reprosil C18, 3 μm, 2 cm × 100 μm) and then separated on an

analytical column (Agilent Poroshell EC-C18, 2.7 μm, 50 cm × 75 μm). Trapping was performed for 5 min in solvent A (0.1% FA) and eluted with following gradient: 4–8% solvent B (0.1% FA in 80% acetonitrile) in 4 min, 8–24% in 158 min, 24–35% in 35 min, 35–60% in 17 min, 60–100% in 4 min and finally 100% for 1 min. Flow was passively split to 300 nl/min. The mass spectrometer was operated in data-dependent mode. At a resolution of 35,000 $m/z$ at 400 $m/z$, MS full scan spectra were acquired from $m/z$ 375–1,600 after accumulation to a target value of $3e^6$. Up to 15 most intense precursor ions were selected for fragmentation. HCD fragmentation was performed at normalized collision energy of 27% after the accumulation to a target value of $1e^5$. MS/MS was acquired at a resolution of 30,000. Dynamic exclusion was enabled with an exclusion duration of 32 s. RAW data files were processed with MaxQuant (v1.6.0.16 (Cox, Mann 2008)), and MS2 spectra were searched with the Andromeda search engine against the SwissProt protein database of Homo Sapiens (20,259 entries, downloaded 31/01/2018) spiked with common contaminants. Cysteine carbamidomethylation was set as a fixed modification, and methionine oxidation and protein N-term acetylation were set as variable modifications. Trypsin was specified as enzyme and up to two miss cleavages were allowed. Filtering was done at 1% false discovery rate (FDR) at the protein and peptide level. Label-free quantification (LFQ) was performed, and "match between runs" was enabled. The data were further processed using Perseus 1.6.0.7 (Tyanova *et al*, 2016).

### Alpha-ketoglutarate measurement by LC-MS/MS

Alpha-ketoglutarate (2-KG; Fluka 75893) calibration standards of, respectively, 0.8, 1.6, 4.0, 8.0 and 20 μM, and an internal standard solution of 2.2 μM 3,3,4,4-D4 2 kg (kind gift Prof. Dr. H. Blom VU Amsterdam) were prepared in dH$_2$O. 50 μl plasma sample or calibration standard together with 50 μl internal standard solution and 50 μl dH$_2$O were pipetted onto a Microcon ultrafilter (30 kDa, Millipore). The resulting ultrafiltrate (15 min 14,000 $g$ 15°C) was acidified with 20 μl 4% formic acid (Merck) in dH$_2$O. 3 μl of the acidified ultrafiltrate was injected onto a peek-lined InertSustain AQ-c18 (2.1*100 mm dp 3μ) column using a I-Class Acquity (Waters). The column was run at 40°C in gradient mode using 0.5% acetic acid in H$_2$O and Acetonitrile (initial conditions 100% 0.5% acetic acid in H$_2$O at 250 μl/min). The column flow was directed to a Xevo TQSμ (Waters) fitted with an electrospray ionization probe operating in the negative mode at unit resolution. The capillary voltage was set at 0.6 kV. The temperature settings for the source and ionblock were, respectively, 550 and 150°C. As a drying gas, nitrogen was used at a flow rate of 800 l/h. The conegas flow was set at 50 l/h. The collision cell was operated with argon as the collision gas at a pressure of 0.35 Pa. An area response was generated by recording the MRM transitions of the neutral loss CO$_2$ for both 2 kg an d4–2 kg. The d4–2 kg was used to normalize the area response. Quantification was performed using a linear regression curve constructed from the normalized area response from the prepared calibration standards.

### Cell viability assays

To estimate the percentage of cell death, cells were seeded in a 96-well plate and after respective treatments, and the percentage cell death was assessed using Presto Blue Cell Viability Reagent according to manufacturer's instructions.

For tubuloid experiments, the drug safety screening was performed using a protocol derived from Driehuis et al (2019). In short, BME was dissolved by incubation in 1 mg/ml dispase II for 30 min at 37°C. Tubuloids were digested into tiny fragments using Accutase. These fragments were plated in BME, grown for 1 day in expansion medium and then switched to differentiation medium for 2 days. Next, BME was dissolved using dispase II as described above, after which tubuloids were sieved using a 70 μm cell strainer (BD Falcon) and counted. 500 tubuloids in 40 μl differentiation medium with 5% BME were plated in a 384-well plate using the Multidrop dispenser (Thermo Fisher) and treated with increasing concentrations of bicalutamide that were always combined with 100 μM cysteamine. DMSO volume was 1% in all conditions. Drugs were added using the D300E digital dispenser (Tecan). After 5 days, tubuloids were lysed with 40 μl Cell-Titer Glo (Promega). Luminescence was measured using the Spark® multimode microplate reader (Tecan) to determine ATP levels as an indicator of the amount of tubuloids that had survived the treatment. Tubuloid viability was normalized to the viability (ATP levels) upon treatment with cysteamine alone (= 100%).

### Apoptosis assay

The ciPTEC were plated in an optic 96-well imaging plates and treated with DMKG in presence or absence of bicalutamide, cysteamine or their combination. Subsequently, the Cell Event Caspase-3/7 Green Detection Reagent (8 μM) was added and incubated for 30 min before imaging. Caspase-active cells were identified as described in manufacturer's instruction. Each well was imaged using the CV7000 confocal microscope and analysed with Columbus™ Image Data Storage and analysis software (PerkinElmer, Groningen, The Netherlands).

### ROS detection assay

The ciPTEC were seeded in 96-well plates and ROS levels were assessed using general oxidative stress indicator (CM-H2DCFDA; Invitrogen) according to the manufacturer's protocol. Briefly, the cells were treated with the CM-H2DCFDA reagent (10 μM) and incubated in the dark. Following 20 min incubation at 37°C, cells were rinsed once with HBSS and incubated with the different treatment conditions as previously stated. Fluorescence was measured using a fluorescent microplate reader (Fluoroskan Ascent, Thermo Fisher Scientific, Vantaa, Finland) at excitation wavelength of 492 nm and emission wavelength of 518 nm. Data are expressed as the fluorescence values normalized to the protein concentration.

### Statistical analysis

Statistical analysis was performed using GraphPad Prism 7.0 (GraphPad Software, Inc., USA). Data are presented as mean ± standard error of the mean (SEM) of minimally three independent experiments performed in triplicate. Significance was evaluated using one-way analysis of variance (ANOVA), or where appropriate, unpaired two-tailed Student's *t*-test was applied. *P*-values < 0.05 were considered to be significant.

**The paper explained**

**Problem**

Young children diagnosed with cystinosis receive cysteamine as a treatment to prevent the accumulation of cystine. This drug effectively lowers cystine values in their blood and tissues, but it is not able to prevent kidney function loss. In this study, we evaluated the effect of *CTNS* deficiency, the gene affected in cystinosis, in kidney proximal tubule cells, organoids and cystinotic zebrafish and determined the ability of different drug compounds to restore cell function.

**Results**

Loss of *CTNS* in proximal tubule cells reduces the ability to effectively degrade proteins, increases reactive oxygen species production and leads to large scale chances in intracellular metabolites and protein expression levels. Alpha-ketoglutarate (αKG) was one of the metabolites elevated in cystinotic cells as well as in patient plasma and may be linked to the increased autophagy and apoptosis in cystinotic cells. When evaluating the effect of different drug compounds on the cells, we found that a combination therapy of cysteamine with bicalutamide was far more effective in restoring cell functions than cysteamine alone and this finding was confirmed in different cystinosis models, including the kidney cell lines, kidney organoids and cystinotic zebrafish.

**Impact**

Cysteamine treatment alone cannot restore all cellular defects resulting from *CTNS* loss, and a combination therapy with a compound targeting the autophagic phenotype could therefore be highly beneficial in managing cystinosis. Our findings indicate that a cysteamine–bicalutamide combination therapy is able to improve proximal tubule cell function *in vitro* and could potentially fulfil the unmet clinical need of preventing kidney failure in cystinotic patients.

The zebrafish sample size was estimated based on the established method published before (Elmonem *et al*, 2017). In order to minimize the effects of subjective bias, the aquatic facility took care of the mating procedures independently of the subsequent analysis and for each strain, n.6 females and n.4 males were mated at the same time. For each group, zebrafish larvae were selected randomly from different pools of larvae.

For the human studies, cystinosis patients and control subjects were randomly selected and alfa-ketoglutarate measurements were performed by a blinded operator and analysed by an independent investigator. For the alfa-ketoglutarate measurements in the human subjects, an F-test was performed to compare variance between the cystinosis patients and controls. Since the variances were significantly different between the cystinosis and control subjects group, a non-parametric test was applied to test for significance between the cystinosis and control group. Informed consent was obtained from all patients and experiments conform the principles set out in the WMA declaration of Helsinki and the Department of Health and Human Services Belmont Report.

## Data availability

The data sets produced in this study are available in the following databases:

Proteomics data: PRoteomics IDEntifications database (PRIDE) partner repository with the dataset identifier PXD020046 (https://www.ebi.ac.uk/pride/archive/projects/PXD020046).

Metabolomics data: EMBL-EBI MetaboLights database with the dataset identifier MTBLS2538 (https://www.ebi.ac.uk/metabolights/MTBLS2538).

**Expanded View** for this article is available online.

## Acknowledgements

This work was financially supported by a grant from the Dutch Kidney Foundation (grant nr.150KG19) and the E-Rare 2-Joint Call 2014, Novel Therapies for Cystinosis grant from Zon-MW (grant nr. 113301402). R.M., F.A.Y.Y., C.M.E.A., M.B.R. and M.C.V. gratefully acknowledge the support of the partners of "Regenerative Medicine Crossing Borders" (RegMed XB), Powered by Health~-Holland, Top Sector Life Sciences & Health; and the Gravitation Program "Materials Driven Regeneration", funded by the Netherlands Organization for Scientific Research (024.003.013).

## Author contributions

AJ, MJJ and RM designed the study; AJ, CAGHvG, FAYY, EAZ, SPB, KRV, CPC, KV, KE, CMEA, RENvdW and MJJ performed experiments; LRR, MRL, MBR, HC, JK, EL, CRB, MCV and MA provided input on experimental design, manuscript content and data representation; AJ, MJJ and RM interpreted the data and wrote the paper with all co-authors' assistance; MJJ and RM provided supervision.

## Conflict of interest

The authors declare that they have no conflict of interest.

## For more information

i   OMIM: CYSTINOSIS, NEPHROPATHIC; *CTNS* https://www.omim.org/entry/219800
ii  ERK-NET: Cystinosis: https://www.erknet.org/index.php?id=disease_detail&disease_uid=213.0&disease_group=all&ref=patient
iii Orphanet: Cystinosis: https://www.orpha.net/consor/cgi-bin/OC_Exp.php?Lng=GB&Expert=213
iv  Cystinosis Research Foundation: https://www.cystinosisresearch.org/
v   Cystinosis Ireland: https://cystinosis.ie/
vi  Cystinose Groep Nederland: https://cystinose.nl/cystinosis/

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
