## [Review Process File · EMBO Molecular Medicine]

Cysteamine-bicalutamide combination therapy corrects proximal tubule phenotype in cystinosis

Amer Jamalpoor, Charlotte van Gelder, Fjodor A Yousef Yengej, Esther Zaal, Sante Princiero Berlingerio, Koenraad R Veys, Carla pou casellas, Koen Voskuil, Khaled essa, Carola ME Ammerlaan, Laura Rita Rega, Reini van der Welle, Marc Lilien, Maarten Rookmaaker, Hans C. Clevers, Judith Klumperman, Elena Levtchenko, Celia Berkers, Marianne Verhaar, Maarten Altelaar, Rosalinde Masereeuw, and Manoe Janssen

DOI: [10.15252/emmm.202013067](https://doi.org/10.15252/emmm.202013067)

Corresponding author: Manoe Janssen (manoe.janssen@gmail.com)

Review Timeline:

Submission Date:	6th Jul 20
Editorial Decision:	4th Aug 20
Revision Received:	4th Nov 20
Editorial Decision:	20th Nov 20
Revision Received:	21st Mar 21
Editorial Decision:	14th Apr 21
Revision Received:	20th May 21
Accepted:	21st May 21

Editor: Zeljko Durdevic

Transaction Report:

4th Aug 2020

Dear Dr. Janssen,

Thank you for the submission of your manuscript to EMBO Molecular Medicine. We have now received feedback from the three reviewers who agreed to evaluate your manuscript. As you will see from the reports below, the referees acknowledge the interest of the study but also raise serious concerns that should be addressed in a major revision. Alongside with addressing the reviewers concerns experimentally the manuscript needs careful editing of the data presentation in the text and figures in order to better convey the main conclusions of your study.

Addressing the reviewers' concerns in full will be necessary for further considering the manuscript in our journal, and acceptance of the manuscript will entail a second round of review. EMBO Molecular Medicine encourages a single round of revision only and therefore, acceptance or rejection of the manuscript will depend on the completeness of your responses included in the next, final version of the manuscript. For this reason, and to save you from any frustrations in the end, I would strongly advise against returning an incomplete revision.

We realize that the current situation is exceptional on the account of the COVID-19/SARS-CoV-2 pandemic. Therefore, please let us know if you need more than three months to revise the manuscript.

I look forward to receiving your revised manuscript.

Yours sincerely,

Zeljko Durdevic

***** Reviewer's comments *****

Referee #1 (Remarks for Author):

Summary of the study.

Response: In this research article, the authors identified α KG as a key metabolite linking autophagy disruption, increased oxidative stress, and proximal tubule dysfunction in cystinotic renal proximal tubule cells. the authors found cysteamine-bicalutamide combination treatment, but not

cysteamine alone, reversed the proteomic and metabolic phenotype and resulted in reduction of α KG levels, resolving the α KG-mediated downstream effects in cystinotic proximal tubular cells. They have used several models to establish their hypothesis - RPTECs isolated from cystinosis pts urine, CRISPR mediated CTNS-KO conditional RPTE cell line, cystinotic zebra fish, human-derived three dimensional (3D) renal tubule epithelial cultures (Tubuloids).

limitations and strengths of the study

Response:

Limitations

- Fig. 1: I do not see any significant difference in IF (1B) between fed CTNS-WT & CTNS-/. They looked very similar to each other and different from CTNS-patient. No visual change was noted in the SQSTM1 protein expression (1H, J) in CTNS patient. Not sure how they got significant difference in expression in the bar chart. Also, they looked at total SQSTM1 protein expression, and not the active form. Therefore, this doesn't represent inactivity of mTORC1 in Cystinosis. You can relate less active SQSTM1 and LC3-II with increased autophagy & inactive mTORC1, which was not shown in this paper. Their CTNS-/- cell line is more similar to WT than patient RPTECs
- Fig2: As shown in 2A-B, CTNS patient, CTNS-WT & CTNS-/- have very different & distinct protein expression. Moreover, CTNS-/- behaves similar to WT than CTNS patient. Not sure why for further experiments (Fig 2C, G, H, I) authors compared WT with CTNS-/. Rather they should have compared WT with CTNS patients. In figure 2D, they have compared CTNS patient with CTNS-/-, which is of no use. This entire figure lacks proper exp planning. Not sure if we can conclude anything substantial from this figure.
- Fig3: DMKG (a form of alphaKG) reduced ROS in CTNS-WT & CTNS-/- & increased ROS in CTNS patient. However, DMKG treatment increased apoptosis in CTNS-/- & CTNS patients. It shows that reduced ROS induces apoptosis in CTNS-/. Whereas increased ROS in CTNS patients also induce apoptosis. How do you explain this? No significant difference between LC3-II expression in WT-ctrl and patient-ctrl (3I). It is not clear why DMKG reduced LC3-II expression in WT and patient, and not in CTNS-/. The cartoon in 3J is too far stretched
- In the rest of the figures they show the effect of cysteamine-bicalutamide combination treatment on these cell lines, tubuloids, and cystinotic zebra fish.
- The authors have done many experiments, but no solid conclusion can be derived based on the way the data is presented. This requires extensive focused editing

Referee #2 (Comments on Novelty/Model System for Author):

Bar histograms with parametric statistical analyses for low numbers should no longer be accepted (see details in report)

Referee #2 (Remarks for Author):

OVERALL STATEMENT

The authors have generated new isogenic human Ctns^{-/-} kidney proximal tubular cell lines that can be more closely compared to WT cells than the previously studied immortalized patient cell line Ctnspatient. By -omics studies, they identified a host of metabolic changes, in addition to cystine accumulation (the target of the only FDA-approved drug, cysteamine) and ROS hallmarks. Among these changes, they focused on the Krebs cycle intermediate metabolite, alpha-ketoglutarate (aKG), as putative key actor of sensitivity to ROS and starvation, which are triggers of autophagy and cell death in cystinotic cells. To manipulate aKG levels, they resorted to incubation with a permeable aKG analog, dimethyl aKG (DMKG) which exacerbated the phenotype of Ctns^{-/-} cells (autophagy defects and cell death). A screen based on aKG identified bicalutamide, used to treat prostate cancer and other lysosomal storage diseases. Most interestingly, cysteamine and bicalutamide separately (and additively) corrected different sets of metabolic changes and the combination of cysteamine with bicalutamide best corrected cystinotic phenotype in human cell lines and organoids. Data on zebrafish are a less impressive contribution to this reviewer. In conclusion, this paper using state-of-the-art approaches reports a series of novel informations that stress the role in cystinosis pathophysiology of metabolic changes beyond cystine accumulation. It also highlights the potential benefit of combining cysteamine with non-cystine-directed drug therapy, as also very recently reported by another group (luteolin; de Leo et al, JASN 2020; to be quoted now). These two studies call for experimentation of cysteamine-combined treatments in closer (mouse, rat) disease models. In this goal, the selectivity of the S-enantiomer of bicalutamide for autophagy would indeed represent an attractive bonus. The present manuscript would significantly benefit by seriously taking into consideration detailed comments below.

MAJOR COMMENTS

1. Data documentation is not optimal.

1.1. In general, bar histograms are now discouraged and should be replaced by dotplots (as actually used at 1KL; 5E). It is also uncertain, if not dubious, that distributions of continuous variables are normal, thus systematic use of parametric statistical tests for low numbers is highly questionable. Indeed, Fig2E compares plasma aKG concentration in 4 healthy vs 6 patients where boxer-whisker plots (thus not mean \pm SEM as implied from legend) clearly suggest non-normal distribution in patients. Do non-parametric tests confirm significance? At Fig2D, no statistical analysis is presented: is the modest (1.5 fold) increase of the focused metabolite, aKG, in Ctns^{-/-} cells statistically significant? Incidentally, broken ordinate combined with change of increment prevents one to appreciate extent of higher increase aKG in Ctnspatient.

1.2. Numbers of samples at Fig 2E seem really too low for such a crucial piece of information (in fact the only one directly relevant to patients). Larger sample collection of patients and matched volunteers should not be a serious problem.

1.3. Quality of imaging is not optimal and morphology seems "lightly" interpreted, as if satisfied by pathway confirmation rather than taking into account structural implications. For example, Fig1B is not convincing as "representative": in WT fed cells (intended to validate the assay of co-localization between mTOR and LAMP-1), whereas yellow objects appear indeed to dominate in the cell with selected area, this is not the case in the cell just above; conversely, in the WT starved (-AA) selected cell to document full dissociation, several objects above the nucleus would appear doubly labeled; to the opposite, whereas the fed Ctnspatient cell shows clear dissociation as expected, fed Ctns^{-/-} cells show a multiplicity of yellow spots that would instead suggest unexpected preserved association of mTOR with Lamp-1. Linescans could help. Moreover, a

constant finding in cultured cystinotic cells is lysosomal enlargement with perinuclear clustering (Ivanova et al, PLOSone, 2015, quoted; Festa et al, Nat Comm, 2018, quoted, see beautiful Fig1A). This information should be immediately accessible by LAMP1 immunolabeling as shown at Fig1B, but is regrettably not commented upon. Is this feature reproduced and corrected by bicalutamide ?

1.4. Careful manuscript revision is still needed. For example, legend of Fig 1C indicates three types of cells, yet only two images are shown. Its quantitation refers to fold-change (from 1 in WT to 2.5), yet image shows at left (WT?) no TFEB nuclear signal (accordingly, baseline should be 0, not 1 ?). Larger fields should be presented.

1.5. This reviewer was not impressed by the complex cartoon presenting the working model at Fig 3J with multiple effects. It is suggested to both simplify to core effects (sequence of steps of the TCA cycle, available in any biochemistry textbook, can be omitted) and to present a sequence of situations : (i) untreated cystinosis; (ii) aKG manipulation by DMKG; (iii) treatment by cysteamine alone; (iv) treatment by bicalutamide ; (v) combined treatments.

2. Lack of mitochondrion imaging.

Considering that ROS production in cystinotic cells (e.g. Festa et al, 2018) and targeted TCA cycle both occur in mitochondria, it is regretted that mitochondrion imaging is not presented, at least for the key points of the demonstration. There are excellent easy tool for discriminating ROS accumulation in mitochondria vs lysosomes (e.g. Denamur et al, Free Radic Biol Med, 2011, Fig3) and immunolabeling can readily evidence distorted (or normalized) mitochondrial shape. Such a subcellular structural study would not be that time-consuming, while mechanistically strengthening the biochemical/cell based demonstration, thus is strongly recommended..

3. Membrane permeant metabolites are tricky and not a priori bona fide equivalents.

In the field of cystinosis itself, membrane permeant cystine derivatives got a bad reputation as being proved toxic by themselves to mitochondria (referenced by Cherqui, Nat Rev Nephrol, 2017; quoted). Fortunately, control of intrinsic lack of toxicity after 24 h at 2mM DMKG in fed cystinotic cells is reinsuring. However, considering DMKG as equivalent to aKG seems far-fetched (e.g. see page 9, para 3, line 5 : "in presence of aKG" when data show "under DMKG supplementation". Moreover, short-term effects of DMKG on ROS are significantly opposite in the two cystinosin-deficient cell lines. Thus, comparison of intracellular levels of aKG and DMKG, and the effect of DMKG supplementation on actual aKG level should be quite informative and thus also strongly recommended. This clarification is all the more important since aKG effects are discussed as context-dependent.

4. Conclusion that aKG is a key metabolite should be explicitly presented with more caution (page 8, first para, last line).

Besides aKG, several other metabolites were modified by cystinosis and corrected by bicalutamide (Fig4A). Thus, while aKG was useful reporter to eventually identify bicalutamide as useful drug, drug benefits cannot be safely attributed to aKG as the "key actor" (or implicitly presented as such).

MINOR COMMENTS

*Abstract , line 1 : monogenic

*Intro

page 3, para 1, line 3 : "leading"; stricto sensu, Fanconi syndrome in general does not per se lead to chronic kidney failure; it certainly does so in nephropathic cystinosis (NC).

Page 3, para 3, line 4 : it would be appropriate to already quote here the landmark paper by Festa et al, Nat Comm, 2018.

*Page 6, first para, last line : please clarify to which cell (Ctns-/- ?) this conclusion applies. Conversely, would the authors conclude that Ctnspatient cells are endocytosis- but not degradation-defective? Please comment.

*Fig legend 1B : scale bar of 13 microm is awkward

* Fig2I. IGF2R/Cation-independent mannose 6-phosphate receptor. It is suggested (page 7, end of first para) that this decrease would cause defective lysosomal targeting of lysosomal enzymes by rerouting to the constitutive secretory pathway, thus could contribute to explain their paradoxical decrease in steady-state cell content, despite global transcriptional activation by TFEB (see Fig 1D). However, defective lysosomal proteolysis is instead attributed to impaired activation of pro-cathepsin C (Festa et al, Nat Comm, 2018, quoted : Fig3c), whose level actually increases. Moreover, as to lysosomal acid phosphatase, it is known not to be targeted to lysosomes as a soluble cargo by a M6P anchor but rather assembled as a transmembrane protein which is eventually released into lysosomal lumen upon proteolytic release from its anchor (classical work from von Figura's lab). Please clarify and/or rewrite accordingly.

*Fig3A : for consistency, replace Ctns KO by Ctnspatient

*Fig5A : cell death in starved (-AA) 2-mM DMKG-treated Ctns-/- cells is 35% here vs 70% at Fig3E. Please clarify.

Discussion, page 14, lines 9-11: should be deleted; following sentences should also be rewritten. (i) Cysteamine can improve, but certainly does not fully correct cystinosis manifestations and lesions in mice. (ii) The authors may be informed of a powerful rat model, with more robust and earlier phenotype, generated by three different laboratories. (iii) The zebrafish model is useful for screening yet may generate symptoms irrelevant for patients (embryonic dysmorphism; cystinotic babies are born normal)

*Supp Fig 3. The rationale for including toxicity data of other drugs such as genistein, disulfiram etc... in this particular study is unclear.

Referee #3 (Remarks for Author):

This is an innovative investigative approach to characterize the cellular mechanisms responsible for the dysregulation of proximal tubular function in experimental cystinosis models. The authors used immortalized cell culture lines of cystinosis and control tissue, and a unique application of organ

tubuloids from cystinosis patients, and finally, a zebrafish model of cystinosis.

This combination of experimental approaches with advanced proteomic and metabolomic analyses, provided new and encouraging data on the specific cellular mechanisms responsible for the varying phenotypes and pathophysiology of cystinosis. Indeed, these novel approaches have totally circumvented the numerous shortcomings of the in vivo models of cystinosis where there are limited renal phenotypes.

One minor suggestion would be to possibly expand upon the findings to point to further development of drug interventions for testing in special populations to advance the field.

Referee: 1**Question 1:**

A) I do not see any significant difference in IF (1B) between fed CTNS-WT & CTNS^{-/-}. They looked very similar to each other and different from CTNS-patient.

We thank the reviewer for this comment. Figure 1B (now figure EV2) shows immunofluorescent staining of lysosomal-associated membrane protein 1 (LAMP1; green) and mTOR (red) in CTNS^{WT}, CTNS^{-/-}, and CTNS^{Patient} cells. In control cells, mTOR respond to the nutrient status of the cells. In the presence of nutrients (standard medium, fed condition) mTOR co-localizes with the lysosomal membrane (in yellow), whereas upon starvation (-AA) mTOR was released from the lysosomes (less colocalization) and upon reintroduction of nutrients the co-localization is re-established. However, mTOR in CTNS^{-/-} and CTNS^{Patient} cells was already released from lysosomes in the fed condition and a shift based on the nutritional status of the cells was not observed. This indicates decreased mTOR activity in cystinotic cells also under fed condition. Moreover, downstream effects of mTOR deactivation such as increased TFEB nuclear translocation and LC3-II accumulation were also observed, further confirming that mTOR activity is decreased in both cystinotic cell lines. This is in accordance with other studies where they have shown the decreased retention of mTOR on the lysosomal membrane, and therefore the decreased activity of mTOR in *in vitro* and *in vivo* models of cystinosis (Andrzejewska, Z. et al. J Am Soc Nephrol, 2016; Festa et al, Nat communication, 2018; Ivanova et al, PLOsone, 2015, Ivanova et al. J Inherit Metab Dis 2016). Unfortunately, technical hurdles did not allow us to quantify the co-localization between mTOR and LAMP-1 to show dissociation of the mTOR complex in cystinosis. In our cell lines, the size of lysosomes are so small and are generally clumped together hampering reliable quantification. Therefore, we moved figure 1B to the figure EV2 and explained the technical hurdles in the result section.

B) No visual change was noted in the SQSTM1 protein expression (1H, J) in CTNS patient. Not sure how they got significant difference in expression in the bar chart. Also, they looked at total SQSTM1 protein expression, and not the active form. Therefore, this doesn't represent inactivity of mTORC1 in Cystinosis. You can relate less active SQSTM1 and LC3-II with increased autophagy & inactive mTORC1, which was not shown in this paper. Their CTNS^{-/-} cell line is more similar to WT than patient RPTECs.

We thank the reviewer for this constructive suggestion. We agree that the Western blot shown was not very clear regarding SQSTM1 protein expression in CTNS^{Patient} cells (now figure 1H, I). For the bar graphs, we combined data from three independent experiments (see blot images below).

In accordance with other studies, we used total SQSTM1 protein expression to demonstrate lysosomal degradation and autophagic flux in cystinosis (Sansanwal, Pediatr Nephrol, 2012; Festa et al, Nat Communication, 2018). Increase in p62/SQSTM1 levels (in presence of the lysosomal fusion blocker bafilomycin A1) in cystinotic cells indicate impaired lysosomal-cargo degradation (Figure 1I) and not mTORC1 inactivity. Overall, our data indicate that CTNS loss induces abnormal autophagy activation (demonstrated by the decreased mTOR activity, increased TFEB nuclear translocation and LC3-accumulation) and hampers lysosomal cargo degradation (demonstrated by increased p62/SQSTM1 levels, decreased processing of DQ-BSA, and overall reduction in lysosomal catalytic-proteins expression) in cystinotic cells. We now explain this in more detail under the result section "CRISPR-generated CTNS^{-/-} ciPTEC display increased cystine accumulation and impaired lysosomal autophagy dynamics".

Question 2:

Fig2: As shown in 2A-B, CTNS patient, CTNS-WT & CTNS^{-/-} have very different & distinct protein expression. Moreover, CTNS^{-/-} behaves similar to WT than CTNS patient. Not sure why for further experiments (Fig 2C, G, H, I) authors compared WT with CTNS^{-/-}. Rather they should have compared WT with CTNS patients. In figure 2D, they have compared CTNS patient with CTNS^{-/-}, which is of no use. This entire figure lacks proper exp planning. Not sure if we can conclude anything substantial from this figure.

One of the main strengths of this study is that we created isogenic cell lines by specifically knocking out *CTNS* using CRISPR technology to study the effect of the cystinosis loss-of-function. Cell lines obtained at different times and from different individuals (*i.e.* patients) often have variable phenotypes, independent of the genetic defect that is being studied. Indeed, figure 2A, B, and F shows that *CTNS*^{Patient} cells account for most of the variability in the data, suggesting that the different genetic background of the patient-derived cells affect the data more than *CTNS* loss itself. As our aim was to study the effect of *CTNS* loss, we focused on the metabolites and proteins that were significantly different between CRISPR-generated *CTNS*^{-/-} cells compared to their isogenic control cells. Further, comparing metabolites that are different in both *CTNS*^{-/-} and *CTNS*^{Patient} cells (figure 2D) allowed us to focus on the metabolites that are robustly changed as a result of *CTNS* loss in both isogenic and non-isogenic cystinotic cells, although in this last comparison more subtle cellular changes will be lost. A schematic figure indicating how *CTNS* loss and genetic background attributes to data variation is demonstrated in the figure below. This strategy is further explained in the results section on *Metabolomic and proteomic profiling*. Using this strategy, we identified increased levels of cystine, cysteine, and aKG in cystinotic cells. Increased cystine and cysteine levels are the real hallmark of the disease so these were expected to be found, but in

addition we identified αKG as an important player. Moreover, we could further confirm this by a significantly increased level of αKG in plasma of cystinotic patients.

Figure on experimental design: Discrimination between genetic background and CTNS loss of cell lines used in the current study

Question 3:

A) Fig3: DMKG (a form of alphaKG) reduced ROS in CTNS-WT & CTNS^{-/-} & increased ROS in CTNS patient. However, DMKG treatment increased apoptosis in CTNS^{-/-} & CTNS patients. It shows that reduced ROS induces apoptosis in CTNS^{-/-}. Whereas increased ROS in CTNS patients also induce apoptosis. How do you explain this?

We thank the reviewer for this question. At baseline, ROS and αKG are consistently increased in both CTNS^{-/-} cells and CTNS^{Patient} cells. αKG is a metabolite known for its antioxidant properties and potential for the treatment of disorders induced by oxidative stress (Liu et al, 2018; Mailloux et al, 2009; Satpute et al, 2010; Starkov, 2013). To study the short-term effect of DMKG on starvation-induced ROS (Figure 2A-D), we supplemented the cells with DMKG for 4hrs. We found that at 4 hrs, DMKG is not toxic in any of the cell lines (Expanded View figure 3A), allowing us to determine the true antioxidant effect of DMKG. In all cell lines, starvation (-AA) led to a strong increase in ROS (starvation induced stress), which could be attenuated by supplementing cells with DMKG for 4 hrs leading to a significant reduction in ROS levels in control cells (2.1-fold). However, this effect was modest in CTNS^{-/-} cells (1.2-fold), and in CTNS^{Patient} cells DMKG even induced ROS production. This shows that DMKG is not as effective as antioxidant in CTNS^{-/-} cells and CTNS^{Patient} as it is in control cells. This could be due to the fact that CTNS loss hampers glutathione synthesis (Chol et al, 2004; Levtchenko et al, 2005; Wilmer, M.J., et al, 2011), and therefore less anti-oxidant effect of DMKG in cystinotic cell lines is observed.

On the other hand, Villar *et al.* showed that during starvation long term incubation with DMKG can induce glutaminolysis-mediated apoptosis though activation of mTOR and subsequent inhibition of autophagy (Villar et al, 2017). Indeed, DMKG treatment for 24 hrs (but not at 4 hrs) induced apoptosis activation in both CTNS^{-/-} & CTNS^{patients}. We here demonstrate that the addition of DMKG to cystinotic proximal tubule cells activates autophagy and mediates apoptosis (Figure 3E-I). Of note, control cells,

handled DMKG well both at 4 and 24 hrs. We now explain this in more detail in the discussion section of our revised manuscript.

B) No significant difference between LC3-II expression in WT-ctrl and patient-ctrl (3I). It is not clear why DMKG reduced LC3-II expression in WT and patient, and not in CTNS^{-/-}.

We indeed demonstrated that at baseline there is no difference in LC3-II expression (Figures 1H-I). However, after inhibiting the autophagic flux (in presence of the lysosomal fusion blocker bafilomycin A1) it becomes clear that LC3-II levels are increased in both CTNS^{-/-} and CTNS^{patient} cells. Stable LC3-II levels in steady state conditions can be explained by the fact that the autophagic flux is very efficient in CTNS^{-/-} and CTNS^{patient} cells, therefore hampering reliable quantification of LC3-II without bafilomycin (Ivanova et al. J Inherit Metab Dis, 2016).

Furthermore, in Figure 3I, we show that treatment with DMKG induces autophagy only in cystinotic cells, demonstrated by increased LC3-II accumulation. DMKG had no effect in control cells hence did not increase the LC3-II accumulation.

C) The cartoon in 3J is too far stretched.

We improved figure 3J (now Figure 7) by focusing on the findings related to CTNS loss and the role of compounds under different conditions.

The blue arrows indicate wild type situation, red arrows/text indicate the up or down regulation of the processes in cystinotic cells, green arrows indicate intervention with medication (cancels out cystinotic Phenotype). DMKG; dimethyl α -ketoglutarate, ROS; Relative reactive oxygen species.

Question 4:

The authors have done many experiments, but no solid conclusion can be derived based on the way the data is presented. This requires extensive focused editing.

We thank the reviewer for the helpful comments. Having adapted our manuscript according to all comments provided, we believe we significantly improved our manuscript.

Referee: 2

MAJOR COMMENTS

Question 1: Data documentation is not optimal.

A) In general, bar histograms are now discouraged and should be replaced by dotplots (as actually used at 1KL; 5E). It is also uncertain, if not dubious, that distributions of continuous variables are normal, thus systematic use of parametric statistical tests for low numbers is highly questionable. Indeed, Fig2E compares plasma aKG concentration in 4 healthy vs 6 patients where boxer-whisker plots (thus not mean \pm SEM as implied from legend) clearly suggest non-normal distribution in patients. Do non-parametric tests confirm significance? At Fig2D, no statistical analysis is presented: is the modest (1.5 fold) increase of the focused metabolite, aKG, in Ctns^{-/-} cells statistically significant? Incidentally, broken ordinate combined with change of increment prevents one to appreciate extent of higher increase aKG in Ctns patient.

We thank the reviewer for this constructive suggestion. In the revised manuscript, we now replaced bar histograms by dot plots wherever this would improve clarity.

In Figure 2E, we indeed used a non-parametric Mann-Whitney test to confirm the significance of the data. However, this was not indicated in our manuscript. We now included the explanation in the legend to the figure to avoid the confusion. Moreover, we also changed the boxer-whisker plots into dot plots of the data and presented as mean \pm SEM.

Figure 2D demonstrates all metabolites that are shared and significantly changed in CTNS^{-/-} and CTNS^{Patient} compared to control cells. We now clearly indicate this in the legend to figure 2 of the revised manuscript. We changed the ordinate to better represent the data and also indicated the statistical analysis in the figure.

B) Numbers of samples at Fig 2E seem really too low for such a crucial piece of information (in fact the only one directly relevant to patients). Larger sample collection of patients and matched volunteers should not be a serious problem.

We thank the reviewer for this constructive suggestion. A larger patient population would be advisable to confirm these preliminary findings. However, approval of an updated study protocol which replaces the older study protocol under which previous samples have been taken, by the Ethical Committee has been significantly delayed due to the Covid-19 crisis. Therefore, unfortunately, we are not able to provide additional data yet.

C) Quality of imaging is not optimal and morphology seems "lightly" interpreted, as if satisfied by pathway confirmation rather than taking into account structural implications. For example, Fig1B is not convincing as "representative": in WT fed cells (intended to validate the assay of co-localization between mTOR and LAMP-1), whereas yellow objects appear indeed to dominate in the cell with selected area, this is not the case in the cell just above; conversely, in the WT starved (-AA) selected cell to document full dissociation, several objects above the nucleus would appear doubly labeled; to the opposite, whereas the fed Ctnspatient cell shows clear dissociation as expected, fed Ctns^{-/-} cells show a multiplicity of yellow spots that would instead suggest unexpected preserved association of mTOR with Lamp-1. Linescans could help. Moreover, a constant finding in cultured cystinotic cells is lysosomal enlargement with perinuclear clustering (Ivanova et al, 2015, quoted; Festa et al, Nat Comm, 2018,

quoted, see beautiful Fig1A). This information should be immediately accessible by LAMP1 immunolabeling as shown at Fig1B, but is regrettably not commented upon. Is this feature reproduced and corrected by bicalutamide?

We thank the reviewer for this comment. Figure 1B (now figure EV2) shows immunofluorescent staining of lysosomal-associated membrane protein 1 (LAMP1; green) and mTOR (Red) in $CTNS^{WT}$, $CTNS^{-/-}$, and $CTNS^{Patient}$ cells. We used different quantification methods and software, but unfortunately were not able to reliably quantify the co-localization between mTOR and LAMP-1 to show the dissociation of the mTOR complex in cystinotic cells. In our cell lines, the size of lysosomes are so small and are generally clumped together hampering reliable analysis of lysosomal perinuclear clustering or quantification of mTOR and LAMP-1 co-localization, hence, provide less reliable readout to assess the effect of potential treatments like bicalutamide. Of note, the downstream effects of mTOR deactivation such as increased TFEB nuclear translocation and LC3-II accumulation can suggest that mTOR activity is decreased in both cystinotic cell lines. To avoid suboptimal interpretation of the data, we therefore moved figure 1B to the fig EV2 and explained the technical hurdles in the result section.

D) Careful manuscript revision is still needed. For example, legend of Fig 1C indicates three types of cells, yet only two images are shown. Its quantitation refers to fold-change (from 1 in WT to 2.5), yet image shows at left (WT?) no TFEB nuclear signal (accordingly, baseline should be 0, not 1 ?). Larger fields should be presented.

We thank the reviewer for this observation. Figure 1C (now figure 1B) demonstrates only TFEB cellular distribution irrespective of the cell lines used. In the revised manuscript, we now more clearly indicate this in the legend and also provided a representative confocal micrograph with a larger field demonstrating TFEB cellular distribution.

Figure 1D (now figure 1C) demonstrates the quantification of TFEB-GFP nuclear translocation in $CTNS^{WT}$, $CTNS^{-/-}$, and $CTNS^{Patient}$ cells. We first determined the ratio between number of the cells with nucleus-TFEB positive over the total number of TFEB-transfected cells. The ratios were then presented as fold change compared to control cells. The analysis was carried out blindly by analyzing over 300 cells per cell line. This explanation is now included in the legend to figure 1C.

E) This reviewer was not impressed by the complex cartoon presenting the working model at Fig 3J with multiple effects. It is suggested to both simplify to core effects (sequence of steps of the TCA cycle, available in any biochemistry textbook, can be omitted) and to present a sequence of situations : (i) untreated cystinosis; (ii) aKG manipulation by DKMG; (iii) treatment by cysteamine alone; (iv) treatment by bicalutamide ; (v) combined treatments.

We thank the reviewer for this suggestion and redesigned the working model in the revised manuscript.

Question 2:

2. Lack of mitochondrion imaging. Considering that ROS production in cystinotic cells (e.g. Festa et al, 2018) and targeted TCA cycle both occur in mitochondria, it is regretted that mitochondrion imaging is not presented, at least for the key points of the demonstration. There are excellent easy tool for discriminating ROS accumulation in mitochondria vs lysosomes (e.g. Denamur et al, Free Radic Biol Med, 2011, Fig3) and immunolabeling can readily evidence distorted (or normalized) mitochondrial shape. Such a subcellular structural study would not be that time-consuming, while mechanistically strengthening the biochemical/cell based demonstration, thus is strongly recommended.

We thank the reviewer for this constructive suggestion. To demonstrate mitochondrial involvement, we used MitoSox and CellRox reagents to discriminate ROS accumulation in mitochondria and lysosomes upon starvation in the presences and absence of DMKG, respectively. Unfortunately, we were not able to detect any MitoSox fluorescence intensity in our control and cystinotic cells under the desired experimental conditions, unabling us to incorporate the mitochondrial involvement in the cystinosis pathophysiology. In the revised manuscript, under discussion section, we now propose to unravel the role of mitochondria in the progression of cystinosis as a future work.

Question 3:

Membrane permeant metabolites are tricky and not a priori bona fide equivalents. In the field of cystinosis itself, membrane permeant cystine derivatives got a bad reputation as being proved toxic by themselves to mitochondria (referenced by Cherqui, Nat Rev Nephrol, 2017; quoted). Fortunately, control of intrinsic lack of toxicity after 24 h at 2mM DMKG in fed cystinotic cells is reinsuring. However, considering DMKG as equivalent to aKG seems far-fetched (e.g. see page 9, para 3, line 5 : "in presence of aKG" when data show "under DMKG supplementation". Moreover, short-term effects of DMKG on ROS are significantly opposite in the two cystinosis-deficient cell lines. Thus, comparison of intracellular levels of aKG and DMKG, and the effect of DMKG supplementation on actual aKG level should be quite informative and thus also strongly recommended. This clarification is all the more important since aKG effects are discussed as context-dependent.

We thank the reviewer for this excellent suggestion and agree that DMKG may not be equivalent to aKG. Therefore, in the revised manuscript, we changed aKG to DMKG wherever appropriate.

The metabolite aKG is membrane-impermeable, meaning that it is usually added to cells in the form of esters such as DMKG. Once DMKG crosses the plasma membrane, it is hydrolyzed by esterases to generate aKG, which remains trapped within cells. Baracco et al (Aging, 2019), using mass spectrometric metabolomics, demonstrated that addition of DMKG can indeed increase the intracellular levels of aKG in human osteosarcoma U2OS and human neuroglioma H4 cells. Similarly, Villar et al. (2017) used DMKG (2 mM) as a source of aKG in their experiments, demonstrating that it dysregulates autophagy and induces glutaminolysis-mediated apoptosis in cancer cells. Therefore, we feel confident that addition of DMKG (2 mM) can lead to increased intracellular levels of aKG in our cell model. Further, the metabolite is known for its antioxidant properties and potential for the treatment of disorders induced by oxidative stress (Liu et al, 2018; Mailloux et al, 2009; Satpute et al, 2010; Starkov, 2013). Indeed, supplementing cells with DMKG for 4 hrs led to a significant reduction in ROS levels in control cells (2.1-fold). However, this effect was modest in *CTNS*^{-/-} cells (1.2-fold), and in *CTNS*^{Patient} cells DMKG rather induced ROS production. This could be due to the fact that CTNS loss hampers glutathione synthesis (Chol et al, 2004; Levchenko et al, 2005; Wilmer, M.J., et al, 2011), and therefore less antioxidant effect of DMKG in cystinotic cell lines is observed. We now explain this in the discussion.

Of note, the metabolomic data show that *CTNS*^{patients} cells already have a higher intracellular aKG level compared to *CTNS*^{-/-} cells, which could explain their adverse response to more DMKG. This also holds true in both cell viability and apoptosis assays where *CTNS*^{patients} cells were more sensitive to DMKG compared to *CTNS*^{-/-} cells.

Question 4:

Conclusion that aKG is a key metabolite should be explicitly presented with more caution (page 8, first para, last line). Besides aKG, several other metabolites were modified by cystinosis and corrected by

bicalutamide (Fig4A). Thus, while aKG was useful reporter to eventually identify bicalutamide as useful drug, drug benefits cannot be safely attributed to aKG as the "key actor" (or implicitly presented as such).

We thank the reviewer for her/his constructive suggestion and agree that the role of aKG may have been overstated. In the revised manuscript, the text now reads: 'we identified α KG as an important metabolite bridging cystinosis loss to increased oxidative stress, autophagy disruption, and proximal tubule dysfunction in cystinosis.'

MINOR COMMENTS

We thank the reviewer for pointing out the following errors which we addressed in the revised manuscript.

Question 5:

A) Abstract, line 1 : monogenic

We now replaced the word monogenetic to monogenic in the revised manuscript.

B) Page 3, para 1, line 3 : "leading"; stricto sensu, Fanconi syndrome in general does not per se lead to chronic kidney failure; it certainly does so in nephropathic cystinosis (NC).

In the revised manuscript, the text now reads: 'The first clinical signs develop during infancy (age of ~6 months) in the form of renal Fanconi syndrome and over time patients develop chronic kidney disease and finally renal failure in the first or second decade of life (Gahl et al, 2002).'

C) Page 3, para 3, line 4 : it would be appropriate to already quote here the landmark paper by Festa et al, Nat Comm, 2018.

We added the reference (Festa et al, Nat Comm, 2018) in the revised manuscript.

D) Page 6, first para, last line : please clarify to which cell (Ctns-/- ?) this conclusion applies. Conversely, would the authors conclude that Ctnspatient cells are endocytosis- but not degradation-defective?

The text now reads: indicating the defect is related to degradation rather than protein uptake in *CTNS*^{-/-} cells.

E) Fig legend 1B: scale bar of 13 micron is awkward.

We now readjusted the confocal micrographs to 10 μ m.

G) Fig2I. IGF2R/Cation-independent mannose 6-phosphate receptor. It is suggested (page 7, end of first para) that this decrease would cause defective lysosomal targeting of lysosomal enzymes by rerouting to the constitutive secretory pathway, thus could contribute to explain their paradoxical decrease in

steady-state cell content, despite global transcriptional activation by TFEB (see Fig 1D). However, defective lysosomal proteolysis is instead attributed to impaired activation of pro-cathepsin C (Festa et al, Nat Comm, 2018, quoted : Fig3c), whose level actually increases. Moreover, as to lysosomal acid phosphatase, it is known not to be targeted to lysosomes as a soluble cargo by a M6P anchor but rather assembled as a transmembrane protein which is eventually released into lysosomal lumen upon proteolytic release from its anchor (classical work from von Figura's lab). Please clarify and/or rewrite accordingly.

We thank the reviewer for this constructive suggestion. Indeed, Festa *et al*, postulated that impairment of lysosome proteolysis observed both *in vitro* and *in vivo* is as a result of defective cathepsin activation despite an efficient delivery of newly synthesized cathepsins from Golgi to lysosomes.

The text now reads: 'Consistent with the reduced cargo degradation in cystinotic cells, we found an overall reduction in lysosomal catalytic-proteins expression including, lysosomal acid lipase (LIPA), lysosomal acid phosphatase (ACP2), and cysteine proteases cathepsin (CTSS) (Fig 2I). Moreover, cation-independent mannose 6-phosphate receptor (IGF2R) was found downregulated in *CTNS*^{-/-} cells (Fig 2I). IGF2R is responsible for the delivery of many newly synthesized lysosomal enzymes from Golgi to the lysosome, and its downregulation could result in decreased lysosomal activity and production of ROS (Probst et al, 2006; Takeda et al, 2019).'

H) Fig3A : for consistency, replace Ctns KO by Ctnspatient.

Thank you for your thorough reading and pointing out this error. We now replaced the *CTNS*^{KO} by *CTNS*^{patient} in figure 3A of the revised manuscript.

I) Fig5A: cell death in starved (-AA) 2-mM DMKG-treated Ctns-/- cells is 35% here vs 70% at Fig3E. Please clarify.

In figure 3E, the data is presented relative to the fed condition, however, in figure 5A data is presented relative to the starved condition.

In figure 3E, addition of DMKG to starved (-AA) *CTNS*^{-/-} cells resulted in additional 35% cell death compared to -AA *CTNS*^{-/-} without DMKG. Since, the data is presented relative to the fed condition, it is shown that cell death in starved DMKG-treated *CTNS*^{-/-} cells is 70%. In figure 5A, however, the data is presented relative to the starved condition. Hence, addition of DMKG resulted in a similar 35% cell death. For clarity, in the revised manuscript we now explain this in the figure's legends.

J) Discussion, page 14, lines 9-11: should be deleted; following sentences should also be rewritten. (i) Cysteamine can improve, but certainly does not fully correct cystinosis manifestations and lesions in mice. (ii) The authors may be informed of a powerful rat model, with more robust and earlier phenotype, generated by three different laboratories. (iii) The zebrafish model is useful for screening yet may generate symptoms irrelevant for patients (embryonic dysmorphism; cystinotic babies are born normal)

We fully agree with reviewer 2 that the discussion on animal models could be expanded. However, most of this information, such as details on the rat and mouse models, is not publicly available which hampers a clear evaluation.

Discussion, page 14, lines 9-11 read: “Furthermore, unlike the human situation where treatment with cysteamine does not improve the renal phenotype and only improves some parts of the cystinotic phenotype (Cherqui, 2012), it can completely resolve any renal symptoms in mice.” As we initially aimed to evaluate the cysteamine-bicalutamide treatment in mice, we already discussed this issue extensively with our collaborators at the Renal Diseases Research Unit at Bambino Gesù Children’s Hospital (Rome). Their data shows that cysteamine can completely resolve any renal symptoms in the above mentioned mouse model. This data has not yet been published (manuscript currently under revision), but we obtained permission from the group of Prof Dr. Francesco Emma to confidentially share these results with you, ahead of print (below). Their data show that daily treatment of *Ctns*^{-/-} mice (from 2-to-14 months of age) with cysteamine (500 mg/kg body weight/day) drastically reduces cystine and urinary levels of low molecular weight protein CC16 to the levels found in the wild type mice, opposing clinical features in human.

		Kidney cystine (mmol / mg proteins)		
		Ctns +/+	Ctns -/-	Ctns -/- + MEA
mean		0.40	7.24	0.89
SEM		0.14	1.56	0.41

		CC16 (ug / g creatinine)		
		Ctns +/+	Ctns -/-	Ctns -/- + MEA
mean		65.5	502.3	92.0
SEM		31.6	118.7	26.2

We, therefore, considered other models that could be used to evaluate the cysteamine/bicalutamide combination therapy and were able to develop a cystinotic patient-derived kidney organoids without genetic modifications, as described earlier for healthy kidney and urine samples (Schutgens et al, 2019). This unique primary human tissue culture along with cystinotic zebrafish (Elmonem 2017; Elmonem 2018) appeared to be robust and versatile models of cystinosis with early phenotypic characteristics of the disease, which are also suitable for the *in vitro* and *in vivo* screening of novel therapeutic agents. These emerging models are now gaining interest in the field as seen by the recent publication on the beneficial effect of cysteamine combined with the mTOR inhibitor everolimus in cystinotic kidney organoids (Holiwood 2020), and the use of cystinotic zebrafish in evaluating luteolin as a potential treatment for cystinosis (De Leo 2020, JASN).

Three different *Ctns* rat models have been developed so far, however, to our knowledge only one has been published (Shimizu 2019, Mammalian Genome). This *CTNS*^{-/-} spontaneously formed in Long-Evans Agouti rat and its relevance to study the cystinosis phenotype has yet to be established. The two other models (generated using CRISPR-Cas9 in the labs of Devuyst and Hollywood) look promising but have not yet been published or shared with the academic society. We hence feel it is better at this time not to expand on unpublished models.

The zebrafish is gaining popularity for drug screening and toxicity studies as it combines multiple organ functions that cannot be recapitulated in *in vitro* models. Indeed, the dysmorphic features are not

directly linked to clinical features in patients, but they provide a phenotypic readout that can be quantified for analysis. We now explain this in the discussion section.

K) Supp Fig 3. The rationale for including toxicity data of other drugs such as genistein, disulfiram etc..in this particular study is unclear.

In our study, we screened different candidate drugs based on their ability to restore the metabolic profile using metabolomics. Based on this, we continued with bicalutamide as our most promising compound. To ensure that the concentrations of the drugs tested were within a non-cytotoxic range, we performed toxicity curves for all the drugs tested which are indicated in EV3. This is also explained in the result section, lines 212-215.

Referee: 3

This is an innovative investigative approach to characterize the cellular mechanisms responsible for the dysregulation of proximal tubular function in experimental cystinosis models. The authors used immortalized cell culture lines of cystinosis and control tissue, and a unique application of organ tubuloids from cystinosis patients, and finally, a zebrafish model of cystinosis. This combination of experimental approaches with advanced proteomic and metabolomic analyses, provided new and encouraging data on the specific cellular mechanisms responsible for the varying phenotypes and pathophysiology of cystinosis. Indeed, these novel approaches have totally circumvented the numerous shortcomings of the in vivo models of cystinosis where there are limited renal phenotypes.

Question 1:

One minor suggestion would be to possibly expand upon the findings to point to further development of drug interventions for testing in special populations to advance the field.

We highly appreciate the positive remarks by this reviewer. To expand on the finding and to point to further development of our drug interventions, we now included the following text in the discussion section: 'Preclinical studies in a suitable animal model should determine if cysteamine-bicalutamide combination therapy is able to improve or prevent the development of renal Fanconi syndrome and renal failure. Translation to the clinic should be further facilitated by the fact that bicalutamide is already an approved drug with a known safety profile, although a separate study should be performed to determine the appropriate dose for cystinotic patients.'

20th Nov 2020

Dear Dr. Janssen,

Thank you for the submission of your manuscript to EMBO Molecular Medicine. We have now received feedback from the two reviewers who agreed to re-evaluate your manuscript. As you will see from the reports below, the referees acknowledge your efforts in addressing the referees' comments but also raise serious concerns that should be addressed in a second, final round of revision. Addressing the reviewers' concerns in full will be necessary for further considering the manuscript in our journal, and acceptance of the manuscript will entail a third round of review. In addition to addressing the referees' comments please amend the following:

- 1) Figures: Please upload 1 file per figure.
- 2) Tables: Please rename Tables EV1 and EV2 to Table 1 and Table 2.
- 3) In the main manuscript file, please do the following:
 - Most of the panels in the EV figures are not referenced in the text, please correct.
 - In M&M, the statistical paragraph should reflect all information that you have filled in the Author Checklist, especially regarding randomization, blinding, replication.
 - Indicate in legends exact n= and exact p= values, not a range, along with the statistical test used. To keep the figures "clear" some authors found providing an Appendix table Sx with all exact p-values preferable. You are welcome to do this if you want to.
 - Use the following format to report the accession number of your data:

The datasets produced in this study are available in the following databases:
[data type]: [full name of the resource] [accession number/identifier] ([doi or URL or identifiers.org/DATABASE:ACCESSION])

Please check "Author Guidelines" for more information.

<https://www.embopress.org/page/journal/17574684/authorguide#availabilityofpublishedmaterial>

- 4) Source Data: Please split source data for Expanded View figures and upload one file per figure.
- 5) Appendix: Please remove the appendix figure legend from main MS and add to appendix file together with table of content.
- 6) Expanded View: Please merge the files for Dataset EV1 into one excel file, remove the legends for Dataset EV1 and EV2 from the main manuscript file and add them to the files in a new tab of dataset excel files.
- 7) For more information: There is space at the end of each article to list relevant web links for further consultation by our readers. Could you identify some relevant ones and provide such information as well? Some examples are patient associations, relevant databases, OMIM/proteins/genes links, author's websites, etc...
- 8) As part of the EMBO Publications transparent editorial process initiative (see our Editorial at <http://embomolmed.embopress.org/content/2/9/329>), EMBO Molecular Medicine will publish online a Review Process File (RPF) to accompany accepted manuscripts. This file will be published in conjunction with your paper and will include the anonymous referee reports, your point-by-point response and all pertinent correspondence relating to the manuscript. Let us know whether you agree with the publication of the RPF and as here, if you want to remove or not any figures from it prior to publication. Please note that the Authors checklist will be published at the end of the RPF.

9) Please provide a point-by-point letter INCLUDING my comments as well as the reviewer's reports and your detailed responses (as Word file).

We realize that the current situation is exceptional on the account of the COVID-19/SARS-CoV-2 pandemic. Therefore, please let us know if you need more than three months to revise the manuscript.

I look forward to receiving your revised manuscript.

Yours sincerely,

Zeljko Durdevic

***** Reviewer's comments *****

Referee #1 (Comments on Novelty/Model System for Author):

Detailed comments are provided below

Referee #1 (Remarks for Author):

Referee: 1

Question 1:

A) I do not see any significant difference in IF (1B) between fed CTNS-WT & CTNS^{-/-}. They looked very similar to each other and different from CTNS-patient.

We thank the reviewer for this comment. Figure 1B (now figure EV2) shows immunofluorescent staining of lysosomal-associated membrane protein 1 (LAMP1; green) and mTOR (red) in CTNS^{WT}, CTNS^{-/-}, and CTNS^{Patient} cells. In control cells, mTOR respond to the nutrient status of the cells. In the presence of nutrients (standard medium, fed condition) mTOR co-localizes with the lysosomal membrane (in yellow), whereas upon starvation (-AA) mTOR was released from the lysosomes (less colocalization) and upon reintroduction of nutrients the co-localization is re-established. However, mTOR in CTNS^{-/-} and CTNS^{Patient} cells was already released from lysosomes in the fed condition and a shift based on the nutritional status of the cells was not observed. This indicates decreased mTOR activity in cystinotic cells also under fed condition. Moreover, downstream effects of mTOR deactivation such as increased TFEB nuclear translocation and LC3-II accumulation were also observed, further confirming that mTOR activity is decreased in both cystinotic cell lines. This is in accordance with other studies where they have shown the decreased retention of mTOR on the lysosomal membrane, and therefore the decreased activity of mTOR in in vitro and in vivo models of cystinosis (Andrzejewska, Z. et al. J Am Soc Nephrol, 2016; Festa et al, Nat communication, 2018; Ivanova et al, PLOSone, 2015, Ivanova et

al. J Inherit Metab Dis 2016).

Unfortunately, technical hurdles did not allow us to quantify the co-localization between mTOR and LAMP-1 to show dissociation of the mTOR complex in cystinosis. In our cell lines, the size of lysosomes are so small and are generally clumped together hampering reliable quantification. Therefore, we moved figure 1B to the figure EV2 and explained the technical hurdles in the result section.

Re-Rev: In this figure, LAMP1 colocalizes with mTOR in CTNS-WT but there is significant less colocalization in CTNS-patient group as interpreted by the authors. However, I agree with referee 1 that we can still see significant amount of colocalization (yellow puncta) in the CTNS -/- group, maybe less than the wild type but the staining pattern is more similar to the wild type than CTNS-patient. In other words, CTNS -/- group is more similar to the CTNS-WT than the CTNS-patient cells. Overall, the colocalization is high in wildtype, less in CTNS -/- group and almost no colocalization in CTNS-patient. I think this needed to be mentioned in the manuscript.

B) No visual change was noted in the SQSTM1 protein expression (1H, J) in CTNS patient. Not sure how they got significant difference in expression in the bar chart. Also, they looked at total SQSTM1 protein expression, and not the active form. Therefore, this doesn't represent inactivity of mTORC1 in Cystinosis. You can relate less active SQSTM1 and LC3-II with increased autophagy & inactive mTORC1, which was not shown in this paper. Their CTNS-/- cell line is more similar to WT than patient RPTECs.

We thank the reviewer for this constructive suggestion. We agree that the Western blot shown was not very clear regarding SQSTM1 protein expression in CTNSPatient cells (now figure 1H, I). For the bar graphs, we combined data from three independent experiments (see blot images below).

In accordance with other studies, we used total SQSTM1 protein expression to demonstrate lysosomal degradation and autophagic flux in cystinosis (Sansanwal, *Pediatr Nephrol*, 2012; Festa et al, *Nat Communication*, 2018). Increase in p62/SQSTM1 levels (in presence of the lysosomal fusion blocker bafilomycin A1) in cystinotic cells indicate impaired lysosomal-cargo degradation (Figure 1I) and not mTORC1 inactivity. Overall, our data indicate that CTNS loss induces abnormal autophagy activation (demonstrated by the decreased mTOR activity, increased TFEB nuclear translocation and LC3- accumulation) and hampers lysosomal cargo degradation (demonstrated by increased p62/SQSTM1 levels, decreased processing of DQ-BSA, and overall reduction in lysosomal catalytic-proteins expression) in cystinotic cells. We now explain this in more detail under the result section "CRISPR- generated CTNS-/- ciPTEC display increased cystine accumulation and impaired lysosomal autophagy dynamics".

Re-Rev: Authors have efficiently justified this concern. However, could they provide the blots for the housekeeping genes of the given protein expression?

Question 2:

Fig2: As shown in 2A-B, CTNS patient, CTNS-WT & CTNS-/- have very different & distinct protein expression. Moreover, CTNS-/- behaves similar to WT than CTNS patient. Not sure why for further experiments (Fig 2C, G, H, I) authors compared WT with CTNS-/-. Rather they should have compared WT with CTNS patients. In figure 2D, they have compared CTNS patient with CTNS-/-,

which is of no use. This entire figure lacks proper exp planning. Not sure if we can conclude anything substantial from this figure.

One of the main strengths of this study is that we created isogenic cell lines by specifically knocking out CTNS using CRISPR technology to study the effect of the cystinosis loss-of-function. Cell lines obtained at different times and from different individuals (i.e. patients) often have variable phenotypes, independent of the genetic defect that is being studied. Indeed, figure 2A, B, and F shows that CTNSPatient cells account for most of the variability in the data, suggesting that the different genetic background of the patient-derived cells affect the data more than CTNS loss itself. As our aim was to study the effect of CTNS loss, we focused on the metabolites and proteins that were significantly different between CRISPR-generated CTNS^{-/-} cells compared to their isogenic control cells. Further, comparing metabolites that are different in both CTNS^{-/-} and CTNSPatient cells (figure 2D) allowed us to focus on the metabolites that are robustly changed as a result of CTNS loss in both isogenic and non-isogenic cystinotic cells, although in this last comparison more subtle cellular changes will be lost. A schematic figure indicating how CTNS loss and genetic background attributes to data variation is demonstrated in the figure below. This strategy is further explained in the results section on Metabolomic and proteomic profiling. Using this strategy, we identified increased levels of cystine, cysteine, and aKG in cystinotic cells. Increased cystine and cysteine levels are the real hallmark of the disease so these were expected to be found, but in addition we identified aKG as an important player. Moreover, we could further confirm this by a significantly increased level of α KG in plasma of cystinotic patients.

Re-Rev: I agree with the reviewer that CTNS^{-/-} cells behave more like the wild type cells and do not mimic the cystinosis disease phenotype. In spite of the background differences all the patients grouped together in the PCA plot, which is completely separate than the CTNS^{-/-}. Authors should include proper normal control to compare with cystinosis patients sample instead of using these wildtype cells as the common control for both the cell types. And wildtype should only be compared with CTNS^{-/-} cells.

Figure on experimental design: Discrimination between genetic background and CTNS loss of cell lines used in the current study

Question 3:

A) Fig3: DMKG (a form of α KG) reduced ROS in CTNS-WT & CTNS^{-/-} & increased ROS in CTNS patient. However, DMKG treatment increased apoptosis in CTNS^{-/-} & CTNS patients. It shows that reduced ROS induces apoptosis in CTNS^{-/-}. Whereas increased ROS in CTNS patients also induce apoptosis. How do you explain this?

We thank the reviewer for this question. At baseline, ROS and aKG are consistently increased in both CTNS^{-/-} cells and CTNSPatient cells. aKG is a metabolite known for its antioxidant properties and potential for the treatment of disorders induced by oxidative stress (Liu et al, 2018; Mailloux et al, 2009; Satpute et al, 2010; Starkov, 2013). To study the short-term effect of DMKG on starvation-induced ROS (Figure 2A-D), we supplemented the cells with DMKG for 4hrs. We found that at 4 hrs, DMKG is not toxic in any of the cell lines (Expanded View figure 3A), allowing us to determine the true antioxidant effect of DMKG. In all cell lines, starvation (-AA) led to a strong increase in ROS (starvation induced stress), which could be attenuated by supplementing cells with DMKG for 4 hrs leading to a significant reduction in ROS levels in control cells (2.1-fold). However, this effect was

modest in CTNS^{-/-} cells (1.2-fold), and in CTNSPatient cells DMKG even induced ROS production. This shows that DMKG is not as effective as antioxidant in CTNS^{-/-} cells and CTNSPatient as it is in control cells. This could be due to the fact that CTNS loss hampers glutathione synthesis (Chol et al, 2004; Levtchenko et al, 2005; Wilmer, M.J., et al, 2011), and therefore less anti-oxidant effect of DMKG in cystinotic cell lines is observed.

On the other hand, Villar et al. showed that during starvation long term incubation with DMKG can induce glutaminolysis-mediated apoptosis through activation of mTOR and subsequent inhibition of autophagy (Villar et al, 2017). Indeed, DMKG treatment for 24 hrs (but not at 4 hrs) induced apoptosis activation in both CTNS^{-/-} & CTNSpatients. We here demonstrate that the addition of DMKG to cystinotic proximal tubule cells activates autophagy and mediates apoptosis (Figure 3E-I). Of note, control cells, handled DMKG well both at 4 and 24 hrs. We now explain this in more detail in the discussion section of our revised manuscript.

Re-Review: DMKG acts as an antioxidant for both wildtype and CTNS^{-/-} as it significantly reduces the ROS in both these cell types. No matter how modest the decrease of ROS is in CTNS^{-/-}, it is still a significant decrease, which cannot be ignored. On the other hand, DMKG significantly induces ROS in cystinosis-patients. The difference in pattern can't be explained. Also, when all the cells were fed, they had comparable level of ROS but when starved and treated with DMKG then the pattern in the isogenic cells are completely opposite of cystinosis-patients. This further proves that cystinosis-patients should have its own control and can't be compared to a common control.

As per Villars et al, long-term exposure of DMKG inhibits autophagy and induces apoptosis, but your data shows that DMKG induces autophagy and apoptosis? Both apoptosis and autophagy are very distinct cell death/recycle process, please could you explain how DMKG is activating autophagy and mediating apoptosis at the same time by activating mTORC pathway?

B) No significant difference between LC3-II expression in WT-ctrl and patient-ctrl (3I). It is not clear why DMKG reduced LC3-II expression in WT and patient, and not in CTNS^{-/-}.

We indeed demonstrated that at baseline there is no difference in LC3-II expression (Figures 1H-I). However, after inhibiting the autophagic flux (in presence of the lysosomal fusion blocker bafilomycin A1) it becomes clear that LC3-II levels are increased in both CTNS^{-/-} and CTNSPatient cells. Stable LC3-II levels in steady state conditions can be explained by the fact that the autophagic flux is very efficient in CTNS^{-/-} and CTNSPatient cells, therefore hampering reliable quantification of LC3-II without bafilomycin (Ivanova et al. J Inherit Metab Dis, 2016). Furthermore, in Figure 3I, we show that treatment with DMKG induces autophagy only in cystinotic cells, demonstrated by increased LC3-II accumulation. DMKG had no effect in control cells hence did not increase the LC3-II accumulation.

Re-Review: On the contrary to what the authors mention, visually (Fig. 3H) all the 3 cell types have significant increase in LC3-II with BafA1 under starvation, and not only CTNS^{-/-} and CTNS-Patient cells. I agree with the authors on their take on DMKG treatment and LC3-II induction.

C) The cartoon in 3J is too far stretched.

We improved figure 3J (now Figure 7) by focusing on the findings related to CTNS loss and the role of compounds under different conditions.

The blue arrows indicate wild type situation, red arrows/text indicate the up or down regulation of the processes in cystinotic cells, green arrows indicate intervention with medication (cancels out cystinotic Phenotype). DMKG; dimethyl α -ketoglutarate, ROS; Relative reactive oxygen species. Re-Review: Based on the data, I do not think it is fair to combine CTNS -/- and Cystinosis-patient as cystinotic cells. Since, both these cell types are unique and have very less commonality at the molecular level.

Question 4:

The authors have done many experiments, but no solid conclusion can be derived based on the way the data is presented. This requires extensive focused editing.

We thank the reviewer for the helpful comments. Having adapted our manuscript according to all comments provided, we believe we significantly improved our manuscript.

Re-Review: I agree with the reviewer that this paper still requires focused editing and data driven conclusions.

Referee: 2

MAJOR COMMENTS

Question 1: Data documentation is not optimal.

A) In general, bar histograms are now discouraged and should be replaced by dotplots (as actually used at 1KL; 5E). It is also uncertain, if not dubious, that distributions of continuous variables are normal, thus systematic use of parametric statistical tests for low numbers is highly questionable. Indeed, Fig2E compares plasma aKG concentration in 4 healthy vs 6 patients where boxer-whisker plots (thus not mean \pm SEM as implied from legend) clearly suggest non-normal distribution in patients. Do non- parametric tests confirm significance? At Fig2D, no statistical analysis is presented: is the modest (1.5 fold) increase of the focused metabolite, aKG, in Ctns-/- cells statistically significant? Incidentally, broken ordinate combined with change of increment prevents one to appreciate extent of higher increase aKG in Ctns patient.

We thank the reviewer for this constructive suggestion. In the revised manuscript, we now replaced bar histograms by dot plots wherever this would improve clarity.

In Figure 2E, we indeed used a non-parametric Mann-Whitney test to confirm the significance of the data. However, this was not indicated in our manuscript. We now included the explanation in the legend to the figure to avoid the confusion. Moreover, we also changed the boxer-whisker plots into dot plots of the data and presented as mean {plus minus} SEM.

Figure 2D demonstrates all metabolites that are shared and significantly changed in CTNS-/- and CTNSPatient compared to control cells. We now clearly indicate this in the legend to figure 2 of the revised manuscript. We changed the ordinate to better represent the data and also indicated the statistical analysis in the figure.

Re-Review: The concerns raised here are sufficiently addressed.

B) Numbers of samples at Fig 2E seem really too low for such a crucial piece of information (in fact the only one directly relevant to patients). Larger sample collection of patients and matched

volunteers should not be a serious problem.

We thank the reviewer for this constructive suggestion. A larger patient population would be advisable to confirm these preliminary findings. However, approval of an updated study protocol which replaces the older study protocol under which previous samples have been taken, by the Ethical Committee has been significantly delayed due to the Covid-19 crisis. Therefore, unfortunately, we are not able to provide additional data yet.

Re-Review: ok.

C) Quality of imaging is not optimal and morphology seems "lightly" interpreted, as if satisfied by pathway confirmation rather than taking into account structural implications. For example, Fig1B is not convincing as "representative": in WT fed cells (intended to validate the assay of co-localization between mTOR and LAMP-1), whereas yellow objects appear indeed to dominate in the cell with selected area, this is not the case in the cell just above; conversely, in the WT starved (-AA) selected cell to document full dissociation, several objects above the nucleus would appear doubly labeled; to the opposite, whereas the fed Ctnspatient cell shows clear dissociation as expected, fed Ctns^{-/-} cells show a multiplicity of yellow spots that would instead suggest unexpected preserved association of mTOR with Lamp-1. Linescans could help. Moreover, a constant finding in cultured cystinotic cells is lysosomal enlargement with perinuclear clustering (Ivanova et al, 2015, quoted; Festa et al, Nat Comm, 2018,

quoted, see beautiful Fig1A). This information should be immediately accessible by LAMP1 immunolabeling as shown at Fig1B, but is regrettably not commented upon. Is this feature reproduced and corrected by bicalutamide?

We thank the reviewer for this comment. Figure 1B (now figure EV2) shows immunofluorescent staining of lysosomal-associated membrane protein 1 (LAMP1; green) and mTOR (Red) in CTNSWT, CTNS^{-/-}, and CTNSPatient cells. We used different quantification methods and software, but unfortunately were not able to reliably quantify the co-localization between mTOR and LAMP-1 to show the dissociation of the mTOR complex in cystinotic cells. In our cell lines, the size of lysosomes are so small and are generally clumped together hampering reliable analysis of lysosomal perinuclear clustering or quantification of mTOR and LAMP-1 co-localization, hence, provide less reliable readout to assess the effect of potential treatments like bicalutamide. Of note, the downstream effects of mTOR deactivation such as increased TFEB nuclear translocation and LC3-II accumulation can suggest that mTOR activity is decreased in both cystinotic cell lines. To avoid suboptimal interpretation of the data, we therefore moved figure 1B to the fig EV2 and explained the technical hurdles in the result section.

Re-Review: The question is do we even need this figure? The authors interpretation of this figure is misleading. I agree with Referee 1 & 2 that there isn't any visual difference between Fed & -AA in wild types. Coexpression pattern in CTNS^{-/-} is similar to the wild type and not even close enough to the CTNS-Patient pattern of coexpression. The fed CTNS-patient cell are the only type that shows clear dissociation as expected.

D) Careful manuscript revision is still needed. For example, legend of Fig 1C indicates three types of cells, yet only two images are shown. Its quantitation refers to fold-change (from 1 in WT to 2.5), yet image shows at left (WT?) no TFEB nuclear signal (accordingly, baseline should be 0, not 1 ?). Larger fields should be presented.

We thank the reviewer for this observation. Figure 1C (now figure 1B) demonstrates only TFEB cellular distribution irrespective of the cell lines used. In the revised manuscript, we now more clearly indicate this in the legend and also provided a representative confocal micrograph with a larger field demonstrating TFEB cellular distribution.

Figure 1D (now figure 1C) demonstrates the quantification of TFEB-GFP nuclear translocation in CTNSWT, CTNS^{-/-}, and CTNSPatient cells. We first determined the ratio between number of the cells with nucleus- TFEB positive over the total number of TFEB-transfected cells. The ratios were then presented as fold change compared to control cells. The analysis was carried out blindly by analyzing over 300 cells per cell line. This explanation is now included in the legend to figure 1C.

Re-Review: If figure 1 B is not specific to any cell types mentioned in this paper then I do not think it is necessary at all. If the authors really want to include this IF figure then they might as well include each of the celltypes highlighting the TFEB localization change in each of them.

E) This reviewer was not impressed by the complex cartoon presenting the working model at Fig 3J with multiple effects. It is suggested to both simplify to core effects (sequence of steps of the TCA cycle, available in any biochemistry textbook, can be omitted) and to present a sequence of situations : (i) untreated cystinosis; (ii) aKG manipulation by DMKG; (iii) treatment by cysteamine alone; (iv) treatment by bicalutamide ; (v) combined treatments.

We thank the reviewer for this suggestion and redesigned the working model in the revised manuscript.

Re-Review: The authors have simplified the representation of the study design but it still does not convey the entire story. They have not included the antioxidant effect of DMKG on CTNS ^{-/-} cell types, which was opposite in CTNS-Patient cell types (which they have included).

Question 2:

2. Lack of mitochondrion imaging. Considering that ROS production in cystinotic cells (e.g. Festa et al, 2018) and targeted TCA cycle both occur in mitochondria, it is regretted that mitochondrion imaging is not presented, at least for the key points of the demonstration. There are excellent easy tool for discriminating ROS accumulation in mitochondria vs lysosomes (e.g. Denamur et al, Free Radic Biol Med, 2011, Fig3) and immunolabeling can readily evidence distorted (or normalized) mitochondrial shape. Such a subcellular structural study would not be that time-consuming, while mechanistically strengthening the biochemical/cell based demonstration, thus is strongly recommended.

We thank the reviewer for this constructive suggestion. To demonstrate mitochondrial involvement, we used MitoSox and CellRox reagents to discriminate ROS accumulation in mitochondria and lysosomes upon starvation in the presences and absence of DMKG, respectively. Unfortunately, we were not able to detect any MitoSox fluorescence intensity in our control and cystinotic cells under the desired experimental conditions, unabling us to incorporate the mitochondrial involvement in the cystinosis pathophysiology. In the revised manuscript, under discussion section, we now propose to unravel the role of mitochondria in the progression of cystinosis as a future work.

Re-Review: Authors mention that mitochondria is future work. However, since aKG is a key molecule in the Krebs cycle (this paper claim to identify aKG as an important metabolite bridging cystinosis loss) determining the overall rate of the citric acid cycle in the mitochondria, studying the mitochondria is important.

Question 3:

Membrane permeant metabolites are tricky and not a priori bona fide equivalents. In the field of cystinosis itself, membrane permeant cystine derivatives got a bad reputation as being proved toxic by themselves to mitochondria (referenced by Cherqui, *Nat Rev Nephrol*, 2017; quoted). Fortunately, control of intrinsic lack of toxicity after 24 h at 2mM DMKG in fed cystinotic cells is reinsuring. However, considering DMKG as equivalent to aKG seems far-fetched (e.g. see page 9, para 3, line 5 : "in presence of aKG" when data show "under DMKG supplementation". Moreover, short-term effects of DMKG on ROS are significantly opposite in the two cystinosis-deficient cell lines. Thus, comparison of intracellular levels of aKG and DMKG, and the effect of DMKG supplementation on actual aKG level should be quite informative and thus also strongly recommended. This clarification is all the more important since aKG effects are discussed as context-dependent.

We thank the reviewer for this excellent suggestion and agree that DMKG may not be equivalent to aKG. Therefore, in the revised manuscript, we changed aKG to DMKG wherever appropriate.

The metabolite aKG is membrane-impermeable, meaning that it is usually added to cells in the form of esters such as DMKG. Once DMKG crosses the plasma membrane, it is hydrolyzed by esterases to generate aKG, which remains trapped within cells. Baracco et al (*Aging*, 2019), using mass spectrometric metabolomics, demonstrated that addition of DMKG can indeed increase the intracellular levels of aKG in human osteosarcoma U2OS and human neuroglioma H4 cells. Similarly, Villar et al. (2017) used DMKG (2 mM) as a source of aKG in their experiments, demonstrating that it dysregulates autophagy and induces glutaminolysis-mediated apoptosis in cancer cells. Therefore, we feel confident that addition of DMKG (2 mM) can lead to increased intracellular levels of aKG in our cell model. Further, the metabolite is known for its antioxidant properties and potential for the treatment of disorders induced by oxidative stress (Liu et al, 2018; Mailloux et al, 2009; Satpute et al, 2010; Starkov, 2013). Indeed, supplementing cells with DMKG for 4 hrs led to a significant reduction in ROS levels in control cells (2.1-fold). However, this effect was modest in CTNS^{-/-} cells (1.2-fold), and in CTNS^{patient} cells DMKG rather induced ROS production. This could be due to the fact that CTNS loss hampers glutathione synthesis (Chol et al, 2004; Levtchenko et al, 2005; Wilmer, M.J., et al, 2011), and therefore less antioxidant effect of DMKG in cystinotic cell lines is observed. We now explain this in the discussion.

Of note, the metabolomic data show that CTNS^{patient} cells already have a higher intracellular aKG level

compared to CTNS^{-/-} cells, which could explain their adverse response to more DMKG. This also holds true in both cell viability and apoptosis assays where CTNS^{patient} cells were more sensitive to DMKG compared to CTNS^{-/-} cells.

Re-Review: I agree with the reviewer that it is important to show the effect of DMKG supplementation on aKG in each of these cell types used. Though the authors referred other publications to support the effect of DMKG supplementation on aKG, this experiment is still recommended as the effect of DMKG on renal cells are not known and also, we are seeing two different effect of DMKG in CTNS^{-/-} and CTNS^{patient} cell types.

Supplementing cells with DMKG for 4 hrs led to a modest but significant reduction in ROS levels in CTNS^{-/-} cells, whereas DMKG rather significantly induced ROS production in CTNS^{patient} cell types. These are significant changes and cannot be overlooked to fit in to the hypothesis.

Question 4:

Conclusion that aKG is a key metabolite should be explicitly presented with more caution (page 8, first para, last line). Besides aKG, several other metabolites were modified by cystinosis and

corrected by bicalutamide (Fig4A). Thus, while aKG was useful reporter to eventually identify bicalutamide as useful drug, drug benefits cannot be safely attributed to aKG as the "key actor" (or implicitly presented as such).

We thank the reviewer for her/his constructive suggestion and agree that the role of aKG may have been overstated. In the revised manuscript, the text now reads: 'we identified α KG as an important metabolite bridging cystinosis loss to increased oxidative stress, autophagy disruption, and proximal tubule dysfunction in cystinosis.'

Re-Review: The authors failed to address the main issue as in - why aKG is an important metabolites than others? There are other metabolites, such as hypoxanthine, which is the third highest metabolite to be affected after cystine and cysteine in both CTNA -/- and CTNS-Patient cells. Hypoxanthine is known to induce apoptosis by increasing oxidative stress and expression of the associated downstream apoptotic signaling proteins and caspase activation [PMID: 27888108; PMID: 28593435].

MINOR COMMENTS

We thank the reviewer for pointing out the following errors which we addressed in the revised manuscript.

Question 5:

A) Abstract, line 1 : monogenic

We now replaced the word monogenetic to monogenic in the revised manuscript.

B) Page 3, para 1, line 3 : "leading"; stricto sensu, Fanconi syndrome in general does not per se lead to chronic kidney failure; it certainly does so in nephropathic cystinosis (NC).

In the revised manuscript, the text now reads: 'The first clinical signs develop during infancy (age of ~6 months) in the form of renal Fanconi syndrome and over time patients develop chronic kidney disease and finally renal failure in the first or second decade of life (Gahl et al, 2002).'

C) Page 3, para 3, line 4 : it would be appropriate to already quote here the landmark paper by Festa et al, Nat Comm, 2018.

We added the reference (Festa et al, Nat Comm, 2018) in the revised manuscript.

D) Page 6, first para, last line : please clarify to which cell (Ctns-/- ?) this conclusion applies. Conversely, would the authors conclude that Ctnspatient cells are endocytosis- but not degradation-defective?

The text now reads: indicating the defect is related to degradation rather than protein uptake in CTNS-/- cells.

E) Fig legend 1B: scale bar of 13 micron is awkward.

We now readjusted the confocal micrographs to 10 μ m.

G) Fig2I. IGF2R/Cation-independent mannose 6-phosphate receptor. It is suggested (page 7, end of first para) that this decrease would cause defective lysosomal targeting of lysosomal enzymes by rerouting to the constitutive secretory pathway, thus could contribute to explain their paradoxical decrease in

steady-state cell content, despite global transcriptional activation by TFEB (see Fig 1D). However, defective lysosomal proteolysis is instead attributed to impaired activation of pro-cathepsin C (Festa et al, Nat Comm, 2018, quoted : Fig3c), whose level actually increases. Moreover, as to lysosomal acid phosphatase, it is known not to be targeted to lysosomes as a soluble cargo by a M6P anchor but rather assembled as a transmembrane protein which is eventually released into lysosomal lumen upon proteolytic release from its anchor (classical work from von Figura's lab). Please clarify and/or rewrite accordingly.

We thank the reviewer for this constructive suggestion. Indeed, Festa et al, postulated that impairment of lysosome proteolysis observed both in vitro and in vivo is as a result of defective cathepsin activation despite an efficient delivery of newly synthesized cathepsins from Golgi to lysosomes.

The text now reads: 'Consistent with the reduced cargo degradation in cystinotic cells, we found an overall reduction in lysosomal catalytic-proteins expression including, lysosomal acid lipase (LIPA), lysosomal acid phosphatase (ACP2), and cysteine proteases cathepsin (CTSS) (Fig 2I). Moreover, cation- independent mannose 6-phosphate receptor (IGF2R) was found downregulated in CTNS-/- cells (Fig 2I). IGF2R is responsible for the delivery of many newly synthesized lysosomal enzymes from Golgi to the lysosome, and its downregulation could result in decreased lysosomal activity and production of ROS (Probst et al, 2006; Takeda et al, 2019).'

H) Fig3A : for consistency, replace Ctns KO by Ctnspatient.

Thank you for your thorough reading and pointing out this error. We now replaced the CTNSKO by CTNSpatient in figure 3A of the revised manuscript.

I) Fig5A: cell death in starved (-AA) 2-mM DMKG-treated Ctns-/- cells is 35% here vs 70% at Fig3E. Please clarify.

In figure 3E, the data is presented relative to the fed condition, however, in figure 5A data is presented relative to the starved condition.

In figure 3E, addition of DMKG to starved (-AA) CTNS-/- cells resulted in additional 35% cell death compared to -AA CTNS-/- without DMKG. Since, the data is presented relative to the fed condition, it is shown that cell death in starved DMKG-treated CTNS-/- cells is 70%. In figure 5A, however, the data is presented relative to the starved condition. Hence, addition of DMKG resulted in a similar 35% cell death. For clarity, in the revised manuscript we now explain this in the figure's legends.

J) Discussion, page 14, lines 9-11: should be deleted; following sentences should also be rewritten.

(i) Cysteamine can improve, but certainly does not fully correct cystinosis manifestations and lesions in mice. (ii) The authors may be informed of a powerful rat model, with more robust and earlier phenotype, generated by three different laboratories. (iii) The zebrafish model is useful for screening yet may generate symptoms irrelevant for patients (embryonic dysmorphism; cystinotic babies are born normal)

We fully agree with reviewer 2 that the discussion on animal models could be expanded. However, most of this information, such as details on the rat and mouse models, is not publicly available which hampers a clear evaluation.

Discussion, page 14, lines 9-11 read: "Furthermore, unlike the human situation where treatment with cysteamine does not improve the renal phenotype and only improves some parts of the cystinotic phenotype (Cherqui, 2012), it can completely resolve any renal symptoms in mice." As we initially aimed to evaluate the cysteamine-bicalutamide treatment in mice, we already discussed this issue extensively with our collaborators at the Renal Diseases Research Unit at Bambino Gesù Children's Hospital (Rome). Their data shows that cysteamine can completely resolve any renal symptoms in the above mentioned mouse model. This data has not yet been published (manuscript currently under revision), but we obtained permission from the group of Prof Dr. Francesco Emma to confidentially share these results with you, ahead of print (below). Their data show that daily treatment of *Ctns*^{-/-} mice (from 2-to-14 months of age) with cysteamine (500 mg/kg body weight/day) drastically reduces cystine and urinary levels of low molecular weight protein CC16 to the levels found in the wild type mice, opposing clinical features in human.

mean SEM

mean SEM

We, therefore, considered other models that could be used to evaluate the cysteamine/bicalutamide combination therapy and were able to develop a cystinotic patient-derived kidney organoids without genetic modifications, as described earlier for healthy kidney and urine samples (Schutgens et al, 2019). This unique primary human tissue culture along with cystinotic zebrafish (Elmonem 2017; Elmonem 2018) appeared to be robust and versatile models of cystinosis with early phenotypic characteristics of the disease, which are also suitable for the in vitro and in vivo screening of novel therapeutic agents. These emerging models are now gaining interest in the field as seen by the recent publication on the beneficial effect of cysteamine combined with the mTOR inhibitor everolimus in cystinotic kidney organoids (Hollywood 2020), and the use of cystinotic zebrafish in evaluating luteolin as a potential treatment for cystinosis (De Leo 2020, JASN).

Three different *Ctns* rat models have been developed so far, however, to our knowledge only one has been published (Shimizu 2019, Mammalian Genome). This *CTNS*^{-/-} spontaneously formed in Long-Evans Agouti rat and its relevance to study the cystinosis phenotype has yet to be established. The two other models (generated using CRISPR-Cas9 in the labs of Devuyst and Hollywood) look promising but have not yet been published or shared with the academic society. We hence feel it is better at this time not to expand on unpublished models.

The zebrafish is gaining popularity for drug screening and toxicity studies as it combines multiple organ functions that cannot be recapitulated in in vitro models. Indeed, the dysmorphic features are not directly linked to clinical features in patients, but they provide a phenotypic readout that can be

quantified for analysis. We now explain this in the discussion section.

K) Supp Fig 3. The rationale for including toxicity data of other drugs such as genistein, disulfiram etc..in this particular study is unclear.

In our study, we screened different candidate drugs based on their ability to restore the metabolic profile using metabolomics. Based on this, we continued with bicalutamide as our most promising compound. To ensure that the concentrations of the drugs tested were within a non-cytotoxic range, we performed toxicity curves for all the drugs tested which are indicated in EV3. This is also explained in the result section, lines 212-215.

Re-Review: As per the toxicity curves, after Cysteamine 8-Br-cAMP has the best viability. Have the authors used this drug? If bicalutamide is chosen based on its effect on restoring the metabolic profile then showing that bicalutamide does not affect the cell viability is enough. No need to show the cell viability with other drugs, but if you do then you should also show the effect of these drugs on restoring metabolic profiles.

Referee: 3

This is an innovative investigative approach to characterize the cellular mechanisms responsible for the dysregulation of proximal tubular function in experimental cystinosis models. The authors used immortalized cell culture lines of cystinosis and control tissue, and a unique application of organ tubuloids from cystinosis patients, and finally, a zebrafish model of cystinosis. This combination of experimental approaches with advanced proteomic and metabolomic analyses, provided new and encouraging data on the specific cellular mechanisms responsible for the varying phenotypes and pathophysiology of cystinosis. Indeed, these novel approaches have totally circumvented the numerous shortcomings of the in vivo models of cystinosis where there are limited renal phenotypes.

Question 1:

One minor suggestion would be to possibly expand upon the findings to point to further development of drug interventions for testing in special populations to advance the field.

We highly appreciate the positive remarks by this reviewer. To expand on the finding and to point to further development of our drug interventions, we now included the following text in the discussion section: 'Preclinical studies in a suitable animal model should determine if cysteamine-bicalutamide combination therapy is able to improve or prevent the development of renal Fanconi syndrome and renal failure. Translation to the clinic should be further facilitated by the fact that bicalutamide is already an approved drug with a known safety profile, although a separate study should be performed to determine the appropriate dose for cystinotic patients.'

Referee #2 (Remarks for Author):

COMMENTS TO AUTHORS

The authors have done their best to address most of my previous comments, which largely overlapped with those of reviewer #1. My comments were not however exhaustive as I intended

rather to give examples so as to promote accuracy and open-mindedness towards alternative pathways/explanations so as to improve the paper. Hence the residual comments.

SPECIFIC CHANGES

Line 131-133 on very small lysosomes even in their *Ctns*^{-/-} cells. ... hampering reliable quantitation. Please add the following sentence. This contrasts with other reports demonstrating enlarged lysosomes in cystinotic cells *in vivo* (Gaide Chevronnay et al, 2014 2015) and *in vitro* (Ivanova et al, 2015; Festa et al, 2018).

Line 173 : cysteine protease (No "s") cathepsin S (CTSS). The authors might consider adding ... the "potentially representative" cysteine protease... Most of numerous lysosomal proteases are cysteine proteases, the aspartyl protease, cathepsin D, being a well-known exception. Of note, the bulk of lysosomal proteolysis is generally NOT attributed to cathepsin S (hence "potentially representative") and individual lysosomal hydrolases are generally not rate-limiting (in KO mice stains, hemizygous showing no phenotype). The reader should not conclude that the lower cargo degradation in cystinotic cells can be readily explained by the decreased abundance of cathepsin S alone. Other proteases and additional changes (such as change in lysosomal redox level, allowing opening of intramolecular disulfide bonds for further hydrolysis) are certainly key parameters.

Hence LATER :

Line 305 : "demonstrating" is incorrect here. Proteomic analyses did not demonstrate the delay of proteolysis (DQ-BSA experiments did). Replace by "which was consistent with delayed...

Line 220-4: The two following statements appear contradictory.

*Bicalutamide did not restore the high cystine and cysteine levels (thus no change) ...

*In line with its metabolic effects...(In line suggesting a related, possibly explanatory change) bicalutamide upregulated enzymes in cysteine conversion (thus a related change).

PLEASE CLARIFY

Lines 315-7. the NEW interpretation of the DMKG toxicity after 24 h specifically in their *Ctns*-deficient cells based on ROS effect is not really convincing for this reviewer. This is an important point since aKG effects are very much context-dependent. In the absence of DMKG, *Ctns*-deficient cells manage well the strong ROS production induced by starvation (and presumably other changes, possibly related to AMPK activation a.o., Fig3B-D), but benefit less (*Ctns*^{-/-}) or not (*Ctns* patient) by DMKG addition. The authors seem to conclude rightaway that anti-oxidant effect of aKG derived from DMKG is blunted in *Ctns*-deficient cells, which they attribute to hampered glutathione biosynthesis. Yet, Fig 4A (two first bars) shows that in untreated *Ctns*^{-/-} cells, glutathione level is identical to isogenic WT controls, and increases vigorously to cysteamine addition, demonstrating a clear glutathione biosynthesis potential in their system. This should be clearly stated. After which, "We also found" would better start by "However". How aKG imbalance impacts on other pathways which cause abnormal activation of autophagy should be briefly discussed at line 319.

Line 401 : "normalize" is not correct. Fig 4A shows that bicalutamide not only suppresses the (modest) increase of aKG in *Ctns*^{-/-} cells, but results into a two-fold lower level as compared to untreated WT cells.

MINOR CHANGES

Fig 2D, upper right : replace "Ctns KO" by Ctns patient (I think it was wrongly changed during revision; was correct in version 1)

Fig3A. define cells lines in italics, as everywhere else

Line 273 : replace a non-nephrotoxic "dose" by "concentration" (these are in vitro experiments ; amounts are expressed in microM at line 470)

Line 337 : aKG is a (not the) mitochondrial metabolite

Line 358 : replace including high "blood" cystine levels by "tissue" (blood cystine was not measured in the following references)

Line 388: safe in our ADD "short-term " models (as opposed to long-term toxicity of racemic bicalutamide in human patients).

***** Reviewer's comments *****

Referee: 1

Question 1:

A) I do not see any significant difference in IF (1B) between fed *CTNS*-WT & *CTNS*^{-/-}. They looked very similar to each other and different from *CTNS*-patient.

We thank the reviewer for this comment. Figure 1B (now figure EV2) shows immunofluorescent staining of lysosomal-associated membrane protein 1 (LAMP1; green) and mTOR (red) in *CTNS*^{WT}, *CTNS*^{-/-}, and *CTNS*^{Patient} cells. In control cells, mTOR respond to the nutrient status of the cells. In the presence of nutrients (standard medium, fed condition) mTOR co-localizes with the lysosomal membrane (in yellow), whereas upon starvation (-AA) mTOR was released from the lysosomes (less co-localization) and upon reintroduction of nutrients the co-localization is re-established. However, mTOR in *CTNS*^{-/-} and *CTNS*^{Patient} cells was already released from lysosomes in the fed condition and a shift based on the nutritional status of the cells was not observed. This indicates decreased mTOR activity in cystinotic cells also under fed condition. Moreover, downstream effects of mTOR deactivation such as increased TFEB nuclear translocation and LC3-II accumulation were also observed, further confirming that mTOR activity is decreased in both cystinotic cell lines. This is in accordance with other studies where they have shown the decreased retention of mTOR on the lysosomal membrane, and therefore the decreased activity of mTOR in in vitro and in vivo models of cystinosis (Andrzejewska, Z. et al. J Am Soc Nephrol, 2016; Festa et al, Nat communication, 2018; Ivanova et al, PLOsone, 2015, Ivanova et al. J Inherit Metab Dis 2016). Unfortunately, technical hurdles did not allow us to quantify the co-localization between mTOR and LAMP-1 to show dissociation of the mTOR complex in cystinosis. In our cell lines, the size of lysosomes are so small and are generally clumped together hampering reliable quantification. Therefore, we moved figure 1B to the figure EV2 and explained the technical hurdles in the result section.

Re-Rev: In this figure, LAMP1 colocalizes with mTOR in *CTNS*-WT but there is significant less co-localization in *CTNS*-patient group as interpreted by the authors. However, I agree with referee 1 that we can still see significant amount of co-localization (yellow puncta) in the *CTNS* ^{-/-} group, maybe less than the wild type but the staining pattern is more similar to the wild type than *CTNS*-patient. In other words, *CTNS* ^{-/-} group is more similar to the *CTNS*-WT than the *CTNS*-patient cells. Overall, the co-localization is high in wildtype, less in *CTNS* ^{-/-} group and almost no co-localization in *CTNS*-patient. I think this needed to be mentioned in the manuscript.

We thank the reviewer for this comment and agree that this needs to be addressed in the manuscript. The text now reads: *"In the presence of nutrients (standard medium, fed condition), mTOR located to the lysosomal membranes of CTNS^{WT} cells (Fig EV2). Upon starvation (-AA), mTOR was released from the lysosomes and re-localised upon reintroduction of nutrients. In contrast, in CTNS^{-/-} and CTNS^{Patient} cells the fed condition revealed a less pronounced colocalization, and no difference was seen between the fed and starved condition (Fig EV2). Furthermore, a perinuclear localization of lysosomes was observed for both the CTNS^{-/-} and CTNS^{Patient} cells, as reported previously for cystinotic cells (Festa et al., 2018; Ivanova et al., 2015)."*

B) No visual change was noted in the SQSTM1 protein expression (1H, J) in *CTNS* patient. Not sure how they got significant difference in expression in the bar chart. Also, they looked at total SQSTM1 protein expression, and not the active form. Therefore, this doesn't represent inactivity of mTORC1 in Cystinosis. You can relate less active SQSTM1 and LC3-II with increased autophagy & inactive mTORC1, which was not shown in this paper. Their *CTNS*^{-/-} cell line is more similar to WT than patient RPEECs.

We thank the reviewer for this constructive suggestion. We agree that the Western blot shown was not very clear regarding SQSTM1 protein expression in *CTNS*^{Patient} cells (now figure 1G, H). For the bar graphs, we combined data from three independent experiments (see blot images below). In accordance with other studies, we used total SQSTM1 protein expression to demonstrate lysosomal degradation and autophagic flux in cystinosis (Sansanwal, *Pediatr Nephrol*, 2012; Festa et al, *Nat Communication*, 2018). Increase in p62/SQSTM1 levels (in presence of the lysosomal fusion blocker bafilomycin A1) in cells indicate impaired lysosomal-cargo degradation ((Figure 1I) and not mTORC1 inactivity.

Overall, our data indicate that *CTNS* loss induces abnormal autophagy activation (demonstrated by the decreased mTOR activity, increased TFEB nuclear translocation and LC3- accumulation) and hampers lysosomal cargo degradation (demonstrated by increased p62/SQSTM1 levels, decreased processing of DQ-BSA, and overall reduction in lysosomal catalytic-proteins expression) in cystinotic cells. We now explain this in more detail under the result section "CRISPR- generated *CTNS*^{-/-} ciPTEC display increased cystine accumulation and impaired lysosomal autophagy dynamics".

Question 2:

Fig2: As shown in 2A-B, *CTNS* patient, *CTNS*-WT & *CTNS*^{-/-} have very different & distinct protein expression. Moreover, *CTNS*^{-/-} behaves similar to WT than *CTNS* patient. Not sure why for further experiments (Fig 2C, G, H, I) authors compared WT with *CTNS*^{-/-}. Rather they should have compared WT with *CTNS* patients. In figure 2D, they have compared *CTNS* patient with *CTNS*^{-/-}, which is of no use. This entire figure lacks proper exp planning. Not sure if we can conclude anything substantial from this figure.

One of the main strengths of this study is that we created isogenic cell lines by specifically knocking out *CTNS* using CRISPR technology to study the effect of the cystinosis loss-of-function. Cell lines obtained at different times and from different individuals (*i.e.* patients) often have variable phenotypes, independent of the genetic defect that is being studied. Indeed, figure 2A, B, and F shows that *CTNS*^{Patient} cells account for most of the variability in the data, suggesting that the different genetic background of the patient-derived cells affect the data more than *CTNS* loss itself. As our aim was to study the effect of *CTNS* loss, we focused on the metabolites and proteins that were significantly different between CRISPR-generated *CTNS*^{-/-} cells compared to their isogenic control cells. Further, comparing metabolites that are different in both *CTNS*^{-/-} and *CTNS*^{Patient} cells (figure 2D) allowed us to focus on the metabolites that are robustly changed as a result of *CTNS* loss in both isogenic and non-isogenic cystinotic cells, although in this last comparison more subtle cellular changes will be lost. A schematic figure indicating how *CTNS* loss and genetic background attributes to data variation is demonstrated in the figure below. This strategy is further explained in the results section on Metabolomic and proteomic profiling. Using this strategy, we identified increased levels of cystine, cysteine, and αKG in cystinotic cells. Increased cystine and cysteine levels are the real hallmark of the disease so these were expected to be found, but in addition we identified αKG as an important player. Moreover, we could further confirm this by a significantly increased level of αKG in plasma of cystinotic patients.

Re-Rev: I agree with the reviewer that *CTNS* -/- cells behave more like the wild type cells and do not mimic the cystinosis disease phenotype. In spite of the background differences all the patients grouped together in the PCA plot, which is completely separate than the *CTNS* -/-. Authors should include proper normal control to compare with cystinosis patients sample instead of using these wildtype cells as the common control for both the cell types. And wildtype should only be compared with *CTNS* -/- cells.

We thank the reviewer for this question. Note that the wild type cell line was originally created as control cell line for the patient cell lines established by Wilmer et al. (Wilmer et al. 2005; Wilmer et al. 2011) some fifteen years ago, thoroughly characterized (Wilmer et al. 2010) and used in >60 experimental studies up until now. As there appeared phenotypical heterogeneity between the various cell lines, we decided to develop the isogenic cell lines in the current study to understand the metabolic changes due to *CTNS*-deficiency solely. So, in fact, the wild type cell line is the 'proper control' as this was an age-matched cell line developed previously (see for details on matching Wilmer et al. 2011). We now realize that a clear explanation of the origin of the ciPTEC cells is lacking in our manuscript and that

explaining this in more detail will help to clarify the rationale behind our study. To address this, we included the following information in the beginning of the *results* section:

“To understand the effect of CTNS loss on proximal tubule cell function we used two well-characterized conditionally immortalized proximal tubule epithelial cell (ciPTEC) lines previously generated from urine samples of a cystinosis patient (ciPTEC CTNS^{Patient}) and an age matched healthy volunteer (ciPTEC CTNS^{WT}) (Wilmer et al, 2005; Wilmer et al., 2011; Wilmer et al, 2010). However, as these cell lines were obtained from different individuals the genetic variation is likely to introduce phenotypical changes independent of CTNS loss. To overcome this limitation, we created an isogenic CTNS-deficient cell line (ciPTEC CTNS^{-/-}) by specifically knocking out CTNS in the control ciPTEC (referred to as ciPTEC CTNS^{WT}).”

Further, we included the following information in the *Materials and Methods* section:

“The ciPTEC used in this study (ciPTEC CTNS^{WT} and ciPTEC CTNS^{Patient}) were obtained from Cell4Pharma (Nijmegen, The Netherlands, MTA #A16-0147). These cell lines have been generated previously from proximal tubule cells isolated from urine samples of a cystinotic patient and an age matched healthy control, followed by immortalization (Wilmer et al, 2005; Wilmer et al., 2011; Wilmer et al, 2010). In brief, primary cells were immortalized by transfection with SV40T ts A58 (SV40T) and hTERT (human telomerase reverse transcriptase) followed by subcloning to obtain homozygous cell populations. Of each donor, one subclone was selected based on its proximal tubular characteristics including morphology, expression pattern and transport activity. In this study we used the existing ciPTEC CTNS^{WT} cell line to generate an isogenic CTNS knockout line, named CTNS^{-/-}.”

With respect to the analysis, the aim of the metabolomics was to understand the effect of CTNS loss on proximal tubule cell metabolism. For this, we initially used ciPTEC CTNS^{WT} and the patient derived ciPTEC CTNS^{Patient} which have been used in multiple cystinosis studies (Gorvin CM et al. Proc Natl Acad Sci 2013; Ivanova et al. J Inherit Metab Dis 2016; Jamalpoor et al. Biomed Chromatogr 2018; De Leo et al. J Am Soc Nephrol 2020). In the PCA plot, the patient samples indeed grouped together because these are replicate experiments performed with the same patient cell line. It shows that the (technical) variation between different experiments is low. However, when comparing ciPTEC CTNS^{WT} and the patient derived ciPTEC CTNS^{Patient} it became clear that we cannot discriminate between CTNS loss and genetic background or clonal variation between the cell lines. We therefore generated the CTNS knockout line, using the original control cell line. This essentially rendered the ciPTEC CTNS^{Patient} cell line obsolete, but we could still use the data to identify the metabolic changes that are robust enough to be maintained across different genetic backgrounds (up or down in both ciPTEC CTNS^{Patient} and ciPTEC CTNS^{-/-}). These metabolites (indicated by the red fields in the schematic picture below) are worth investigating further. A good control for the CTNS^{Patient} cells would be to perform a CTNS gene correction of the patient cell line, which is something we address in our ongoing research but currently not yet established. However, this will only yield useful data if the damage to the cells is indeed reversible, which would also be interesting to know from a therapeutic perspective. However, for this study the CTNS knockout (ciPTEC CTNS^{-/-}) is the best way to study the role of CTNS and to have a reliable isogenic cell line.

Schematic overview of metabolomic analysis to robustly identify metabolites changed due to *CTNS* loss.

We have included this schematic representation to the appendix (**Appendix figure S2**).

Question 3:

A) Fig3: DMKG (a form of alphaKG) reduced ROS in *CTNS*-WT & *CTNS*^{-/-} & increased ROS in *CTNS* patient. However, DMKG treatment increased apoptosis in *CTNS*^{-/-} & *CTNS* patients. It shows that reduced ROS induces apoptosis in *CTNS*^{-/-}. Whereas increased ROS in *CTNS* patients also induce apoptosis. How do you explain this?

We thank the reviewer for this question. At baseline, ROS and aKG are consistently increased in both *CTNS*^{-/-} cells and *CTNS*^{patient} cells. aKG is a metabolite known for its antioxidant properties and potential for the treatment of disorders induced by oxidative stress (Liu et al, 2018; Mailloux et al, 2009; Satpute et al, 2010; Starkov, 2013). To study the short-term effect of DMKG on starvation-induced ROS (Figure 2A-D), we supplemented the cells with DMKG for 4hrs. We found that at 4 hrs, DMKG is not toxic in any of the cell lines (Expanded View figure 3A), allowing us to determine the true antioxidant effect of DMKG. In all cell lines, starvation (-AA) led to a strong increase in ROS (starvation induced stress), which could be attenuated by supplementing cells with DMKG for 4 hrs leading to a significant reduction in ROS levels in control cells (2.1-fold). However, this effect was modest in *CTNS*^{-/-} cells (1.2-fold), and in *CTNS*^{patient} cells DMKG even induced ROS production. This shows that DMKG is not as effective as antioxidant in *CTNS*^{-/-} cells and *CTNS*^{patient} as it is in control cells. This could be due to the fact that *CTNS* loss hampers glutathione synthesis (Chol et al, 2004; Levchenko et al, 2005; Wilmer, M.J., et al, 2011), and therefore less anti-oxidant effect of DMKG in cystinotic cell lines is observed.

On the other hand, Villar et al. showed that during starvation long term incubation with DMKG can induce glutaminolysis-mediated apoptosis through activation of mTOR and subsequent inhibition of autophagy (Villar et al, 2017). Indeed, DMKG treatment for 24 hrs (but not at 4 hrs) induced apoptosis activation in both *CTNS*^{-/-} & *CTNS*^{patient}. We here demonstrate that the addition of DMKG to cystinotic proximal tubule cells activates autophagy and mediates apoptosis (Figure 3E-I). Of note, control cells handled DMKG well both at 4 and 24 hrs. We now explain this in more detail in the discussion section of our revised manuscript.

Re-Review: DMKG acts as an antioxidant for both wildtype and *CTNS* ^{-/-} as it significantly reduces the ROS in both these cell types. No matter how modest the decrease of ROS is in *CTNS* ^{-/-}, it is still a

significant decrease, which cannot be ignored. On the other hand, DMKG significantly induces ROS in cystinosis-patients. The difference in pattern can't be explained. Also, when all the cells were fed, they had comparable level of ROS but when starved and treated with DMKG then the pattern in the isogenic cells are completely opposite of cystinosis-patients. This further proves that cystinosis-patients should have its own control and can't be compared to a common control. As per Villars et al, long-term exposure of DMKG inhibits autophagy and induces apoptosis, but your data shows that DMKG induces autophagy and apoptosis? Both apoptosis and autophagy are very distinct cell death/recycle process, please could you explain how DMKG is activating autophagy and mediating apoptosis at the same time by activating mTORC pathway?

We thank the reviewer for these discussion points. In the ROS analysis (Figure 3A-D) we show the ability of DMKG to modulate the “starvation induced ROS”, and to what extent DMKG is able to restore the ROS levels to normal levels (before starvation). In both *CTNS*^{-/-}, and *CTNS*^{Patient} cells this protective effect is not present or reduced. Furthermore, as the effect of DMKG is dose dependent (see Figure EV3), and the intracellular levels of αKG are already higher in the *CTNS*^{Patient} cells compared to *CTNS*^{-/-} cells (Figure 2D) it is therefore possible that after addition of DMKG toxic levels are reached sooner in the *CTNS*^{Patient} cells. For more information on the ROS and DMKG levels also see our reply to reviewer 2 (question 1E and question 3).

Indeed, Villar et al. demonstrated that starvation and long-term incubation with DMKG induced glutaminolysis-mediated apoptosis through activation of mTOR and subsequent inhibition of autophagy. We believe that under normal conditions this pathway is not active, as in our ciPTECS mTOR is inhibited and autophagy is activated. However, we hypothesize that the already increased intracellular levels of αKG in cystinotic ciPTEC (Figure 2D) combined with delayed cargo degradation (Figure 1I) could trigger the glutaminolysis-induced apoptosis. Our data (Fig. 5A) also shows that bicalutamide (but not cysteamine), is able to prevent the DMKG induced apoptosis, consistent with the ability of bicalutamide to upregulate the autophagic flux (Hao et al. 2017) and preventing glutaminolysis-mediated apoptosis.

To address this, we added the following to the discussion:

“In our cells, addition of bicalutamide but not cysteamine could prevent the DMKG induced cell death and reduced LC3-II/LC3-I levels. As cysteamine alone is able to efficiently induce glutathione synthesis in cystinotic cells, it is likely that the DMKG decreased cell viability was not due to a reduced availability of antioxidants. The marked increase in LC3-II/LC3-I levels in CTNS deficient cells indicates activation of autophagy. However, the reduced protein degradation (shown here by DQ-BSA analysis and in previously published reports) (Festa et al., 2018; Ivanova et al., 2015) question the efficiency of this autophagic process. The high intracellular levels of αKG together with a defective autophagy may therefore induce glutaminolysis-mediated apoptosis during starvation specifically in cystinotic cells. As bicalutamide is known to upregulate the autophagic flux (Hao et al, 2017), the addition of bicalutamide could restore the autophagic flux in cystinotic cells and prevent this glutaminolysis-mediated apoptosis.”

B) No significant difference between LC3-II expression in WT-ctrl and patient-ctrl (3I). It is not clear why DMKG reduced LC3-II expression in WT and patient, and not in *CTNS*^{-/-}.

We indeed demonstrated that at baseline there is no difference in LC3-II expression (Figures 1G, H). However, after inhibiting the autophagic flux (in presence of the lysosomal fusion blocker bafilomycin A1) it becomes clear that LC3-II levels are increased in both *CTNS*^{-/-} and *CTNS*^{Patient} cells. Stable LC3-II levels in steady state conditions can be explained by the fact that the autophagic flux is very efficient in *CTNS*^{-/-} and *CTNS*^{Patient} cells, therefore hampering reliable quantification of LC3-II without bafilomycin

(Ivanova et al. J Inherit Metab Dis, 2016). Furthermore, in Figure 3I, we show that treatment with DMKG induces autophagy only in cystinotic cells, demonstrated by increased LC3-II accumulation. DMKG had no effect in control cells hence did not increase the LC3-II accumulation.

Re-Review: On the contrary to what the authors mention, visually (Fig. 3H) all the 3 cell types have significant increase in LC3-II with BafA1 under starvation, and not only *CTNS*^{-/-} and *CTNS*-Patient cells. I agree with the authors on their take on DMKG treatment and LC3-II induction.

We thank the reviewer for this question. Indeed, as expected, blocking the lysosomal fusion with bafilomycin showed a significant increase in LC3-II levels in all cell types, and this is a method to evaluate LC3-II levels in absence of autophagic flux (Yoshi et al. 2017). Therefore, the levels of LC3-II should be compared between the different cells only after addition of bafilomycin, which shows that the LC3-II increase was significantly higher in *CTNS*^{-/-} and in *CTNS*^{Patient} cells compared to *CTNS*^{WT} cells (Fig 1D-H, and Fig 3I).

To clarify this in the results section we have included the following sentence:

'Although levels of LC3-II and p62/SQSTM1 are similar at baseline, blocking the lysosomal fusion with bafilomycin showed a significant increase in both markers in *CTNS*^{-/-} and *CTNS*^{Patient} cells compared to *CTNS*^{WT} cells (Fig 1D-H), indicating increased autophagic flux (Yoshii & Mizushima, 2017). .

C) The cartoon in 3J is too far stretched.

We improved figure 3J (now Figure 7) by focusing on the findings related to *CTNS* loss and the role of compounds under different conditions.

The blue arrows indicate wild type situation, red arrows/text indicate the changes in cystinotic cells and green arrows indicate intervention with medication. DMKG; dimethyl α -ketoglutarate, ROS; Reactive oxygen species.

Re-Review: Based on the data, I do not think it is fair to combine *CTNS* $-/-$ and Cystinosis-patient as cystinotic cells. Since, both these cell types are unique and have very less commonality at the molecular level.

We thank the reviewer for this comment, but we believe we now have addressed this point. Please see our response on the comments concerning cell lines and data analysis reflected by questions 2 and 4.

Question 4:

The authors have done many experiments, but no solid conclusion can be derived based on the way the data is presented. This requires extensive focused editing.

We thank the reviewer for the helpful comments. Having adapted our manuscript according to all comments provided, we believe we significantly improved our manuscript.

Referee: 2

MAJOR COMMENTS

Question 1: Data documentation is not optimal.

A) In general, bar histograms are now discouraged and should be replaced by dotplots (as actually used at 1KL; 5E). It is also uncertain, if not dubious, that distributions of continuous variables are normal, thus systematic use of parametric statistical tests for low numbers is highly questionable. Indeed, Fig2E compares plasma aKG concentration in 4 healthy vs 6 patients where boxer-whisker plots (thus not mean \pm SEM as implied from legend) clearly suggest non-normal distribution in patients. Do non-parametric tests confirm significance? At Fig2D, no statistical analysis is presented: is the modest (1.5 fold) increase of the focused metabolite, aKG, in *CTNS*^{-/-} cells statistically significant? Incidentally, broken ordinate combined with change of increment prevents one to appreciate extent of higher increase aKG in *CTNS* patient.

We thank the reviewer for this constructive suggestion. In the revised manuscript, we now replaced bar histograms by dot plots wherever this would improve clarity. In Figure 2E, we indeed used a non-parametric Mann-Whitney test to confirm the significance of the data. However, this was not indicated in our manuscript. We now included the explanation in the legend to the figure to avoid the confusion. Moreover, we also changed the boxer-whisker plots into dot plots of the data and presented as mean \pm SEM.

Figure 2D demonstrates all metabolites that are shared and significantly changed in *CTNS*^{-/-} and *CTNS*^{Patient} compared to control cells. We now clearly indicate this in the legend to figure 2 of the revised manuscript. We changed the ordinate to better represent the data and also indicated the statistical analysis in the figure.

B) Numbers of samples at Fig 2E seem really too low for such a crucial piece of information (in fact the only one directly relevant to patients). Larger sample collection of patients and matched volunteers should not be a serious problem.

We thank the reviewer for this constructive suggestion. A larger patient population would be advisable to confirm these preliminary findings. However, approval of an updated study protocol which replaces the older study protocol under which previous samples have been taken, by the Ethical Committee has been significantly delayed due to the Covid-19 crisis. Therefore, unfortunately, we are not able to provide additional data at this time.

C) Quality of imaging is not optimal and morphology seems "lightly" interpreted, as if satisfied by pathway confirmation rather than taking into account structural implications. For example, Fig1B is not convincing as "representative": in WT fed cells (intended to validate the assay of co-localization between mTOR and LAMP-1), whereas yellow objects appear indeed to dominate in the cell with selected area, this is not the case in the cell just above; conversely, in the WT starved (-AA) selected cell to document full dissociation, several objects above the nucleus would appear doubly labeled; to the opposite, whereas the fed *CTNS*^{Patient} cell shows clear dissociation as expected, fed *CTNS*^{-/-} cells show a multiplicity of yellow spots that would instead suggest unexpected preserved association of mTOR with Lamp-1. Linescans could help. Moreover, a constant finding in cultured cystinotic cells is lysosomal enlargement with perinuclear clustering (Ivanova et al, 2015, quoted; Festa et al, Nat Comm, 2018, quoted, see beautiful Fig1A). This information should be immediately accessible by LAMP1 immunolabeling as shown

at Fig1B, but is regrettably not commented upon. Is this feature reproduced and corrected by bicalutamide?

We thank the reviewer for this comment. Figure 1B (now figure EV2) shows immunofluorescent staining of lysosomal-associated membrane protein 1 (LAMP1; green) and mTOR (Red) in $CTNS^{WT}$, $CTNS^{-/-}$, and $CTNS^{Patient}$ cells. We used different quantification methods and software, but unfortunately were not able to reliably quantify the co-localization between mTOR and LAMP-1 to show the dissociation of the mTOR complex in cystinotic cells. In our cell lines, the size of lysosomes are so small and are generally clumped together hampering reliable analysis of lysosomal perinuclear clustering or quantification of mTOR and LAMP-1 co-localization, hence, provide less reliable readout to assess the effect of potential treatments like bicalutamide. Of note, the downstream effects of mTOR deactivation such as increased TFEB nuclear translocation and LC3-II accumulation can suggest that mTOR activity is decreased in both cystinotic cell lines. To avoid suboptimal interpretation of the data, we therefore moved Fig 1B to the Fig EV2 and explained the technical hurdles in the result section.

Re-Review: The question is do we even need this figure? The authors interpretation of this figure is misleading. I agree with Referee 1 & 2 that there isn't any visual difference between Fed & -AA in wild types. Coexpression pattern in $CTNS^{-/-}$ is similar to the wild type and not even close enough to the $CTNS$ -Patient pattern of co-expression. The fed $CTNS$ -patient cell are the only type that shows clear dissociation as expected.

We thank the reviewer for this comment. In the revised manuscript, the text now reads:

“In the presence of nutrients (standard medium, fed condition), mTOR located to the lysosomal membranes of $CTNS^{WT}$ cells (Fig EV2). Upon starvation (-AA), mTOR was released from the lysosomes and re-localised upon reintroduction of nutrients. In contrast, in $CTNS^{-/-}$ and $CTNS^{Patient}$ cells the fed condition revealed a less pronounced colocalization, and no difference was seen between the fed and starved condition (Fig EV2). Furthermore, a perinuclear localization of lysosomes was observed for both the $CTNS^{-/-}$ and $CTNS^{Patient}$ cells, as reported previously for cystinotic cells (Festa et al., 2018; Ivanova et al., 2015).”

We still would like to include the figure as similar images have been used in other manuscripts on this topic and it provides a reference to readers in this field. Furthermore, the expert eye will also be able to note the perinuclear localisation of the lysosomes in both the $CTNS^{-/-}$ cells and $CTNS^{Patient}$ cells, in line with the reported cystinotic phenotype.

D) Careful manuscript revision is still needed. For example, legend of Fig 1C indicates three types of cells, yet only two images are shown. Its quantitation refers to fold-change (from 1 in WT to 2.5), yet image shows at left (WT?) no TFEB nuclear signal (accordingly, baseline should be 0, not 1 ?). Larger fields should be presented.

We thank the reviewer for this observation. Figure 1C (now figure 1B) demonstrates only TFEB cellular distribution irrespective of the cell lines used. In the revised manuscript, we now more clearly indicate this in the legend and also provided a representative confocal micrograph with a larger field demonstrating TFEB cellular distribution. Figure 1D (now figure 1C) demonstrates the quantification of TFEB-GFP nuclear translocation in $CTNS^{WT}$, $CTNS^{-/-}$, and $CTNS^{Patient}$ cells. We first determined the ratio between number of the cells with nucleus- TFEB positive over the total number of TFEB-transfected cells. The ratios were then presented as fold change compared to control cells. The analysis was carried out blindly by analyzing over 300 cells per cell line. This explanation is now included in the legend to figure 1C.

Re-Review: If figure 1 B is not specific to any cell types mentioned in this paper then I do not think it is necessary at all. If the authors really want to include this IF figure then they might as well include each of the cell types highlighting the TFEB localization change in each of them.

We thank the reviewer for this comment. In the revised manuscript, we removed the representative confocal micrographs of transcription factor EB (TFEB) cellular distribution.

E) This reviewer was not impressed by the complex cartoon presenting the working model at Fig 3J with multiple effects. It is suggested to both simplify to core effects (sequence of steps of the TCA cycle, available in any biochemistry textbook, can be omitted) and to present a sequence of situations : (i) untreated cystinosis; (ii) aKG manipulation by DMKG; (iii) treatment by cysteamine alone; (iv) treatment by bicalutamide ; (v) combined treatments.

We thank the reviewer for this suggestion and redesigned the working model in the revised manuscript.

Re-Review: The authors have simplified the representation of the study design but it still does not convey the entire story. They have not included the antioxidant effect of DMKG on *CTNS*^{-/-} cell types, which was opposite in *CTNS*-Patient cell types (which they have included).

We thank the reviewer for this question. In the overview figure (Fig. 7) at the end of the manuscript and below, we highlight the main findings related to *CTNS* loss shared between *CTNS*^{-/-}, and *CTNS*^{Patient} cells, and the effect of DMKG or bicalutamide treatments. To summarize:

- In all cells, starvation leads to ROS. In wild type cells this ROS is efficiently reduced by addition of DMKG: in both *CTNS*^{-/-}, and *CTNS*^{Patient} cells this protective effect is not present (*CTNS*^{Patient}) or reduced (*CTNS*^{-/-}).
- Addition of DMKG leads to an increase in LC3-II/LC3-I and apoptosis in *CTNS* null cells (both *CTNS*^{-/-} and *CTNS*^{Patient}), which can be prevented by the addition of bicalutamide.
- *CTNS* loss leads to accumulation of cystine, which can be reduced to wild type levels by combination treatment of bicalutamide and cysteamine.

In the ROS analysis (Figure 3A-D), we only show the ability of DMKG to modulate the “starvation induced ROS”, and to what extent DMKG is able to restore the ROS levels to normal levels (before starvation). In both *CTNS*^{-/-}, and *CTNS*^{Patient} cells this protective effect is not present or reduced. However, this experiment cannot be used to draw a conclusion on the antioxidant effect of DMKG on ROS under normal (fed) conditions when ROS levels are also much lower. In Figure 7, we therefore did not connect DMKG directly to ROS but instead focused on the ROS induction through starvation.

Figure 7. Working model summarizing the results obtained in this work. The blue arrows indicate wild type situation, red arrows/text indicate the changes in cystinotic cells and green arrows indicate intervention with medication. DMKG; dimethyl α -ketoglutarate, ROS; Reactive oxygen species.

Question 2:

Lack of mitochondrion imaging. Considering that ROS production in cystinotic cells (e.g. Festa et al, 2018) and targeted TCA cycle both occur in mitochondria, it is regretted that mitochondrion imaging is not presented, at least for the key points of the demonstration. There are excellent easy tool for discriminating ROS accumulation in mitochondria vs lysosomes (e.g. Denamur et al, Free Radic Biol Med, 2011, Fig3) and immunolabeling can readily evidence distorted (or normalized) mitochondrial shape. Such a subcellular structural study would not be that time-consuming, while mechanistically strengthening the biochemical/cell based demonstration, thus is strongly recommended.

We thank the reviewer for this constructive suggestion. To demonstrate mitochondrial involvement, we used MitoSox and CellRox reagents to discriminate ROS accumulation in mitochondria and lysosomes upon starvation in the presences and absence of DMKG, respectively. Unfortunately, we were not able to detect any MitoSox fluorescence intensity in our control or cystinotic cells under the desired experimental conditions, unabling us to incorporate the mitochondrial involvement in the cystinosis pathophysiology. In the revised manuscript, under the discussion section, we now propose to unravel the role of mitochondria in the progression of cystinosis as a future work.

Re-Review: Authors mention that mitochondria is future work. However, since aKG is a key molecule in the Krebs cycle (this paper claim to identify aKG as an important metabolite bridging cystinosis loss) determining the overall rate of the citric acid cycle in the mitochondria, studying the mitochondria is important.

We thank the reviewer for this critical remark and agree with this. A better understanding of the role of mitochondria in cystinosis is an exciting field of research. A recent report showed that mitochondrial function and dynamics were affected in cystinotic proximal tubular cells (De Rasmio et al., Int J Mol Sci, 2020), and another study found that defective autophagy-mediated clearance of damaged mitochondria was resulting in increased ROS in their cells (Festa et al, Nat Com, 2018). Mitochondrial function seems also a common feature in lysosomal storage diseases as a recent review on lysosomal storage diseases pointed out: *“It is now clear that **substrate accumulation** triggers complex pathogenetic cascades that are responsible for disease pathology, such as aberrant vesicle trafficking, impairment of autophagy, dysregulation of signaling pathways, abnormalities of calcium homeostasis, and mitochondrial dysfunction.”* (Parenti et al., EMBO Mol Med, 2021). So we believe there is a clear role for mitochondria in cystinosis, which may be related to either their abnormal shape and function, defective autophagy or could be a secondary effect of substrate accumulation in the lysosomes. In this complex network in which mitochondria/autophagy/lysosomal function we would like to challenge these groups to look into the role of aKG.

However, when looking at our metabolomics data, aKG was the only metabolite from the TCA cycle that was significantly changed (we also evaluated cis-aconitate, succinic acid, fumarate, malate and citrate). As there are multiple pathways in the cell that can generate and catabolize aKG, we can not be sure which pathway is primarily responsible for the increase in aKG. We also used EM to visualize the mitochondria in our cells but there were no clear morphological abnormalities (data not shown).

We have included the following section in the discussion of the manuscript:

“Several reports also indicate a role for mitochondria in cystinosis disease pathology, including affected mitochondrial function and dynamics (De Rasmio et al, 2019), as well as reduced autophagy-mediated clearance of damaged mitochondria resulting in increased ROS in their cells (Festa et al.,

2018; Sansanwal et al., 2010). This is also in line with the decreased expression of AKGDH in our *CTNS*^{-/-} cells, which can then lead to the observed increase in αKG. However, other TCA metabolites that we evaluated (cis-aconitate, succinic acid, fumarate, malate and citrate) remained unchanged. As there are multiple pathways in the cell that can either generate or catabolize αKG, it is also possible that another pathway is responsible for the increase in αKG. Studying the role of mitochondria in the progression of cystinosis remains important to further expand our knowledge on the complexity of the disease and prioritize drug targets for cystinosis."

Question 3:

Membrane permeant metabolites are tricky and not a priori bona fide equivalents. In the field of cystinosis itself, membrane permeant cystine derivatives got a bad reputation as being proved toxic by themselves to mitochondria (referenced by Cherqui, Nat Rev Nephrol, 2017; quoted). Fortunately, control of intrinsic lack of toxicity after 24 h at 2mM DMKG in fed cystinotic cells is reinsuring. However, considering DMKG as equivalent to αKG seems far-fetched (e.g. see page 9, para 3, line 5 : "in presence of αKG" when data show "under DMKG supplementation". Moreover, short-term effects of DMKG on ROS are significantly opposite in the two cystinosin-deficient cell lines. Thus, comparison of intracellular levels of αKG and DMKG, and the effect of DMKG supplementation on actual αKG level should be quite informative and thus also strongly recommended. This clarification is all the more important since αKG effects are discussed as context-dependent.

We thank the reviewer for this excellent suggestion and agree that DMKG may not be equivalent to αKG. Therefore, in the revised manuscript, we changed αKG to DMKG wherever appropriate. The metabolite αKG is membrane-impermeable, meaning that it is usually added to cells in the form of esters such as DMKG. Once DMKG crosses the plasma membrane, it is hydrolyzed by esterases to generate αKG, which remains trapped within cells. Baracco et al (Aging, 2019), using mass spectrometric metabolomics, demonstrated that addition of DMKG can indeed increase the intracellular levels of αKG in human osteosarcoma U2OS and human neuroglioma H4 cells. Similarly, Villar et al. (2017) used DMKG (2 mM) as a source of αKG in their experiments, demonstrating that it dysregulates autophagy and induces glutaminolysis-mediated apoptosis in cancer cells. Therefore, we feel confident that addition of DMKG (2 mM) can lead to increased intracellular levels of αKG in our cell model. Further, the metabolite is known for its antioxidant properties and potential for the treatment of disorders induced by oxidative stress (Liu et al, 2018; Mailloux et al, 2009; Satpute et al, 2010; Starkov, 2013). Indeed, supplementing cells with DMKG for 4 hrs led to a significant reduction in ROS levels in control cells (2.1-fold). However, this effect was modest in *CTNS*^{-/-} cells (1.2-fold), and in *CTNS*^{Patient} cells DMKG rather induced ROS production. This could be due to the fact that *CTNS* loss hampers glutathione synthesis (Chol et al, 2004; Levtchenko et al, 2005; Wilmer, M.J., et al, 2011), and therefore less antioxidant effect of DMKG in cystinotic cell lines is observed. We now explain this in the discussion. Of note, the metabolomic data show that *CTNS*^{Patient} cells already have a higher intracellular αKG level compared to *CTNS*^{-/-} cells, which could explain their adverse response to more DMKG. This also holds true in both cell viability and apoptosis assays where *CTNS*^{Patient} cells were more sensitive to DMKG compared to *CTNS*^{-/-} cells.

Re-Review: I agree with the reviewer that it is important to show the effect of DMKG supplementation on αKG in each of these cell types used. Though the authors referred other publications to support the effect of DMKG supplementation on αKG, this experiment is still recommended as the effect of DMKG on renal cells are not known and also, we are seeing two different effect of DMKG in *CTNS*^{-/-} and *CTNS*^{Patient} cell types. Supplementing cells with DMKG for 4 hrs led to a modest but significant reduction

in ROS levels in $CTNS^{-/-}$ cells, whereas DMKG rather significantly induced ROS production in $CTNS$ -Patient cell types. These are significant changes and cannot be overlooked to fit in to the hypothesis.

We thank the reviewer for this comment and evaluated the DMKG supplementation on aKG, as suggested (see graph below). In the ROS analysis (Figure 3A-D), we showed the ability of DMKG to modulate the "starvation induced ROS", and to what extent DMKG is able to restore the ROS levels to normal levels (before starvation). In both $CTNS^{-/-}$, and $CTNS^{Patient}$ cells this protective effect is not present ($CTNS^{Patient}$) or reduced ($CTNS^{-/-}$; see also our answer to question 1E). Furthermore, as the effect of DMKG is dose-dependent (see Figure EV3), and the intracellular levels of aKG are already higher in the $CTNS^{Patient}$ cells compared to $CTNS^{-/-}$ cells (Figure 2D) it is possible that after addition of DMKG toxic levels are reached sooner in the $CTNS^{Patient}$ cells.

Question 4:

Conclusion that aKG is a key metabolite should be explicitly presented with more caution (page 8, first para, last line). Besides aKG, several other metabolites were modified by cystinosis and corrected by bicalutamide (Fig4A). Thus, while aKG was useful reporter to eventually identify bicalutamide as useful drug, drug benefits cannot be safely attributed to aKG as the "key actor" (or implicitly presented as such).

We thank the reviewer for her/his constructive suggestion and agree that the role of aKG may have been overstated. In the revised manuscript, the text now reads: ***'we identified aKG as an important metabolite bridging cystinosis loss to increased oxidative stress, autophagy disruption, and proximal tubule dysfunction in cystinosis.'***

Re-Review: The authors failed to address the main issue as in - why aKG is an important metabolites than others? There are other metabolites, such as hypoxanthine, which is the third highest metabolite to be affected after cystine and cysteine in both $CTNA^{-/-}$ and $CTNS$ -Patient cells. Hypoxanthine is known to induce apoptosis by increasing oxidative stress and expression of the associated downstream apoptotic signaling proteins and caspase activation [PMID: 27888108; PMID: 28593435].

We thank the reviewer for this comment and agree that other metabolites are affected that may be of interest to cystinosis pathology (maybe also ones that were not included our metabolic analysis). The high levels of hypoxanthine are indeed also characteristic of Lesch-Nyhan Disease, an inherited X-linked recessive genetic disorder caused by a deficiency of the enzyme hypoxanthine-guanine

phosphoribosyltransferase (HPRT1) leading to increased levels of hypoxanthine and uric acid in cerebrospinal fluid. Furthermore, hypoxanthine levels in plasma can be increased due to exogenous factors such as smoking, alcohol drinking and hemodialysis, and have been linked to increased cholesterol accumulation and atherosclerosis in polipoprotein E-deficient mice (Hye-Myung Ryu et al. 2016). Injecting hypoxanthine in the brain has shown to affect mitochondrial function and lead to decreased ATP levels in young rats (Helena Biasibetti-Brendler et al. 2018). On the other hand, dietary supplementation of hypoxanthine was able to improve barrier function and wound healing and was associated with increased intracellular ATP and cytoskeletal capability in colon tissue (Scott Lee et al. 2018). At this time it is unclear if hypoxanthine is contributing to cystinosis disease pathology, but our proteomic data, unlike the situation in Lesch-Nyhan Disease, indicates HPRT1 enzyme levels are increased. Although the (signaling) role of hypoxanthine is not well understood, it will be an interesting candidate to analyze further in the future.

In the revised manuscript, the text now reads: ***“Using an omics-based strategy, we identified several proteins and metabolites to be consistently altered across the proximal tubule cell models used. Increased levels of α KG were particularly of interest as this metabolite is involved in autophagy regulation, oxidative stress and apoptosis, which are known to be dysregulated in cystinosis.”***

MINOR COMMENTS

We thank the reviewer for pointing out the following errors which we addressed in the revised manuscript.

Question 5:

A) Abstract, line 1 : monogenic

We now replaced the word monogenetic to monogenic in the revised manuscript.

B) Page 3, para 1, line 3 : "leading"; stricto sensu, Fanconi syndrome in general does not per se lead to chronic kidney failure; it certainly does so in nephropathic cystinosis (NC).

In the revised manuscript, the text now reads: 'The first clinical signs develop during infancy (age of ~6 months) in the form of renal Fanconi syndrome and over time patients develop chronic kidney disease and finally renal failure in the first or second decade of life (Gahl et al, 2002).'

C) Page 3, para 3, line 4 : it would be appropriate to already quote here the landmark paper by Festa et al, Nat Comm, 2018.

We added the reference (Festa et al, Nat Comm, 2018) in the revised manuscript.

D) Page 6, first para, last line : please clarify to which cell ($CTNS^{-/-}$?) this conclusion applies. Conversely, would the authors conclude that $CTNS^{patient}$ cells are endocytosis- but not degradation-defective? The text now reads: indicating the defect is related to degradation rather than protein uptake in $CTNS^{-/-}$ cells.

E) Fig legend 1B: scale bar of 13 micron is awkward.

We now readjusted the confocal micrographs to 10 μ m.

G) Fig2I. IGF2R/Cation-independent mannose 6-phosphate receptor. It is suggested (page 7, end of first para) that this decrease would cause defective lysosomal targeting of lysosomal enzymes by rerouting to the constitutive secretory pathway, thus could contribute to explain their paradoxical decrease in steady-state cell content, despite global transcriptional activation by TFEB (see Fig 1D). However, defective lysosomal proteolysis is instead attributed to impaired activation of pro-cathepsin C (Festa et al, Nat Comm, 2018, quoted : Fig3c), whose level actually increases. Moreover, as to lysosomal acid phosphatase, it is known not to be targeted to lysosomes as a soluble cargo by a M6P anchor but rather assembled as a transmembrane protein which is eventually released into lysosomal lumen upon proteolytic release from its anchor (classical work from von Figura's lab). Please clarify and/or rewrite accordingly.

We thank the reviewer for this constructive suggestion. Indeed, Festa et al, postulated that impairment of lysosome proteolysis observed both in vitro and in vivo is as a result of defective cathepsin activation despite an efficient delivery of newly synthesized cathepsins from Golgi to lysosomes. The text now reads: 'Consistent with the reduced cargo degradation in cystinotic cells, we found an overall reduction in lysosomal catalytic-proteins expression including, lysosomal acid lipase (LIPA), lysosomal acid phosphatase (ACP2), and cysteine proteases cathepsin (CTSS) (Fig 2I). Moreover, cation-independent mannose 6-phosphate receptor (IGF2R) was found downregulated in *CTNS*^{-/-} cells (Fig 2I). IGF2R is responsible for the delivery of many newly synthesized lysosomal enzymes from Golgi to the lysosome, and its downregulation could result in decreased lysosomal activity and production of ROS (Probst et al, 2006; Takeda et al, 2019).'

H) Fig3A : for consistency, replace *CTNS*^{KO} by *CTNS*^{Patient}.

Thank you for your thorough reading and pointing out this error. We now replaced the *CTNS*^{KO} by *CTNS*^{Patient} in figure 3A of the revised manuscript.

I) Fig5A: cell death in starved (-AA) 2-mM DMKG-treated *CTNS*^{-/-} cells is 35% here vs 70% at Fig3E. Please clarify.

In figure 3E, the data is presented relative to the fed condition, however, in figure 5A data is presented relative to the starved condition. In figure 3E, addition of DMKG to starved (-AA) *CTNS*^{-/-} cells resulted in additional 35% cell death compared to -AA *CTNS*^{-/-} without DMKG. Since, the data is presented relative to the fed condition, it is shown that cell death in starved DMKG-treated *CTNS*^{-/-} cells is 70%. In figure 5A, however, the data is presented relative to the starved condition. Hence, addition of DMKG resulted in a similar 35% cell death. For clarity, in the revised manuscript we now explain this in the figure's legends.

J) Discussion, page 14, lines 9-11: should be deleted; following sentences should also be rewritten. (i) Cysteamine can improve, but certainly does not fully correct cystinosis manifestations and lesions in mice. (ii) The authors may be informed of a powerful rat model, with more robust and earlier phenotype, generated by three different laboratories. (iii) The zebrafish model is useful for screening yet may generate symptoms irrelevant for patients (embryonic dysmorphism; cystinotic babies are born normal)

We fully agree with reviewer 2 that the discussion on animal models could be expanded. However, most of this information, such as details on the rat and mouse models, is not publicly available which hampers a clear evaluation. Discussion, page 14, lines 9-11 read: "Furthermore, unlike the human situation where

treatment with cysteamine does not improve the renal phenotype and only improves some parts of the cystinotic phenotype (Cherqui, 2012), it can completely resolve any renal symptoms in mice." As we initially aimed to evaluate the cysteamine-bicalutamide treatment in mice, we already discussed this issue extensively with our collaborators at the Renal Diseases Research Unit at Bambino Gesù Children's Hospital (Rome). Their data shows that cysteamine can completely resolve any renal symptoms in the above mentioned mouse model. This data has not yet been published (manuscript currently under revision), but we obtained permission from the group of Prof Dr. Francesco Emma to confidentially share these results with you, ahead of print (below). Their data show that daily treatment of *CTNS*^{-/-} mice (from 2-to-14 months of age) with cysteamine (500 mg/kg body weight/day) drastically reduces cystine and urinary levels of low molecular weight protein CC16 to the levels found in the wild type mice, opposing clinical features in human. We, therefore, considered other models that could be used to evaluate the cysteamine/bicalutamide combination therapy and were able to develop a cystinotic patient-derived kidney organoids without genetic modifications, as described earlier for healthy kidney and urine samples (Schutgens et al, 2019). This unique primary human tissue culture along with cystinotic zebrafish (Elmonem 2017; Elmonem 2018) appeared to be robust and versatile models of cystinosis with early phenotypic characteristics of the disease, which are also suitable for the in vitro and in vivo screening of novel therapeutic agents. These emerging models are now gaining interest in the field as seen by the recent publication on the beneficial effect of cysteamine combined with the mTOR inhibitor everolimus in cystinotic kidney organoids (Holiwood 2020), and the use of cystinotic zebrafish in evaluating luteolin as a potential treatment for cystinosis (De Leo 2020, JASN).

Three different *CTNS* rat models have been developed so far, however, to our knowledge only one has been published (Shimizu 2019, Mammalian Genome). This *CTNS*^{-/-} spontaneously formed in Long-Evans Agouti rat and its relevance to study the cystinosis phenotype has yet to be established. The two other models (generated using CRISPR-Cas9 in the labs of Devuyst and Hollywood) look promising but have not yet been published or shared with the academic society. We hence feel it is better at this time not to expand on unpublished models.

The zebrafish is gaining popularity for drug screening and toxicity studies as it combines multiple organ functions that cannot be recapitulated in in vitro models. Indeed, the dysmorphic features are not directly linked to clinical features in patients, but they provide a phenotypic readout that can be quantified for analysis. We now explain this in the discussion section.

K) Supp Fig 3. The rationale for including toxicity data of other drugs such as genistein, disulfiram etc. in this particular study is unclear.

In our study, we screened different candidate drugs based on their ability to restore the metabolic profile using metabolomics. Based on this, we continued with bicalutamide as our most promising compound. To ensure that the concentrations of the drugs tested were within a non-cytotoxic range, we performed toxicity curves for all the drugs tested which are indicated in EV3. This is also explained in the result section, lines 212-215.

Re-Review: As per the toxicity curves, after Cysteamine 8-Br-cAMP has the best viability. Have the authors used this drug? If bicalutamide is chosen based on its effect on restoring the metabolic profile then showing that bicalutamide does not affect the cell viability is enough. No need to show the cell viability with other drugs, but if you do then you should also show the effect of these drugs on restoring metabolic profiles.

Indeed we have also included the data on the effect of this drug on the metabolic profiles (in Dataset EV1 and online repository of the metabolic data). Here you can see the effect of cysteamine (100 μ M), disulfiram (18 μ M), genestein (10 μ M), bicalutamide (35 μ M), NAC (N-acetyl cysteine) (1.8 mM), luteolin (35 μ M) and 8-Br-cAMP (100 μ M). People are free to use this data for future studies.

Referee #2 (Remarks for Author):

The authors have done their best to address most of my previous comments, which largely overlapped with those of reviewer #1. My comments were not however exhaustive as I intended rather to give examples so as promote accuracy and open-mind towards alternative pathways/explanations so as to improve the paper. Hence the residual comments.

We would like to thank reviewer 2 for these helpful suggestions and adjusted the manuscript as detailed below.

SPECIFIC CHANGES

Line 131-133 on very small lysosomes even in their *CTNS*^{-/-} cells...hampering reliable quantitation. Please add the following sentence. This contrasts with other reports demonstrating enlarged lysosomes in cystinotic cells in vivo (Gaide Chevrionnay et al, 2014 2015) and in vitro (Ivanova et al, 2015; Festa et al, 2018).

We thank the reviewer for pointing this out. However, we could not use the immunofluorescent analysis (used for evaluating colocalization of LAMP with mTOR) to draw a conclusion on the lysosome size of cystinotic versus wild type cells. The inability to quantify the images is most likely a result of the culture conditions and cell-type used. Our maturation protocol of 7 days leads to a strong monolayer but also thickening of the cells. Even when using a high resolution confocal microscope one z-stack can hold multiple levels of lysosomes, and lysosomes can not be separated by the software quantifying the colocalization. We consulted the group of prof Klumperman, who are expert in the field of lysosomal trafficking and imaging, but they also experienced difficulties when performing staining's and analyses on our cells. It is a common problem for this type of analysis and also the paper of Ivanova 2015 reports for some cell lines they used that "*accurate measurement of the lysosomal size was not feasible due to high level of clusterization of endosomal vesicles*". Furthermore, we did make EM staining's to analyze our cells (wild type and *CTNS*^{-/-} under normal culture conditions, data not shown) but did not find a significant change in lysosome size.

In the revised manuscript, the text now reads: "*Accurate measurement of the lysosomal size and quantifying the colocalization with mTOR was not feasible due to high level of clusterization of endosomal vesicles.*"

Line 173 : cysteine protease (No "s") cathepsin S (CTSS). The authors might consider adding ... the "potentially representative" cysteine protease... Most of numerous lysosomal proteases are cysteine proteases, the aspartyl protease, cathepsin D, being a well-known exception. Of note, the bulk of lysosomal proteolysis is generally NOT attributed to cathepsin S (hence "potentially representative") and individual lysosomal hydrolases are generally not for rate-limiting (in KO mice stains, hemi-zygous showing no phenotype). The reader should not conclude that the lower cargo degradation in cystinotic cells can be readily explained by the decreased abundance of cathepsin S alone. Other proteases and additional changes (such as change in lysosomal redox level, allowing opening of intramolecular disulfide bonds for further hydrolysis) are certainly key parameters.

We thank the reviewer for this comment and added the information suggested. In the revised manuscript, the text now reads *'Consistent with the reduced cargo degradation in cystinotic cells, we found an overall reduction in lysosomal catalytic-proteins expression including, lysosomal acid lipase (LIPA), lysosomal acid phosphatase (ACP2), and the potentially representative cysteine protease, cathepsin S (CTSS)'*.

Hence LATER :

Line 305 : "demonstrating" is incorrect here. Proteomic analyses did not demonstrate the delay of proteolysis (DQ-BSA experiments did). Replace by "which was consistent with delayed...

We thank the reviewer for pointing this out. We adjusted the text which now reads *'This was confirmed further by metabolic and proteomic analyses, where we found a reduction in lysosomal catabolic proteins and an upregulation of enzymatic antioxidizing agents like catalase and superoxide dismutase in cystinotic cells, which was consistent with delayed lysosomal cargo degradation and increased oxidative stress, respectively'*.

Line 220-4: The two following statements appear contradictory. *Bicalutamide did not restore the high cystine and cysteine levels (thus no change) ... *In line with its metabolic effects...(In line suggesting a related, possibly explanatory change) bicalutamide upregulated enzymes in cysteine conversion (thus a related change). PLEASE CLARIFY.

We thank the reviewer for pointing this out. The "In line with its metabolic effects" in this sentence refers to the other metabolites that did change and may be related to the beneficial effects of bicalutamide. We did not intend to imply that this is related to high cystine and cysteine levels. We therefore adjusted the text which now reads:

'Among the candidate drugs tested, bicalutamide, an anti-androgenic agent, did not restore the high cystine and cysteine levels, however improved the overall metabolic phenotype, including α KG, serine, betaine, and oleoyl-L-carnitine (Fig 4A). Treatment with bicalutamide alone also resulted in the upregulation of several metabolic enzymes involved in cysteine conversion, lysosomal degradation, and the TCA cycle, including AKGDH (Fig 4C-E, and Fig EV4B).'

Lines 315-7. the NEW interpretation of the DMKG toxicity after 24 h specifically in their *CTNS*-deficient cells based on ROS effect is not really convincing for this reviewer. This is an important point since aKG effects are very much context-dependent. In the absence of DMKG, *CTNS*-deficient cells manage well the strong ROS production induced by starvation (and presumably other changes, possibly related to AMPK activation a.o., Fig3B-D), but benefit less (*CTNS*^{-/-}) or not (*CTNS* patient) by DMKG addition. The authors seem to conclude right away that anti-oxidant effect of aKG derived from DMKG is blunted in *CTNS*-deficient cells, which they attribute to hampered glutathione biosynthesis. Yet, Fig 4A (two first bars) shows that in untreated *CTNS*^{-/-} cells, glutathione level is identical to isogenic WT controls, and increases vigorously to cysteamine addition, demonstrating a clear glutathione biosynthesis potential in their system. This should be clearly stated. After which, "We also found" would better start by "However". How aKG imbalance impacts on other pathways which cause abnormal activation of autophagy should be briefly discussed at line 319.

We thank the reviewer for these excellent discussion points. Indeed, limited availability of glutathione is not likely to be involved in the DMKG induced cell death. We adapted the discussion which now reads:

“In our cells, addition of bicalutamide but not cysteamine could prevent the DMKG induced cell death and reduced LC3-II/LC3-I levels. As cysteamine alone is able to efficiently induce glutathione synthesis in cystinotic cells, it is likely that the DMKG decreased cell viability was not due to a reduced availability of antioxidants. The marked increase in LC3-II/LC3-I levels in CTNS deficient cells indicates activation of autophagy. However, the reduced protein degradation (shown here by DQ-BSA analysis and in previously published reports) (Festa et al., 2018; Ivanova et al., 2015) question the efficiency of this autophagic process. The high intracellular levels of α KG together with a defective autophagy may therefore induce glutaminolysis-mediated apoptosis during starvation specifically in cystinotic cells. As bicalutamide is known to upregulate the autophagic flux (Hao et al, 2017), the addition of bicalutamide could restore the autophagic flux in cystinotic cells and prevent this glutaminolysis-mediated apoptosis.”

Line 401 : "normalize" is not correct. Fig 4A shows that bicalutamide not only suppresses the (modest) increase of α KG in *CTNS*^{-/-} cells, but results into a two-fold lower level as compared to untreated WT cells.

We thank the reviewer for this remark. In the revised manuscript, the text now reads *'Bicalutamide, but not cysteamine, was able to reduce α KG levels and resolve α KG-mediated downstream effects in cystinotic cells'*.

MINOR CHANGES

Fig 2D, upper right : replace "CTNS KO" by *CTNS* patient (I think it was wrongly changed during revision; was correct in version 1) Fig3A. define cells lines in italics, as everywhere else Line 273 : replace a non-nephrotoxic "dose" by "concentration" (these are in vitro experiments ; amounts are expressed in microM at line 470) Line 337 : α KG is a (not the) mitochondrial metabolite Line 358 : replace including high "blood" cystine levels by "tissue" (blood cystine was not measured in the following references) Line 388: safe in our ADD "short-term " models (as opposed to long-term toxicity of racemic bicalutamide in human patients).

We thank the reviewer for carefully evaluating our manuscript and pointing out these errors, which we corrected in the revised manuscript.

Referee 3:

This is an innovative investigative approach to characterize the cellular mechanisms responsible for the dysregulation of proximal tubular function in experimental cystinosis models. The authors used immortalized cell culture lines of cystinosis and control tissue, and a unique application of organ tubuloids from cystinosis patients, and finally, a zebrafish model of cystinosis. This combination of experimental approaches with advanced proteomic and metabolomic analyses, provided new and encouraging data on the specific cellular mechanisms responsible for the varying phenotypes and pathophysiology of cystinosis. Indeed, these novel approaches have totally circumvented the numerous shortcomings of the in vivo models of cystinosis where there are limited renal phenotypes.

Question 1:

One minor suggestion would be to possibly expand upon the findings to point to further development of drug interventions for testing in special populations to advance the field.

We highly appreciate the positive remarks by this reviewer. To expand on the finding and to point to further development of our drug interventions, we now included the following text in the discussion section: 'Preclinical studies in a suitable animal model should determine if cysteamine-bicalutamide combination therapy is able to improve or prevent the development of renal Fanconi syndrome and renal failure. Translation to the clinic should be further facilitated by the fact that bicalutamide is already an approved drug with a known safety profile, although a separate study should be performed to determine the appropriate dose for cystinotic patients.'

14th Apr 2021

Dear Dr. Janssen,

Thank you for the submission of your revised manuscript to EMBO Molecular Medicine. I am pleased to inform you that we will be able to accept your manuscript pending the following final amendments:

1) As you will see from the referee report below, there are some concerns that require your attention, however no additional experiments are necessary.

- Question 1: Please tone down the claims as suggested by the referee.
- Question 3A: Please clarify.
- Question 3B: Please amend so that conclusions are supported by the data.

2) In the main manuscript file, please do the following:

- Correct/answer the track changes suggested by our data editors by working from the attached/uploaded document.
- In M&M, please include statement provided in the "Checklist" that the informed consent was obtained from all human subjects and that the experiments conformed to the principles set out in the WMA Declaration of Helsinki and the Department of Health and Human Services Belmont Report.
- In data availability section we noticed that deposited proteomics data are currently private. Please be aware that all datasets should be made freely available upon acceptance, without restriction. Please check "Author Guidelines" for more information.

<https://www.embopress.org/page/journal/17574684/authorguide#availabilityofpublishedmaterial>

3) Please provide a point-by-point letter INCLUDING my comments as well as the reviewer's reports and your detailed responses (as Word file).

I look forward to reading a new revised version of your manuscript as soon as possible.

Yours sincerely,

Zeljko Durdevic

***** Reviewer's comments *****

Referee #1 (Remarks for Author):

Referee: 1

Question 1:

A) Re-Re-Rev: Unfortunately, the rephrased version did not address the issue - the comment was that the authors need to mention the difference in staining observed between CTNS -/- group and CTNS-patient cells rather than putting both in the same basket.

Also, how can one mention about perinuclear localization of lysosomes without electron microscopy? I do not think authors can make such claims based on the data provided.

B) Re-Re-Rev: Thanks for providing the blots. The authors have efficiently clarified this concern.

Question 2:

Re-Re-Rev: Thanks for providing the detailed explanation. This is helpful.

Question 3:

A) Re-Re-Rev: Thanks for providing the explanation. So how increasing the autophagic flux improve impaired lysosomal-cargo degradation? Also, why the authors did not add bafilomycin, DMKG and bicalutamide together to show increased autophagic flux in their cells in fig 5?

B) Re-Re-Rev: Figures 3 H and I, do not match. Blocking with bafilomycin did not increase in LC3-II in diseased versus wild type as shown in figure 3H.

C) Re-Re-Rev: Ok.

Question 4:

Re-Re-Rev: Ok.

The authors performed the requested editorial changes.

We are pleased to inform you that your manuscript is accepted for publication and is now being sent to our publisher to be included in the next available issue of EMBO Molecular Medicine.

Corresponding Author Name: Manoe J. janssen

Manuscript Number: EMM-2020-13067